# Multidecadal Arctic sea ice thickness and volume derived from ice age

Yinghui Liu[1], Jeffrey R. Key[1], Xuanji Wang[2], and Mark Tschudi[3]

[1]Center for Satellite Applications and Research, NOAA/NESDIS, Madison, Wisconsin
[2]Cooperative Institute for Meteorological Satellite Studies, University of Wisconsin, Madison, WI
[3]Colorado Center for Astrodynamics Research, University of Colorado, Boulder, CO

*Correspondence to*: Yinghui Liu (Yinghui.Liu@noaa.gov)

**Abstract.** Sea ice is a key component of the Arctic climate system, and has impacts on global climate. Ice concentration, thickness, and volume are among the most important Arctic sea ice parameters. This study presents a new record of Arctic sea
ice thickness and volume from 1984 to 2018 based on an existing satellite-derived ice age product. The relationship between ice age and ice thickness is first established for every month based on collocated ice age and ice thickness from submarine sonar data (1984-2000) and ICESat (2003-2008), and an empirical ice growth model. Based on this relationship, ice thickness is derived for the entire time period from the weekly ice age product, and the Arctic monthly sea ice volume is then calculated. The ice age-based thickness and volume show good agreement in terms of bias and root-mean-square error with submarine,
ICESat, and CryoSat-2 ice thickness, as well as ICESat and CryoSat-2 ice volume, in February/March and October/November. More detailed comparisons with independent data from Envisat for 2003 to 2010 and CryoSat-2 from CPOM, AWI, and NASA GSFC for 2011 to 2018 show low bias in ice age-based thickness. The ratios of the ice volume uncertainties to the means range from 21% to 29%. Analysis of the derived data shows that the ice age-based sea ice volume exhibits a decreasing trend of -411 km$^3$/year from 1984 to 2018, stronger than the trends from other datasets. Of the factors affecting the sea ice volume trends,
changes in sea ice thickness contribute more than changes in sea ice area, with a contribution of at least 80% from changes in sea ice thickness from November to May and near 50% in August and September, while less than 30% is from changes in sea ice area in all months.

## 1 Introduction

Sea ice plays a key role in regulating the energy and mass exchange between the atmosphere and the underlying ocean
in the polar regions. Over the last few decades Arctic sea ice extent, area, thickness, and volume have declined significantly (Stroeve et al. 2012, Kwok 2019). The corresponding decrease in surface albedo and changes in cloud properties have led to additional surface radiation absorption, which results in further sea ice reduction (Letterly et al. 2018, Perovich et al. 2007, Pistone et al. 2014). The anomalous sea ice export out of the Arctic Ocean (Smedstrud et al. 2011) may have an influence on summer sea ice variability, and the decline of Arctic sea ice may affect the strength of the Atlantic Meridional Overturning
Circulation and thus global climate (Sévellec et al. 2017). Arctic sea ice volume is likely a more sensitive climate change index

than ice extent and area for that the reduction in Arctic sea ice volume is as much as two times that of sea ice extent on a percentage basis in global climate model simulations (Gregory et al. 2002, Solomon et al. 2007). Thus, monitoring Arctic sea ice extent, area, thickness, and volume and their changes is becoming increasingly important in understanding the Arctic and global climate systems and improving climate forecasting.

Satellite remote sensing of sea ice properties is advantageous because of the much higher spatial and temporal coverage in the polar regions compared to in situ observations. Uncertainty in satellite-derived Arctic sea ice extent and area is low overall due to the high quality of sea ice concentration retrievals from passive microwave satellite data. Available since the late 1970s, multiple passive microwave sea ice concentration products are valuable for studying trends in sea ice extent and area in the polar regions (Ivanova et al. 2015). Sea ice concentration from satellite sensors in the visible and infrared spectrum

have the potential to provide additional information owing to their higher spatial resolution (Liu et al. 2016).

     Sea ice thickness products have been generated with the space-based lidar altimeter on the Ice, Cloud, and land Elevation Satellite (ICESat) and with radar altimeters onboard Envisat and CryoSat-2 (Connor et al. 2009, Kwok et al. 2009, Laxon et al. 2013), from passive visible and infrared radiometers using the One-dimensional Thermodynamic Ice Model (OTIM) (Wang et al. 2010), from the Soil Moisture and Ocean Salinity (SMOS) satellite and other passive microwave radiometers (Tian-

Kunze et al. 2014). Sea ice thickness products from ICESat-2, launched in September 2018, will soon be available (Kwok et al. 2016, Markus et al. 2017). Sea ice thickness products from lidar and radar altimeters are available from the early 2000s, and SMOS data are available from the late 2000s. The OTIM ice thickness products cover 1982 to the present using the Advanced Very High Resolution Radiometer (AVHRR) on NOAA polar-orbiting satellites.

     Sea ice thickness is not a physical parameter that satellite visible, infrared, or passive microwave sensors can observe

directly; altimeters provide a more direct measurement. Statistical models or physically based thermodynamic models with numerical parameterizations are needed to retrieve ice thickness with satellite observations (Wang et al. 2010, Kwok et al. 2016, Tian-Kunze et al. 2014). The underlying physical processes controlling ice growth and melt are so complex that uncertainties in the parameterizations in those models lead to large uncertainties in the ice thickness products. For example, the depth of snow on sea ice is a critical parameter for all the ice thickness retrieval methods, and yet currently there is no

direct way to accurately measure it from space, especially for snow on ice (Wang et al. 2010, Lawrence et al. 2018).

     In addition to these satellite ice thickness products, sea ice thickness is also available from regional and global numerical models, e.g. the Pan-Arctic Ice-Ocean Modeling and Assimilation System (PIOMAS) (Zhang and Rothrock 2003, Schweiger et al. 2011, 2019, Lindsay et al., 2012), and global climate models. Although the global climate models tend to underestimate the rate of ice volume loss and represent the thickness spatial patterns poorly, multi-model ensemble means provide realistic

trends (Stroeve et al. 2014).

     Sea ice thickness can also be derived from sea ice age. An Arctic sea ice age product covering the period from 1984 to the present has been generated based on Lagrangian tracking of individual sea ice parcels (Tschudi et al. 2019a). Studies have shown that a generally linear relationship exists between ice age and ICESat sea ice thickness from 2003 to 2008 (Maslanik et al. 2007, Tschudi et al. 2016), and such a relationship has been applied to estimate the sea ice thickness in March extending

back to the early 1980s (Maslanik et al. 2007). The uncertainty of this relationship appears to increase from new ice to older ice, with values ranging from approximately 0.2 to 1.0 m (Figure 2 in Maslanik et al. 2007, and Figure 2 in Tschudi et al. 2016). However, how the relationship between age and thickness varies over the course of the year and over the multi-decadal time series was not considered in that work. If sea ice thickness and sea ice age relationships were available for all months of the years when the ice age data are available, a more comprehensive ice thickness dataset could be created. Furthermore, ice

thickness can be combined with ice concentration data to produce a new ice volume product.

This paper presents Arctic Ocean sea ice thickness and volume from 1984 to 2018 based on an existing sea ice age product. Relationships between ice age and ice thickness are established for all months over the period 1984-2018. Weekly ice thickness is then produced based on the weekly ice age product, followed by the calculation of monthly ice volume. Spatial distributions and temporal trends of the derived sea ice thickness and volume are presented. The ice age-based thickness and

volume dataset from 1984 to 2018 is also compared to existing datasets. These ice thickness and ice volume estimates are a proxy based on ice age, thus are not intended as a direct replacement for sea ice thickness observations.

## 2 Data and Methods

### 2.1 Data

#### 2.1.1 Data for Algorithm Development

A weekly sea ice age product from 1984 to 2018 is available from the National Snow and Ice Data Center (NSIDC, Boulder, Colorado, USA; Tschudi et al. 2019b). The latest version of this product, version 4.0, from 1984 to 2018 is used in this study. The ice age category represents how long in years the sea ice has existed since its first appearance, which is estimated through Lagrangian tracking of the ice from week to week using gridded ice motion vectors (Maslanik et al. 2007, Maslanik et al. 2011, Tschudi et al. 2019b). The weekly ice motion vectors are generated by merging the ice motion vectors from

visible/infrared and passive microwave sensors, International Arctic Buoy Program (IABP) buoys, and the NCEP/NCAR Reanalysis. A parcel's age gains a year if it survives the summer minimum sea ice extent, which means that the ice concentration of a grid cell remains at or above 15% throughout the melt season. Each parcel is tracked independently, and the oldest age of all possible ice parcels within each grid cell is assigned to the cell. An ice age value ranging from 1 up to 16 years (since its first appearance) is assigned to each of 722 by 722 grid cells corresponding to the 12.5 km Equal-Area Scalable Earth

Grid (EASE-Grid) covering the Arctic. Any ice age older than four years is classified as one ice age group in this scheme.

U.S. Navy submarines have collected upward looking sonar (ULS) sea ice draft data in the Arctic Ocean since 1958. Originally classified, the data have been declassified and released according to set guidelines, which include restrictions that positions of the data must be rounded to the nearest five minutes of latitude and longitude, the date is to be rounded to the nearest third of a month, and the data are within an irregular polygon in the Arctic Ocean (NSIDC 1998). Submarine data were

also collected in the SCience ICe EXercise (SCICEX) program. The SCICEX data are not classified so that the precise location

and date are available. All the data are processed to provide ice draft profiles in segments and derived statistics of each segment, including ice draft characteristics (e.g., mean draft thickness), leads, etc. The 1984-2000 submarine ice thickness data from NSIDC are used here, including data from SCICEX93, SCICEX96, SCICEX97, SCICEX98, and SCICEX99 (Figure 1). The irregular polygon outlining the SCICEX data release area (DRA) is shown in Figure 1.

Rothrock et al. (2008, hereinafter RPW08) analyzed these submarine data, derived the ice thickness, and studied the annual cycle of the ice thickness and the interannual change in the mean ice thickness. RPW08 showed that the ice draft, which is the thickness of the ice below the waterline, can be converted to ice thickness using the equation

$$T = 1.107D - f(\tau) \tag{1}$$

where $T$ and $D$ are ice thickness and ice draft, respectively, and $f(\tau)$ is the snow ice equivalent as a function of the decimal

fraction of the year $\tau$. The monthly mean of $f(\tau)$ can be found in Table 4 in RPW08 and is also listed in Table A1 of the Appendix.

The averaged ice thickness over the SCICEX box as a function of year and decimal fraction of the year is derived by RPW08 using

$$T = 1.107[\bar{D} + I(t - 1988) - \bar{I} + A(\tau)] - \bar{f} \tag{2}$$

where $\bar{D}$ is 2.97 m, $t$ is the year, $\bar{f}$ with a value of 0.076 is the annual mean of $f(\tau)$, $A(\tau)$ as the ice thickness annual cycle, $I$(t-1988) as the ice thickness interannual change centered around 1988, and $\bar{I}$ with a value of $-0.12$ m is the mean of $I$:

$$I(t - 1988) = I_1(t - 1988) + I_2(t - 1988)^2 + I_3(t - 1988)^3 \tag{3}$$

where $I_1$=-0.0748, $I_2$=-0.00219, and $I_3$=0.000246. In Eq. 2,

$$A(\tau) = A_{S0} \sin(2\pi\tau) + A_{C0}\cos(2\pi\tau) \tag{4}$$

where $A_{s0}$=0.465, $A_{c0}$=−0.250. Eqs 1, 2, 3, and 4 were derived by RPW08, in which details on these equations are available.. Eq. 2 provides the interannual change ($I(t$−1988)) with the annual cycle ($A(\tau)$) superimposed in the averaged ice thickness over the SCICEX box. It will be used to calculate the monthly mean ice thickness in this study. Rothrock and Wensnahan (2007) determined a positive bias of 0.29 m in the ice thickness derived from submarine ULS data and suggested a bias correction. In this study, we therefore reduce individual ice thickness observations by 0.29 m in all the original submarine observations.

Ice thickness and volume values from ICESat are employed here. In particular, we used the average Arctic sea ice volume and ice thickness from ICESat in February and March 2004 to 2008, and in October and November 2003 to 2007 (Figure 2 and 3 in RK18) to develop the age-based ice thickness algorithm. They are therefore not an independent evaluation/validation dataset.

### 2.1.2 Data for Evaluation/Validation

Monthly sea ice concentration from 1984 to 2017 and daily data in 2018 that were produced with the NASA Team algorithm at 25 km polar stereographic grid were obtained from NASA's Distributed Active Archive Center (DAAC) at NSIDC (Cavalieri et al. 1996). Monthly sea ice concentration for 2018 are calculated from the daily data.

Monthly mean sea ice thickness and volume from PIOMAS version 2.1 (Zhang and Rothrock, 2003, Schweiger et al. 2011) for the period 1984-2018 are used for comparison purposes. PIOMAS couples the Parallel Ocean Program with a 12-category thickness and enthalpy distribution sea ice model in a generalized orthogonal curvilinear coordinate (GOCC) system. PIOMAS has the capability of capturing the basic upper-ocean circulation features in the polar regions and assimilating some observations. Boundary inputs at 45 degrees North latitude come from a global ocean model. Sea ice concentration from passive microwave measurements and sea surface temperature from the NCEP/NCAR Reanalysis are assimilated in the system, with atmospheric drivers from the NCEP/NCAR Reanalysis including wind, surface air temperature, and cloud cover (Schweiger et al. 2011). Monthly mean ice thickness data from 1978 are available in a generalized curvilinear coordinate system covering 45 degrees North and poleward with a grid size of 360 by 120.

Sea ice thickness data generated by OTIM with AVHRR data covers 1982 to the present and is included in the AVHRR Polar Pathfinder-extended (APP-x) dataset (Key, et al., 2016). The OTIM ice thickness data are for both poles at a 25 km EASE2 Grid on a twice daily basis. Initially it was based on the surface energy balance at thermal-equilibrium at the interface between the atmosphere and the ice, which may or may not be covered by snow (Wang et al. 2010). OTIM has gradually evolved into a physical-statistical hybrid model that contains all components of the surface energy budget to estimate sea/lake/river ice thickness. Two parameterization schemes of ice thermal-dynamic and physical-dynamic processes have recently been added to account for ice growth/melt and ice rafting/hummocking processes. It should be noted that the OTIM ice thickness estimates are not available for solar zenith angles between 85 and 91 degrees due to large uncertainties in the input surface albedo, cloud mask, and surface shortwave radiation, or when the ice surface temperature is higher than the freezing point. The accuracy of the input parameters – including snow depth, surface humidity, temperature, and wind – can significantly impact the accuracy of the derived ice thickness. Validation studies of OTIM ice thickness were performed with sea ice thickness measurements from ULS on submarines and moorings, as well as ground measurements. The overall accuracy (mean absolute bias) and uncertainty (root-mean-square difference, RMS) of the OTIM ice thickness is approximately 0.20 m and 0.54 m, respectively, over all types of sea ice (Wang et al., 2010, 2016).

CryoSat-2 Arctic sea ice thickness and volume in February and March, and in October and November from 2011 to 2018 are available from RK18 (Figure 2 and 3 in RK18). They are not directly used in the age-based ice thickness algorithm development and are thus used for validation. Besides the CryoSat-2 values in February and March, and in October and November from RK18, we also calculated the area-averaged monthly mean ice thickness and volume of all months over the Arctic Ocean using three CryoSat-2 ice thickness products from 2011 to 2018, and Envisat ice thickness products from 2003 to 2010 for evaluation/validation purpose. Three monthly mean CryoSat-2 ice thickness products in January, February, March, April, October, November, and December from 2011 to 2018 are available from NASA GSFC (Goddard Space Flight Center) (Kurtz et al. 2014), the AWI (Alfred Wegener Institute) (Hendricks and Ricker 2019), and the CPOM (Centre for Polar Observation and Modelling Data Portal) (Laxon et al. 2013). The NASA GSFC data is available from NSIDC on a 25 km polar stereographic grid (Kurtz and Harbeck 2017). The AWI data on a 25 km EASE2 grid is available at ftp://ftp.awi.de/sea_ice/product/cryosat2/v2p2/nh/l3c_grid/monthly. The CPOM data at 5 km spatial resolution is available at

. Monthly mean Envisat ice thickness on a 25 km EASE2 grid in January, February, March, April, October, November, and December 2003 - 2010 are from the European Space Agency's (ESA) Climate Change Initiative (CCI) version 2 product. They are available at 
165 .

Each of these ICESat and CryoSat-2 ice thickness products has its own uncertainty. The major contributors are uncertainties in snow depth and snow density. The overall uncertainty in ice thickness is estimated to be about 0.7 m (Kwok and Cunningham 2008). Kwok and Rothrock (2009) estimated the ICESat ice thickness uncertainty around 0.37 m. Comparisons with in situ ice thickness observations show unbiased ice thickness estimation in CPOM CryoSat-2 ice thickness,
with uncertainties from 0.34 to 0.66 m, and the uncertainties in Arctic-wide sea ice volume at about 13.5% (Tilling et al. 2017). Comparison of NASA GSFC CryoSat-2 ice freeboard to IceBridge data shows a RMS difference range from 7.4 to 11.1cm in ice freeboard retrievals (Kurtz et al. 2014). The percentages of ice thickness uncertainty to the ice thickness from AWI CryoSat-2 monthly mean ice thickness from 2011-2018 range from around 35% at mean thickness of 1.4 m to around 20% at mean thickness of 5 m (Figure A1 in appendix).
All the products are remapped to a 25 km polar stereographic grid. Area averaged monthly mean ice thicknesses over the Arctic Ocean are calculated for each of these products. Monthly mean Arctic sea ice volume is calculated as the product of sea ice thickness, ice concentration, and grid cell area of all grid cells as explained further in the next section.

## 2.2 Method

The first step is to establish the relationships between ice age and thickness in every month from 1984 to 2018. The
relationships in two months of a year are derived first: April and September from 1984 to 2000 using submarine ice thickness data, and March and October from 2004 to 2008 using ICESat ice thickness data. Ice draft of each segment is converted to ice thickness using Eq. 1. The middle point of each segment is remapped to the 12.5 km EASE-Grid and then is collocated with ice age values at 9 surrounding grid points (including the central point) in the corresponding weekly ice age product. For ice draft segments not from SCICEX, because of the restrictions on revealing the exact date, their observational dates are assigned
to day 5, 15, or 25 when they are in the first, second, and third ten days of a month, respectively. We collocate each ice draft segment with its surrounding nine ice age values from its corresponding weekly ice age product, as well as the week before and after, for a total of 27 ice age values. The final collocated ice age is determined as the ice age at the center of the nine points if it has the same ice age as the majority (>60%) of the nine ice age samples for SCICEX (27 for non-SCICEX). Otherwise, no ice age is determined. All collocated ice thickness and age samples in a month within a 10-year moving window
are used to derive the relationship of ice age and ice thickness in that month at the fifth of the ten years. Only ice draft segments longer than 15 km are included; changing the threshold to 10 km, however, does not change the overall relationship. For each ice age category, a relationship is derived if the number of samples in a month is greater than 40. For example, we started with data in April and September from 1984 to 1993 to obtain the relationships in April and September for 1988 (the middle of the 10 years), and ended with data in April and September from 1991 to 2000 to obtain the relationships in April and September

for 1995. Because the submarine measurements are concentrated in the spring and autumn, meaningful relationships are determined only in April and September.

Using the collocated ice age and thickness from ICESat from 2004 to 2008, Tschudi et al. (2016) derived the relationship between the two for February through April over the Arctic Ocean. We assign this relationship to the month of March from 2004 to 2008. However, information for such a relationship is not available for other months. According to Figure 2 in RK18, the mean ice thickness in October and November is approximately 0.7 m less than the mean in February and March. Therefore, in October from 2004 to 2008 we assign the relationship of ice age and ice thickness to be the same as that in March except that ice thickness in each age category is 0.70 m less.

Figure 2 shows the relationships between ice age and ice thickness in April and September from 1988 to 1995 using submarine measurements, and in March and October from 2004 to 2008 from Tschudi et al. (2016). Older sea ice is generally thicker than younger ice, except that ice more than four years old is slightly thinner than four-year old ice based on submarine measurements before 2000. This phenomenon was observed in one of five years (2008 in 2004-2008) using ICESat data (Tschudi et al. 2016), but it is persistent in most years from 1988 to 1995 in the submarine data. The physical mechanism for this relationship is not clear. Since 1984, for every ice age category, sea ice thickness has generally been decreasing. As in Tschudi et al. (2016), we use linear regression to derive the relationships between ice age and thickness for ice age from one to four years, while the relationship for ice older than four years remains unchanged. Then linear regression on ice thickness from 1988 to 1996 is used to smooth the ice thickness in each age category.

Relationships between ice age and thickness for every month are needed to convert the weekly ice age data into ice thickness. Though we have such relationships in two months of every year from 1988 to 1995, and from 2004 to 2008, relationships for all other months are needed. For this purpose, we apply an empirical model of the annual cycle of ice thickness. In this model, ice thickness increases linearly from September to the following May and decreases linearly from May to September in each sea ice category (Figure 3). The selection of September and May is consistent with the fact that the surface has an energy flux gain from the atmosphere from May to September, and an energy flux loss to the atmosphere from September to the following May (Serreze et al. 2007). From May to September the increase/decrease in sea ice thickness can be approximated by

$$T = G \times M + H_1 \tag{5}$$

where $T$ is monthly mean ice thickness, $G$ is the growth rate with units of m/month, $M$ is month index from May to September, and $H_1$ is a constant (m). From September to the following May,

$$T = D \times M + H_2 \tag{6}$$

where $T$ is monthly mean ice thickness, $D$ is growth/declining rate with unit of m/month, $M$ is month index from September to the following May, and $H_2$ is a constant (m). Given that both equations provide the same results in September and May, and the known relationship of ice age and thickness in April and September from 1988 to 1996, as well as in March and October from 2004 to 2008, we derive $G$, $D$, $H_1$, and $H_2$, thereby determining the relationship between ice age and thickness for every month in those years following Eqs. 5 and 6. For the years before 1988 and after 2008, we use the relationship for 1988 and

2008; for years from 1996 to 2003, we derive the relationship using linear interpolation of the relationship for 1995 and 2004.

Figures 2b and 2d show the derived relationships of ice age and thickness for April and September from 1984 to 2018.

The relationship between ice age and thickness in every month is then linearly interpolated to the weekly scale, and we convert weekly ice age to weekly ice thickness. An example of such a conversion is shown in Figure 4. To calculate the monthly mean ice thickness we determine the daily ice thickness using linear interpolation from the weekly ice thickness and thus calculate the monthly mean.

Monthly mean ice thickness in the 12.5 km EASE-Grid is then remapped to 25 km polar stereographic projection to match the spatial resolution of sea ice concentration. The PIOMAS and OTIM monthly mean ice thickness are also remapped to the same polar projection. Monthly mean Arctic sea ice volume is calculated as the product of sea ice thickness, ice concentration, and grid cell area of all cells over an area defined in RK18. Bounded by the gateways into the Pacific (Bering Strait), the Canadian Arctic Archipelago, and the Greenland (Fram Strait) and Barents Seas, the area covers approximately

$7.23 \times 10^6$ km². We refer to this area as the Arctic Ocean, as in RK18, shown as a polygon in Figure 13. Monthly mean sea ice thickness is also calculated over the DRA, as defined in RPW08. Hereinafter, we call the sea ice thickness and sea ice volume derived from the ice age product as "IceAgeDerived."

## 3 Results

### 3.1 Evaluation of ice thickness and Arctic ice volume

Based on submarine sonar data, RPW08 derived an equation – Eq. 9 in their paper and Eq. 2 here – to calculate the annually averaged ice thickness over the DRA with the annual cycle superimposed. Mean sea ice thickness over the DRA in February and March, as well as in October and November, from 1984 to 2000 are calculated here using this equation. RK18 reported the mean sea ice thickness over the DRA from ICESat in February and March 2004-2008, and in October and November 2003-2007, and from CryoSat-2 in February and March and in October and November 2011-2018. RK18 also

reported monthly mean Arctic ice volume over the Arctic Ocean from ICESat in February and March 2004-2008 and in October and November 2003-2007, and from CryoSat-2 in February and March and in October and November 2011-2018. These data are used to evaluate the quality of sea ice thickness and volume of the IceAgeDerived.

IceAgeDerived sea ice thickness over the DRA is close to the one-to-one line in comparison to ice thickness from submarine in February/March, with a bias of 0.03 m, RMSE of 0.074 m, and R-squared value of 0.96 (Figure 5 and Table 1).

In October/November the bias is -0.035 m, the RMSE is 0.14 m, and the R-squared is 0.97. Compared to ICESat, sea ice thickness over the DRA gives a slightly larger bias and RMSE and slightly smaller R-squared, with a bias of -0.014 m, RMSE of 0.096 m, and R-squared of 0.75 in February/March (Figure 5, Table 1). In October/November, the bias is 0.20 m, the RMSE is 0.16 m, and the R-squared is 0.93. The bias and RMSE values are well within the uncertainty of ICESat ice thickness estimates of 0.37 m (Kwok and Rothrock, 2009). The submarine and ICESat data are used in the development of the

IceAgeDerived product, and thus these evaluations are not independent. Comparison to CryoSat-2 sea ice thickness over DRA

shows a bias of -0.21 m and RMSE of 0.079 m for February/March, and a bias of -0.04 m and RMSE of 0.14 m for October/November. These are comparable in magnitude to those from ICESat, and within the uncertainty of CryoSat-2 ice thickness (Kwok 2018). The R-squared in October/November is near zero (0.037). With the relatively small bias and RMSE, this indicates that the IceAgeDerived sea ice thickness has similar values but does not follow the changes in CryoSat-2 sea ice thickness from 2011 to 2018 (Figure 5 and Table 1). Comparing the results of PIOMAS to submarine and ICESat thickness in Table 1 show similar bias and RMSE results as those in Schweiger et al. (2011).

Measurements of IceAgeDerived sea ice volume over the Arctic Ocean agree with those from ICESat in February/March, with a bias of $-0.72 \times 10^3$ km$^3$, RMSE of $0.74 \times 10^3$ km$^3$, and R-squared of 0.87 (Figure 6 and Table 2). In October/November, IceAgeDerived sea ice volume is a large underestimate compared to ICESat, with a bias of $-3.95 \times 10^3$ km$^3$, even though IceAgeDerived sea ice thickness measurements agree well with those from ICESat over DRA. Similar underestimations in October/November are found for PIOMAS and OTIM when compared to ICESat. Comparison to CryoSat-2 sea ice volume shows low bias and low RMSE, where the bias is $0.29 \times 10^3$ km$^3$ ($-0.66 \times 10^3$ km$^3$) and the RMSE is $0.75 \times 10^3$ km$^3$ ($0.98 \times 10^3$ km$^3$) in February/March (October/November).

Comparisons of sea ice thickness over DRA and sea ice volume over the Arctic Ocean from PIOMAS and OTIM to submarine ULS, ICESat and CryoSat-2 are also shown in Figures 5 and 6, and in Tables 1 and 2. IceAgeDerived products show comparable or slightly better results in terms of bias, RMSE, and R-squared. The better agreement with submarine ULS can be attributed to the fact that the IceAgeDerived product is developed based on matched ice age and submarine ULS ice thickness data, and collocated ice age and ICESat thickness data. However, it should be noted that while submarine data in April and September are used in the algorithm development, the comparisons are in February/March and October/November.

A comprehensive assessment of IceAgeDerived ice thickness and ice volume with those from CryoSat-2 is carried out for the period 2011-2018. The CryoSat-2 ice thickness and ice volume from NASA GSFC, AWI, and CPOM are not used in the algorithm development, and thus provide independent evaluation/validation. Figures 7 and 8 show the results, with statistics given in Table 3. The IceAgeDerived has slightly smaller monthly ice thickness and volume compared to AWI CryoSat-2 products in most months, with overall ice thickness mean bias (standard deviations) of -0.02 m (0.11 m) and overall ice volume mean bias of $-0.76 \times 10^3$ km$^3$ ($0.86 \times 10^3$ km$^3$). Comparison to NASA GSFC CryoSat-2 products shows the largest negative bias in those months among the three, with overall mean bias (standard deviations) of -0.27 m (0.15 m) and $-1.79 \times 10^3$ km$^3$ ($0.95 \times 10^3$ km$^3$) for ice thickness and ice volume respectively. The negative biases to CPOM CryoSat-2 products are in between.

Though the comparison to the CryoSat-2 ice products shows overall agreement in both thickness and volume, further investigation and analysis shows that there are differences in the ice thickness retrieval spatial distributions, as shown in Figure 9. It appears the IceAgeDerived ice thickness underestimates the ice thickness for the older ice, while it overestimates ice thickness for the new ice when compared to CryoSat-2. The underestimation of ice thickness north of the Canadian Archipelago and the Greenland from IceAgeDerived may be attributed to the lower sensitivity of sea ice age–thickness towards older sea ice, as will be discussed later and shown in Figure 12. This reduction in sensitivity may come from higher uncertainty with older sea ice age because of higher uncertainty with longer Lagrangian tracking of sea ice parcels, at least in theory. Such an

295 uncertainty estimation is not available in the current sea ice age product. This reduction in sensitivity may also be related to the fact that the oldest age of all possible ice parcels within each grid cell is assigned to the cell, and thus the ice age may overestimate the sea ice age of some cells. It should be noted that CryoSat-2 also has relatively high uncertainties for very thin and very thick sea ice. In total, these underestimates and overestimates may offset each other in the overall mean ice thickness and ice volume comparisons. Diagnosing and resolving this difference will be done in the future.

Similar evaluations and validation are carried out through a comparison to Envisat from 2003 to 2010. Figures 10 and 11 are scatterplots of the results, with statistics given in Table 4. The monthly mean ice thickness shown in the figures is the mean ice thickness of all pixels in a month. It shows that the monthly IceAgeDerived thickness and volume are comparable to the ESA CCI Envisat products in all months, with overall mean biases (standard deviations) of 0.07 m (0.10 m) and -0.08×$10^3$ km$^3$ (0.57×$10^3$ km$^3$).

The monthly mean CryoSat-2 ice thickness from CPOM, AWI, and NASA GSFC from January to April, and from October to December of 2011 to 2018 are used to calculate the spread of CryoSat-2 ice thickness within each ice age category as those in Tschudi et al. (2016). The collocated NSIDC weekly ice age with CryoSat-2 monthly ice thickness from all available months over the period 2011 to 2018 can be used to derive such spreads in all months, as shown for March and November in Figure 12. Ice thickness increases with ice age for ages from 1 to 4 years and then decreases from ages 4 to 5. This is consistent 310 with what was found based on upward looking sonar data. In Tschudi et al. (2016) ice thickness increases from ice age from 1 to 5. Similar to those in Tschudi et al. (2016) (Figure 2 in their paper), one standard deviation of the probability distribution function of CryoSat-2 thickness in an age category overlaps with adjacent age categories. The overlap may be a result of mismatches in the collocation of weekly ice age with monthly ice thickness.

To estimate the random uncertainty of the IceAgeDerived ice volume over the Arctic Ocean we apply the ice thickness 315 uncertainty errors in each ice age category (Figure 12) when converting the weekly ice age to ice thickness from 1984 to 2018. The uncertainty in weekly or monthly ice volume over the Arctic Ocean is the sum of the ice volume uncertainty of all grid cells, where the ice volume uncertainty in a cell is the product of the sea ice concentration, the grid cell area, and the ice thickness uncertainty. This provides the upper limit on the random uncertainty in ice volume. The overall uncertainties in ice thickness and ice volume in every month from 1984 to 2018 are calculated. The average ratios of these ice volume uncertainties 320 to the monthly means range from 21% to 29% over the period 1984 - 2018.

**3.2 Sea ice thickness and volume climatology and trend**

The spatial distributions of the IceAgeDerived ice thickness over the Arctic from 1984 to 2018 show similar spatial patterns, but different magnitudes in the four seasons (Figure 13). Sea ice is thickest along the northern portion of the Canadian Archipelago and Greenland, decreasing radially, with the thinnest ice over the Arctic's peripheral seas on the Eurasia side. The 325 thickest sea ice appears in the spring, around 3 m in the Canada Basin and North Pole areas. The thinnest sea ice is in early fall, around or less than 1 m over the coastal areas of the Kara, Laptev, and Chukchi Seas. The spatial distributions of PIOMAS

and OTIM (Figures A2 and A3 in the Appendix) show similar patterns, while the ice thickness north of the Canadian Archipelago and Greenland is thicker, especially in PIOMAS.

The annual cycle of monthly mean sea ice volume over the Arctic Ocean shows a minimum value in September at around 6770 km$^3$, then increasing to the maximum value in the following May at around 21737 km$^3$, followed by a decrease. This annual cycle is certainly affected by the model used to depict the ice growth/melt as shown in Figure 3. The annual cycle closely follows the sea ice volume annual cycle of the PIOMAS (Figure 14), which uses a different approach to derive ice thickness and ice volume. Compared to PIOMAS, the IceAgeDerived sea ice volume exhibits its difference of 2004 km$^3$ in May. This difference can be attributed to the relatively thicker sea ice from the IceAgeDerived in the years before 2000, which is discussed further below. Ice volume over the Arctic Ocean from OTIM has a similar annual cycle but with a larger magnitude, with the maximum in April and the minimum in September.

The time series of sea ice thickness over DRA in February and March from 1984 to 2018 shows a decreasing trend from 1984 to 2000, a generally decreasing trend from 2004 to 2008, and a relatively unchanging state from 2011 to 2018 (Figure 15a). This is consistent with the decreasing trend in the submarine ULS data from 1984 to 2000 and from ICESat for 2004 to 2008, and the relatively stable state from CryoSat-2 2011 to 2018 as also seen in Haas et al. (2017). A similar conclusion can be drawn for the time series in October and November (Figure 15b). The overall decreasing trends are consistent with observations of the replacement of multiyear sea ice with first year ice in the Arctic Ocean, and partial recovery of multiyear sea ice after the summer of 2008 (Maslanik et al. 2007, Maslanik et al. 2011). This agreement can be attributed to the fact that the sea ice age information in the ice age product, including intrinsic features of general decreasing and partial recovery of multiyear sea ice after 2008, is utilized to derive the ice thickness. Compared to the PIOMAS ice thickness, in February/March, the IceAgeDerived sea ice thickness in the 1980s is mostly greater, close or smaller from 2004 to 2008, and smaller from 2011 to 2018. In October/November the sea ice thickness is greater in the 1980s, comparable from 1990 to 2010, and then larger afterwards. OTIM shows smaller ice thicknesses than both IceAgeDerived and PIOMAS in October/November, and mostly larger ice thickness in February/March except in the 1980s.

The similarities and differences found here are consistent with the results shown in Figure 5 and Table 1, and partly explain the differences in the sea ice volume annual cycles shown in Figure 14. As a result of the differences in ice thickness from 1984 to 2018, the overall trends of ice thickness over the DRA from 1984 to 2018 are -0.054, -0.035, and -0.036 m/year in February/March, and -0.040, -0.042, and -0.026 m/year in October/November for IceAgeDerived, PIOMAS, and OTIM respectively, with significance levels all higher than 95%. The time series of PIOMAS and their comparisons with ICESat shown here are similar to those in Schweiger et al. (2011).

Time series of sea ice volume over the Arctic Ocean show generally decreasing trends from 1984 to around 2008, and relatively stable conditions from 2011 to 2018 both in February/March and October/November, similar to the time series from PIOMAS and OTIM (Figure 16). This overall decrease agrees well with the dramatic decrease in sea ice extent and disappearance of multiyear sea ice reported in the literatures (Stroeve et al. 2012, Maslanik et al. 2007, Maslanik et al. 2011). In February/March, PIOMAS shows smaller ice volume from 1984 to 2000 and similar values after 2000; OTIM shows higher

ice volume after the 1990s. In October/November, PIOMAS shows smaller values in the 1980s and similar values afterwards, while OTIM shows consistently smaller ice volume before 2000. All three sea ice volumes are much lower than those from ICESat for 2003 to 2007 with comparable sea ice thickness over the DRA in those years; all three sea ice volumes are comparable to that from CryoSat-2, with similar results for sea ice thickness over the DRA. All these findings are consistent with what is shown in Figure 6 and Table 2.

The overall trends in ice volume over the Arctic Ocean from 1984 to 2018 are -474, -258, and -311 km$^3$/year in February/March, and -342, -305, and -230 km$^3$/year in October/November for IceAgeDerived, PIOMAS, and OTIM respectively, with significance levels all higher than 95%. IceAgeDerived shows stronger ice volume reduction over the Arctic Ocean in February/March and in October/November when compared to PIOMAS and OTIM.

Over the Arctic Ocean from 1984 to 2018, sea ice volume has been decreasing in every month of the year based on the IceAgeDerived product (Figure 17). The most reductions in volume from December to June occur from the 1990s to the 2000s and from the 2000s to 2010s. From July to November, the volume reductions from the 1980s to 1990s are comparable to those from the 1990s to 2000s. The volume reductions in all months are the least from the 2000s to the 2010s. It should be noted that the data in the 1980s starts in 1984, and the data for the 2010s ends in 2018. Though the decadal mean annual cycles of sea ice volume are similar in shape, the magnitudes of the cycles - in terms of the difference between April and September - have been decreasing, with around 18871 km$^3$ in the 1980s and 12169 km$^3$ in the 2010s.

Time series of the annual mean sea ice volume of all months over the Arctic Ocean from 1984 to 2018 have similar features to those in February/March and October/November, with higher values in the 1980s than those of PIOMAS and OTIM, a generally decreasing trend from 1984 to 2008 as with PIOMAS and OTIM, and relatively stable conditions from 2011 to 2018, similar to PIOMAS and OTIM (Figure 18). As shown, the IceAgeDerived sea ice volume trends are higher than those from PIOMAS and OTIM in every month, except being comparable to PIOMAS from August to October (Figure 19). The monthly trends exhibit an annual cycle, with the maximum magnitude in May at -537 km$^3$/year and minimum magnitude in September of -251 km$^3$/year, which is the opposite of the annual cycle trend of mean sea ice thickness. OTIM also exhibits this feature, while the annual cycle of volume trends from PIOMAS shows no apparent monthly differences. The mean monthly trend of all months over the Arctic Ocean from 1984 to 2018 is -411 km$^3$/year, which is higher in magnitude compared to -282 km$^3$/year from PIOMAS and -269 km$^3$/year from OTIM, with significance levels all higher than 95%. The PIOMAS mean monthly trend is similar to that derived from PIOMAS sea ice volume data for 1979 to 2012, -2.8×103 km$^3$/decade with an uncertainty of 1.0×103 km$^3$/decade as shown in Schweiger et al. (2011). The IceAgeDerived ice volume shows a stronger reduction in ice volume over the Arctic Ocean from 1984 to 2018.

Causes for the changes in the Arctic sea ice volume can be partitioned roughly into two categories: changes in sea ice thickness and changes in sea ice area. In a manner similar to that used by Liu et al. (2009), this partitioning can be estimated by:

$$\frac{dV}{dt} = \frac{d(\sum A_i H_i)}{dt} \cong \frac{d(\bar{A}\bar{H})}{dt} = \bar{A}\frac{d\bar{H}}{dt} + \bar{H}\frac{d\bar{A}}{dt} \qquad (7)$$

where $V$ is the sea ice volume over the Arctic Ocean, $A_i$ and $H_i$ are the sea ice area and thickness in individual grid cells over the Arctic Ocean, and $\bar{A}$ and $\bar{H}$ are the mean sea ice area and thickness over the Arctic Ocean. The term $\bar{A}(d\bar{H}/dt)$ represents the contribution of sea ice thickness changes to the overall trend, and the term $\bar{H}(d\bar{A}/dt)$ represents the contribution of the sea ice area changes. For the Arctic sea ice volume from 1984 to 2018, the changes in sea ice thickness contribute to approximately 80% or more of the total trends from November to May; these contributions decrease to around 50% in August and September (Figure 20). The changes in sea ice area contribute to less than 30% of total trends in all months, with even lower contributions from December to May, which are less than 10%. PIOMAS shows similar features, while OTIM shows a greater contribution from the sea ice area changes and less contribution of sea ice thickness changes from June to October. It should be noted that the sum of these two contributions is not 100% because the production of area means of thickness and ice area is only approximately equal to the total ice volume as shown in Eq. 7.

Figure 21 shows the time series of mean ice volume for 1984 to 2018 using the varying ice age-thickness relationships (as in Figure 16), using fixed relationships in 1984 for the entire time series, and using fixed relationships in 2004-2008 for the entire time series (ICESat period). The overall trends are -411, -136, -156 km$^3$/year from 1984 to 2018, respectively. This indicates that the replacement of multi-year ice by younger ice might only account for a relatively smaller part of the overall trend (~33% or ~38%, -136/-411 or -156/-411), while the changes in ice age and ice thickness relationships contribute more to the overall trend. Since the ice age-thickness relationship change is small between the ICESat period and the CryoSat-2 period (see Figure 12 here and Figure 2 in Tschudi et al. 2016), larger changes in the ice age-thickness relationship may occur primarily between the mid-1980s and mid-2000s, when ice thickness decreases in each corresponding ice age category.

Sea ice extent in September has been decreasing, with a trend from 1997 to 2014 four times as large as that from 1979 to 1996 (Serreze and Stroeve, 2015). More solar heating that the ocean absorbs through the open water area is expected to thin the remaining ice for all ice categories, leading to even less sea ice and more solar heating. This may explain the decreasing ice thickness for corresponding ice ages. However, it appears that the accelerated decrease of ice thickness to corresponding ice age happens before the accelerated decreasing ice extent in September, which needs further investigation.

## 4 Discussion and Conclusions

In this study, a multi-decadal Arctic sea ice thickness dataset covering the period 1984 to 2018 is created from an existing satellite-derived ice age product. The relationship between ice age and ice thickness is first established based on submarine upward-looking sonar ice draft observations from 1984 to 2000, and ICESat ice thickness from 2003 to 2008. Both are available for only two calendar months. Therefore, an empirical model of the annual cycle of sea ice thickness growth/melt is used to derive the ice age and ice thickness relationship for every month from 1984 to 2018. Sea ice volume over the Arctic Ocean is then calculated from ice thickness and concentration. These ice thickness and volume estimates are a proxy from ice age products, thus they are not a direct replacement for sea ice thickness observations. Comparisons of the time series of derived ice thickness and ice volume with those from the literature and other datasets using different approaches show general

similarities but with some notable differences. The similarities prove the soundness of the ice aged-based ice thickness and ice volume dataset, while the differences indicate there is room for further improvement in all the ice thickness datasets.

The major findings of this study include:

- Sea ice thickness derived from ice age ("IceAgeDerived") over the submarine data release area (DRA) shows good agreement with ice thickness from submarine (ULS), ICESat, and CryoSat-2 in both February/March and October/November from RK18, with low bias and RMSE and a high R-squared, except for a near-zero R-squared with CryoSat-2 in October/November. IceAgeDerived sea ice volume over the Arctic Ocean shows good agreement with that from ICESat and CryoSat-2. Compared to ICESat, it has a low bias and RMSE and a high R-squared in February/March. In October/November it has a high negative bias, low RMSE, and high R-squared. Compared to CryoSat-2, it has low bias, RMSE, and R-squared values in both February/March and October/November.

- More detailed comparisons with monthly ice thickness from Envisat 2003-2010 and from CryoSat-2 from AWI, CPOM, and NASA GSFC reveal low biases in the IceAgeDerived ice thickness and volume. The ratios of the ice volume uncertainties to the means range from 21% to 29% over the period 1984 to 2018. Spatially, there is a substantial underestimation over the area north of the Canadian Archipelago and the Greenland compared to CryoSat-2. There are noticeable spreads in the CryoSat-2 ice thickness retrievals and derived ice volume from different products, e.g. AWI, CPOM, and NASA GSFC.

- Sea ice is thickest north of the Canadian Archipelago and Greenland, decreasing radially, with the thinnest ice over the Arctic's peripheral seas on the Eurasia side of the Arctic Ocean. Sea ice volume over the Arctic Ocean has its minimum value in September, increasing to a maximum value in the following May.

- In both February/March and October/November, the time series of sea ice thickness over the DRA from the IceAgeDerived shows a decreasing trend from 1984 to 2000, a generally decreasing trend from 2003 to 2008, and a relatively stable state from 2011 to 2018.

- Sea ice volume over the Arctic Ocean shows a generally decreasing trend from 1984 to around 2008, and relatively stable conditions afterwards in almost every month. The mean monthly trend of all months from 1984 to 2018 is -411 km$^3$/year, which shows a stronger ice volume reduction than PIOMAS (-282 km$^3$/year) and OTIM (-269 km$^3$/year). This difference can be attributed to the higher sea ice volume over the Arctic Ocean from the IceAgeDerived in the 1980s.

- Over the Arctic Ocean, changes in sea ice thickness contribute 80% or more to the sea ice volume trend from 1984 to 2018 from November to May, decreasing to a contribution of about 50% in August and September. The changes in sea ice area contribute less than 30% to the trends in all months, with even lower contributions from November to May.

Although the ice thickness and volume dataset presented here is a consistent and accurate multidecadal product, there are potential areas for improvement. First, a linear relationship between ice age and ice thickness is assumed, which may not be

strictly valid. Of particular interest is the observation that submarine data shows a slightly thinner ice thickness for ice more than four years old than that for four-year old ice, though ICESat shows the same in only one year. To determine whether this relationship is valid, other collocated ice age and ice thickness data, e.g. from the recently launched ICESat-2 (Markus et al. 2017) should be analyzed. Second, the annual cycle of ice thickness growth/melt is assumed to be linear from September to the following May and from May to September, which may not be valid. RPW08 conceptualized sea ice growth and melt as a sine function. A more sophisticated model of the annual cycle of sea ice growth/melt may be needed in deriving the ice age and ice thickness relationship. The annual cycle of trends in ice volume over the Arctic Ocean appears to be opposite to the annual cycle of ice growth, which suggests that this trend feature may be related to the use of a linear sea ice growth/melt model. How they are related and whether a more sophisticated model would remove this feature requires further investigation.

Third, in deriving the relation of ice age to ice thickness in the years before 2000, only ice draft measurements from submarine ULS over the DRA, e.g. over or near the central Arctic Ocean, are available. The derived relationship may be skewed to higher ice thicknesses. Thus, Arctic ice volume derived in this study before 2004 might be overestimated. Correcting this relationship requires more spatially representative ice thickness measurements, or a well-designed parameterization scheme. The ice age-thickness relationship is not available for months other than March, and we assumed that such a relationship is the same in October but with an ice thickness of 0.7 m less. With CryoSat-2 ice thickness available from October to April, we can derive the ice age-thickness relationship in all these months and assess the linear ice thickness growth/melt assumption. Fourth, although the weekly ice age product is converted to weekly ice thickness and interpolated to daily ice thickness for monthly mean calculation, the daily ice age product lacks detailed temporal and spatial information and is not intended for direct comparison to point in situ ice thickness or other daily ice thickness products.

In general, future improvements in ice thickness estimation may require work on improving our understanding and parameterizations of the forcing and physical processes controlling the ice growth and melt, reducing uncertainties in the ancillary data required for ice thickness estimation, collecting extensive temporally and spatially representative ice thickness measurements for better evaluation, designing new models or approaches to estimate ice thickness. More specifically, snow depth over sea ice is one of the key parameters in sea ice thickness retrieval for all existing satellite datasets. Though progress has been made in reducing the uncertainties in estimating snow depth from space, its uncertainty remains high (Lawrence et al. 2018, Shalina et al. 2018). One major challenge for improving sea ice thickness retrievals is the lack of "truth" validation datasets. Because of the severe environmental conditions in the polar regions, in situ ice thickness measurements are scarce, which limits our ability to identify the issues in current datasets and to make further improvement. Ice thickness products using new approaches may provide additional evaluation of existing products. A better overall product benefits from all the above-mentioned efforts and may come as an ensemble of multiple ice thickness products if we know the limitations and strengths of each dataset.

**5 Code availability**

Code in Interactive Data Language (IDL) to process the input data, to generate the datasets, and to analyze the datasets is available upon request from Y.L.

**6 Data availability**

Data used to generate the ice thickness and ice volume datasets are available from the National Snow and Ice Data Center (NSIDC) as detailed in the manuscript. The derived ice thickness and ice volume datasets are available upon request from Y.L.

**7 Author contributions**

Y.L. and J.K. conceived the idea of this study. Y.L. analyzed the data and generated the ice thickness and ice volume datasets, analyzed the results, and wrote the manuscript with contributions from all the co-authors. J.K. provided valuable
guidance on the work and editing of the manuscript. X.W. provided and advised on the usage of the OTIM ice thickness. M.T. advised on the usage of the ice age data. All authors assisted in writing, editing, and revising the manuscript.

**8 Competing interests**

The authors declare that they have no competing interests.

**9 Acknowledgments**

This work was supported by the NOAA National Centers for Environmental Information (NCEI) Climate Data Records Program and the Joint Polar Satellite System (JPSS) Program Office. Y.L. would like to thank Ms. Leanne Avila for editing the manuscript. The views, opinions, and findings contained in this report are those of the author(s) and should not be construed as an official National Oceanic and Atmospheric Administration or U.S. Government position, policy, or decision.

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

**Table 1**: Statistics of comparison of ice thickness from IceAgeDerived, PIOMAS, and OTIM and from submarine upward-looking sonar 1984-2000, ICESat 2004-2008, and CryoSat-2 2011-2018 in February/March (top row), and October/November (bottom row) over the SCICEX data release area. Correlation squared with higher than 95% confidence level is in bold.

| Ice thickness | | Submarine up-looking sonar 1984-2000<br><br>Feb/Mar<br>(Oct/Nov) | ICESat<br><br>2004-2008<br><br>Feb/Mar<br>(Oct/Nov) | CryoSat-2<br><br>2011-2018<br><br>Feb/Mar<br>(Oct/Nov) |
|---|---|---|---|---|
| IceAgeDerived | Bias (m) | 0.03<br><br>-0.035 | -0.014<br><br>0.20 | -0.21<br><br>-0.04 |
| | RMSE (m) | 0.074<br><br>0.14 | 0.096<br><br>0.16 | 0.079<br><br>0.14 |
| | $R^2$ | **0.97**<br>**0.99** | 0.75<br>**0.93** | **0.65**<br>0.037 |
| PIOMAS | Bias (m) | -0.16<br><br>-0.055 | 0.12<br><br>0.14 | -0.10<br><br>-0.24 |
| | RMSE (m) | 0.31<br><br>0.30 | 0.16<br><br>0.097 | 0.13<br><br>0.16 |
| | $R^2$ | **0.50**<br>**0.61** | 0.32<br>**0.94** | 0.079<br>0.031 |

| OTIM | Bias (m) | 0.16 | 0.49 | 0.21 |
| | | -0.60 | -0.13 | -0.37 |
| | RMSE (m) | 0.26 | 0.16 | 0.21 |
| | | 0.28 | 0.22 | 0.22 |
| | $R^2$ | **0.73** | 0.30 | 0.41 |
| | | **0.87** | **0.95** | 0.42 |

**Table 2**: Statistics of comparison of Arctic ice volume from IceAgeDerived, PIOMAS, and OTIM to ICESat 2004-2008, and CryoSat-2 2011-2018 in February/March (top row), and October/November (bottom row) over the Arctic Ocean. Correlation squared with higher than 95% confidence level is in bold.

| Ice volume | | ICESat 2004-2008 Feb/Mar (Oct/Nov) | CryoSat-2 2011-2018 Feb/Mar (Oct/Nov) |
|---|---|---|---|
| IceAgeDerived | Bias ($10^3$ km$^3$) | -0.72 | 0.29 |
| | | -3.95 | -0.66 |
| | RMSE ($10^3$ km$^3$) | 0.74 | 0.75 |
| | | 0.76 | 0.98 |
| | $R^2$ | **0.87** | 0.28 |
| | | **0.95** | 0.051 |
| PIOMAS | Bias ($10^3$ km$^3$) | 0.44 | 0.90 |
| | | -4.21 | -1.70 |
| | RMSE ($10^3$ km$^3$) | 0.98 | 0.96 |
| | | 0.68 | 0.98 |
| | $R^2$ | 0.64 | 0.14 |
| | | **0.93** | 0.19 |

| | | | |
|---|---|---|---|
| OTIM | Bias ($10^3$ km$^3$) | 4.20 | 3.87 |
| | | -4.86 | -1.63 |
| | RMSE ($10^3$ km$^3$) | 1.20 | 1.48 |
| | | 0.96 | 1.23 |
| | $R^2$ | 0.38 | 0.011 |
| | | **0.96** | 0.012 |

**Table 3**: Differences of monthly ice thickness and ice volume between IceAgeDerived and CryoSat-2.

| | | AWI | NASA GSFC | CPOM |
|---|---|---|---|---|
| Comparison of monthly ice thickness of IceAgeDerived and CryoSat-2, 2011-2018 mean (standard deviation) in m | Mean | -0.02 (0.11) | -0.27 (0.15) | -0.18 (0.09) |
| | January | 0.02 (0.09) | -0.24 (0.12) | -0.17 (0.08) |
| | February | -0.03 (0.11) | -0.27 (0.13) | -0.21 (0.10) |
| | March | -0.06 (0.09) | -0.30 (0.11) | -0.24 (0.07) |
| | April | -0.03 (0.08) | -0.14 (0.11) | -0.14 (0.06) |
| | October | 0.00 (0.16) | -0.27 (0.22) | -0.14 (0.12) |
| | November | -0.03 (0.12) | -0.35 (0.14) | -0.19 (0.11) |
| | December | 0.01 (0.10) | -0.29 (0.14) | -0.18 (0.09) |
| Comparison of monthly ice volume of IceAgeDerived and CryoSat-2, 2011-2018 mean (standard deviation) in $10^3$ $km^3$ | Mean | -0.76 (0.86) | -1.79 (0.95) | -0.98 (0.81) |
| | January | -0.46 (0.64) | -1.89 (0.80) | -0.95 (0.51) |
| | February | -1.03 (0.87) | -2.12 (0.94) | -1.35 (0.68) |
| | March | -1.61 (0.74) | -2.39 (0.76) | -1.79 (0.68) |
| | April | -1.38 | -1.37 | -1.35 |

|  |  | (0.59) | (0.83) | (0.55) |
|---|---|---|---|---|
|  | October | -0.11 | -0.68 | -0.05 |
|  |  | (0.66) | (0.73) | (0.66) |
|  | November | -0.46 | -1.94 | -0.80 |
|  |  | (0.76) | (0.87) | (0.71) |
|  | December | -0.35 | -1.79 | -0.98 |
|  |  | (0.75) | (0.95) | (0.81) |


**Table 4**: Comparison of monthly ice thickness and ice volume between IceAgeDerived and Envisat.

| | | ESA CCI Envisat |
|---|---|---|
| Comparison of monthly ice thickness of IceAgeDerived and Envisat, 2003-2010 mean (standard deviation) in m | Mean | 0.07 (0.10) |
| | January | 0.08 (0.06) |
| | February | -0.00 (0.06) |
| | March | -0.00 (0.06) |
| | April | 0.04 (0.05) |
| | October | 0.24 (0.11) |
| | November | 0.06 (0.05) |
| | December | 0.05 (0.05) |
| Comparison of monthly ice volume of IceAgeDerived and Envisat, 2003-2010 mean (standard deviation) in $10^3$ km$^3$ | Mean | -0.08 (0.57) |
| | January | 0.05 (0.34) |
| | February | -0.23 (0.28) |
| | March | -0.84 (0.44) |
| | April | 0.67 (0.24) |
| | October | 0.23 (0.23) |
| | November | 0.13 (0.31) |
| | December | -0.09 (0.57) |


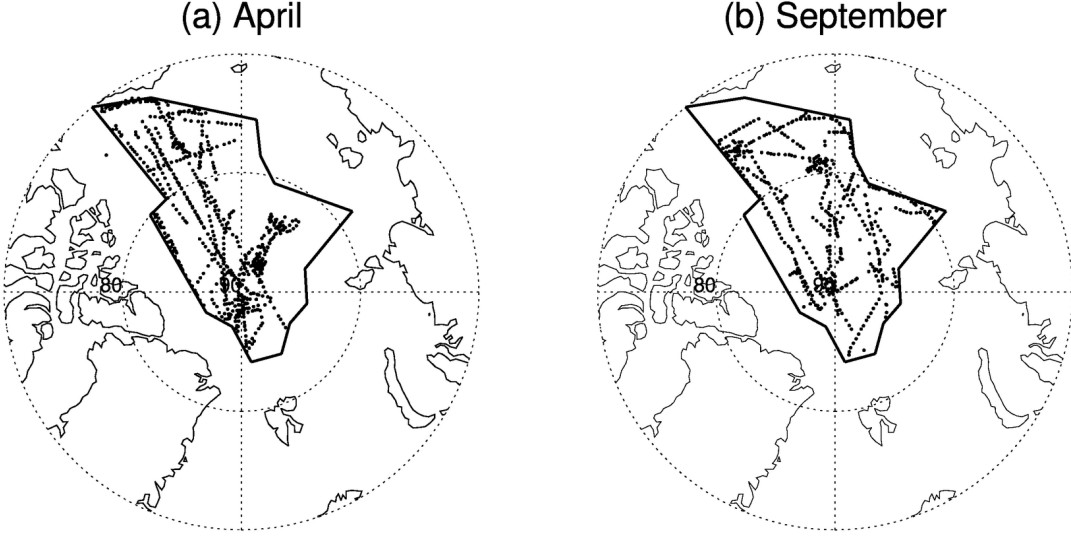

**Figure 1**: U.S. submarine sea ice draft observations in April (a) and September (b) over the Arctic Ocean from 1984 to 2000. The irregular polygon outlines the SCICEX data release area.


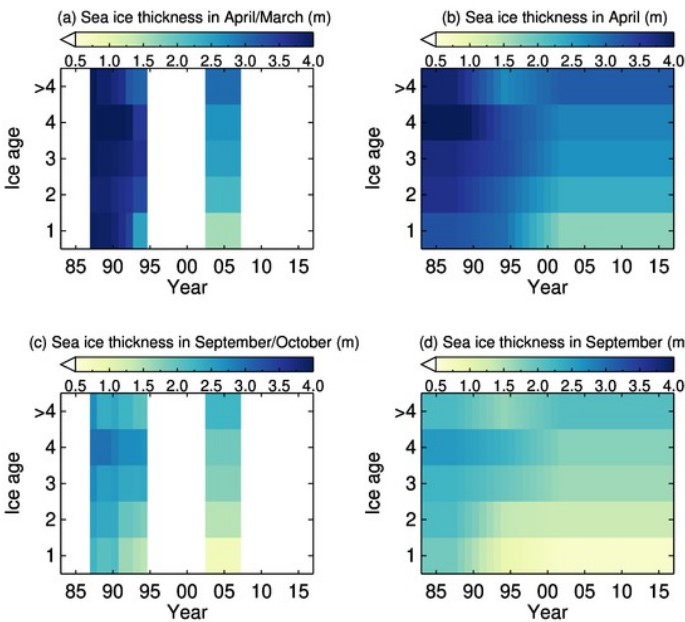

**Figure 2**: Observed relationship of ice age and ice thickness from 1988 to 1995 from submarine data in April (a) and September (c), and from 2004 to 2008 from ICESat in March (a) and October (c), and derived relationship of ice age and ice thickness from 1984 to 2018 in April (b) and September (d).

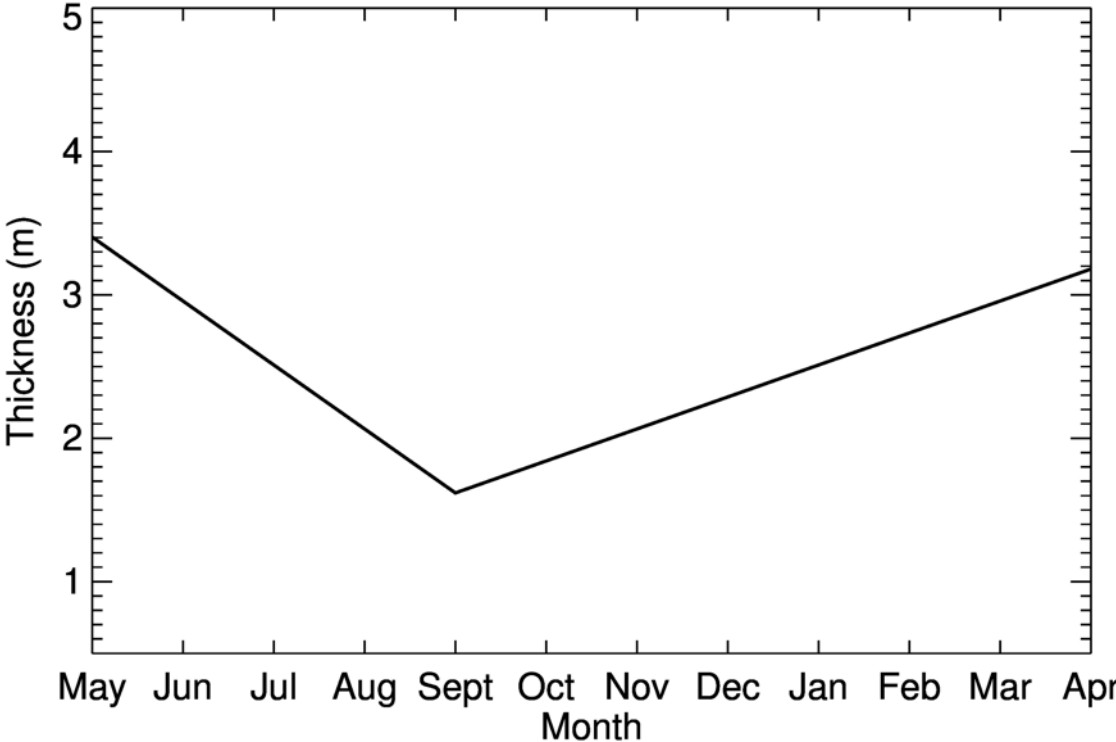

**Figure 3**: Annual cycle of sea ice thickness.

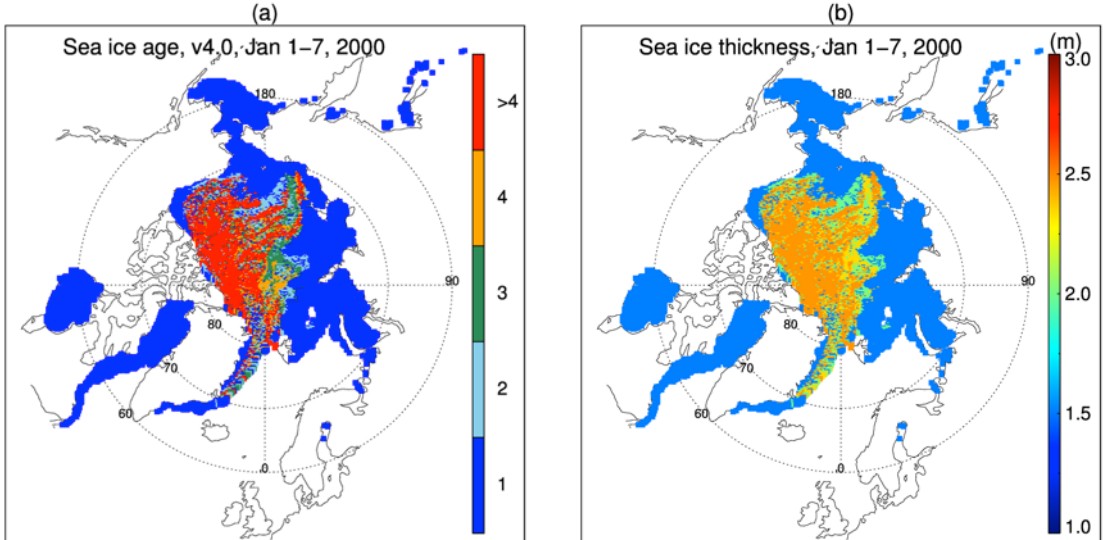

**Figure 4**: Ice age and ice thickness derived from ice age January 1-7, 2000.

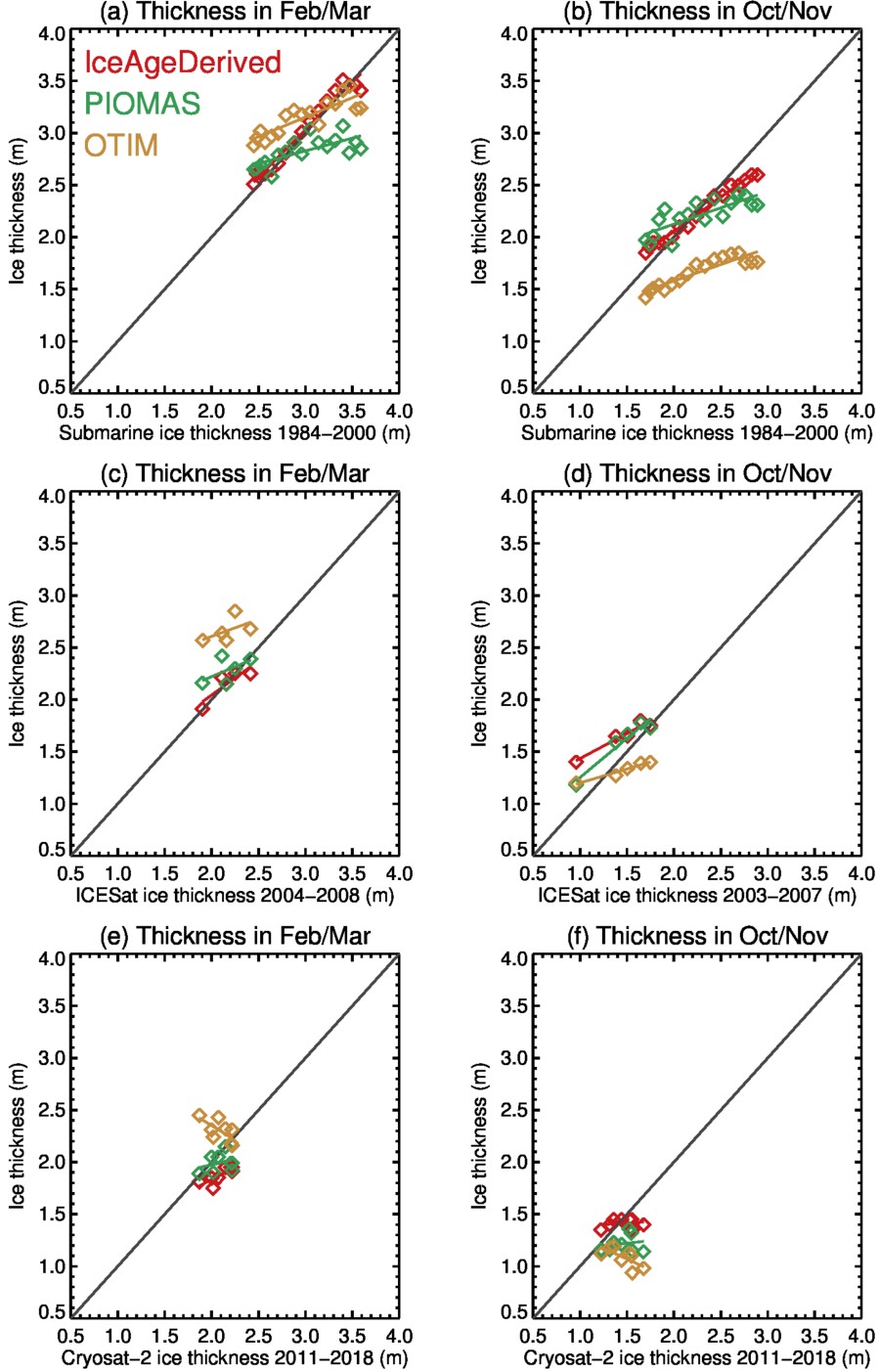


**Figure 5**: Comparison of ice thickness from IceAgeDerived, PIOMAS, and OTIM to ice thickness from submarine up-looking sonar 1984-2000, ICESat 2004-2008, and CryoSat-2 2011-2018 over the SCICEX data release area.

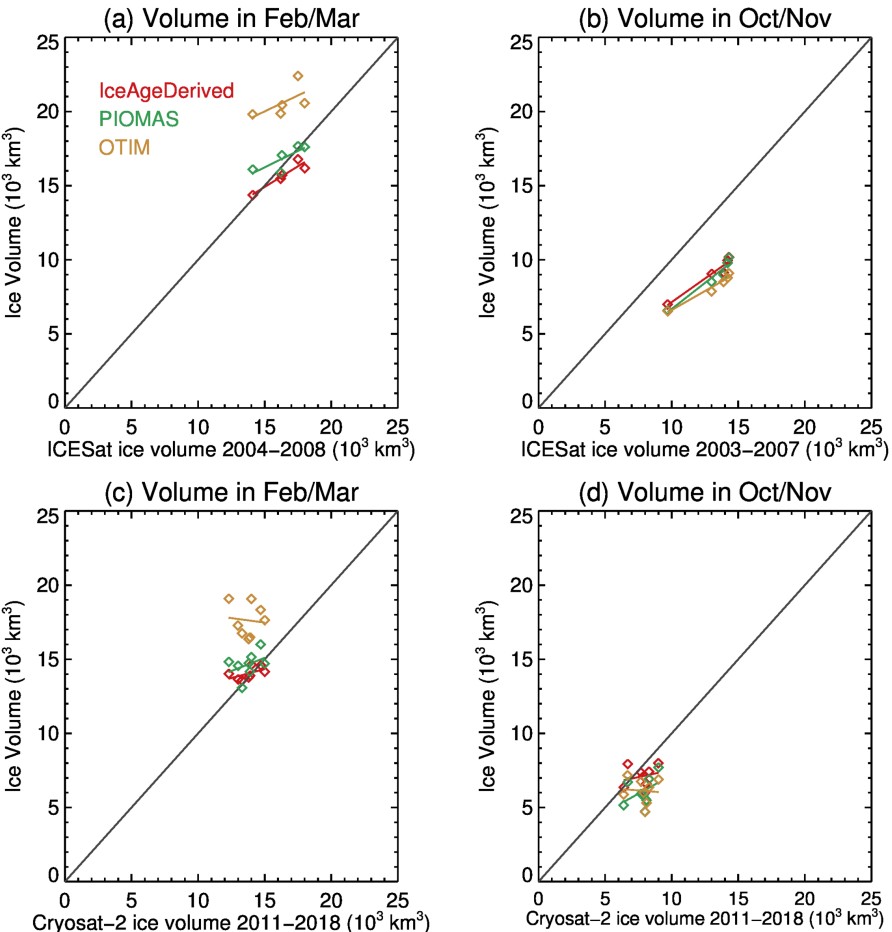

**Figure 6**: Comparison of monthly ice volume from IceAgeDerived, PIOMAS, and OTIM and to ice volume from ICESat 2004-2008, and CryoSat-2 2011-2018 over the Arctic Ocean.

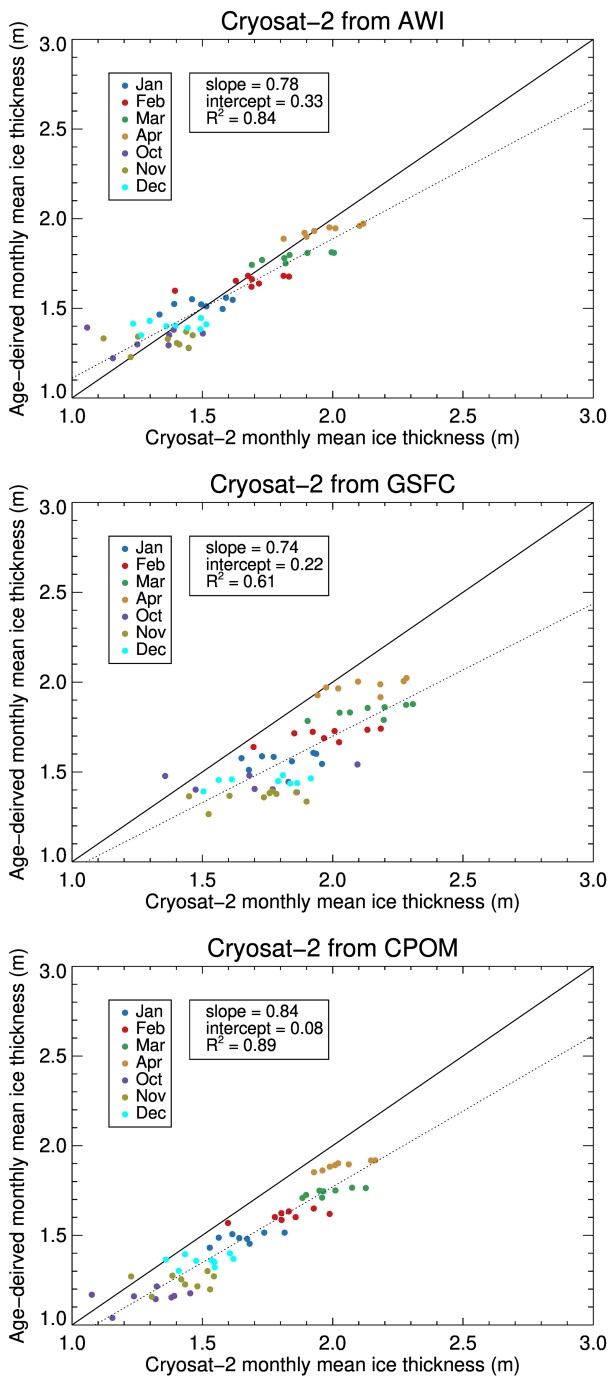

**Figure 7**: Scatterplot of IceAgeDerived and CryoSat-2 monthly mean ice thickness, where the CryoSat-2 data are from (top) AWI, (middle) NASA GSFC, and (bottom) CPOM. The dashed line represents the regression line of IceAgeDerived monthly mean ice thickness on the CryoSat-2 data, with slope, intercept, and $R^2$ indicated.


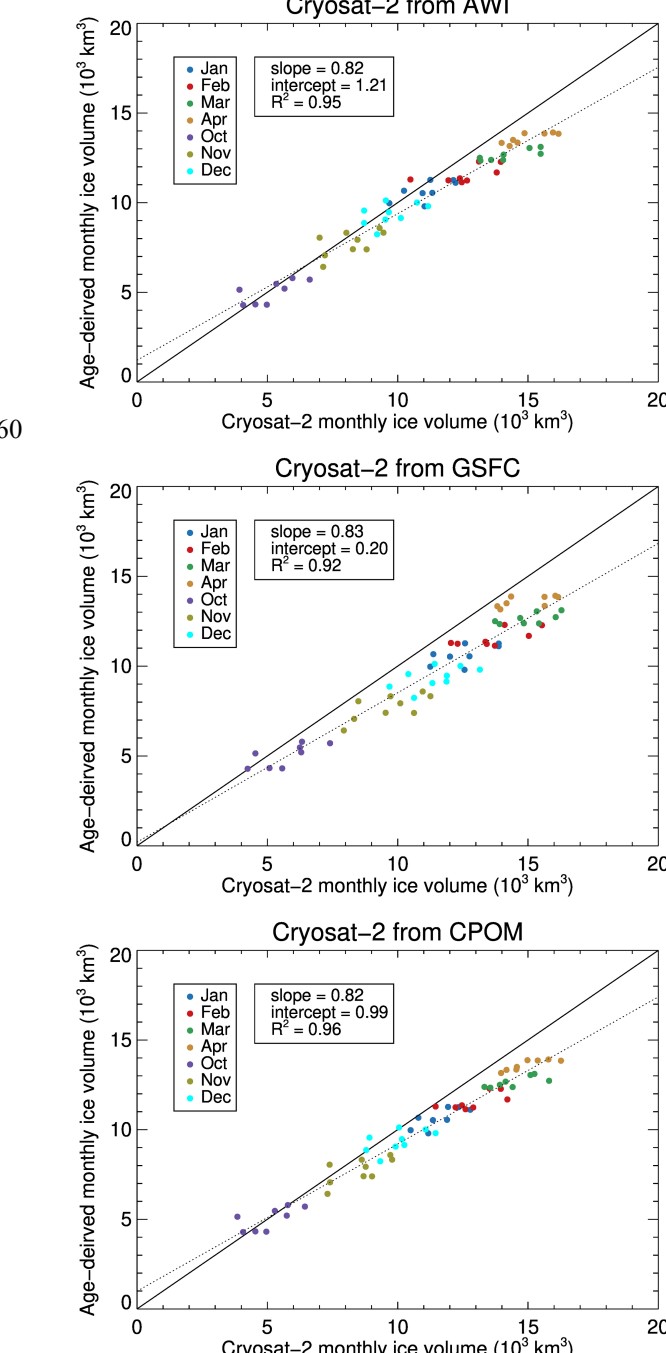


**Figure 8**: Same as Figure 7 but for monthly ice volume.

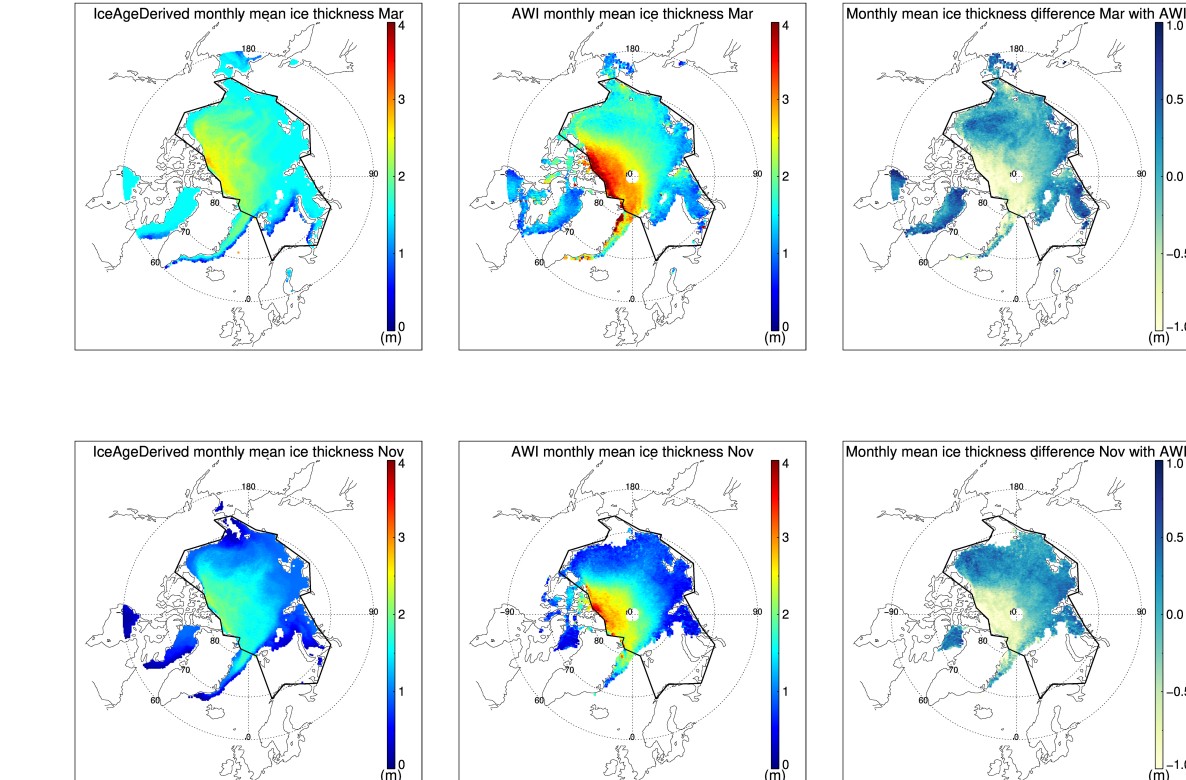


**Figure 9**: Monthly mean ice thickness from IceAgeDerived (top left), from AWI CryoSat-2 (top middle), and their difference in March 2011-2018; and monthly mean ice thickness from IceAgeDerived (bottom left), from AWI CryoSat-2 (bottom middle), and their difference in November 2011-2018.


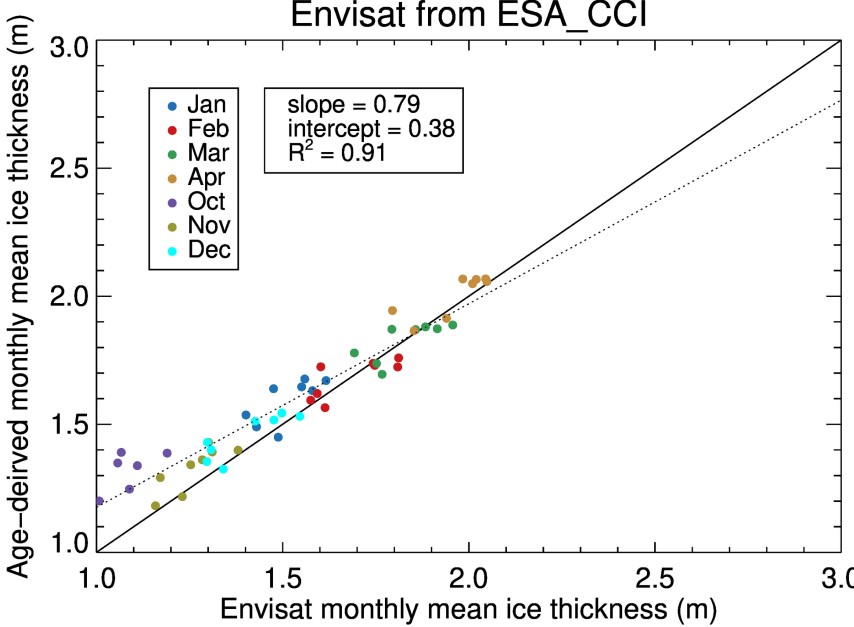

**Figure 10**: Scatterplot of IceAgeDerived and Envisat monthly mean ice thickness, where Envisat data are from ESA CCI. Dashed line represents the regression line of IceAgeDerived monthly mean ice thickness on the Envisat data, with the slope, intercept, and $R^2$ indicated.


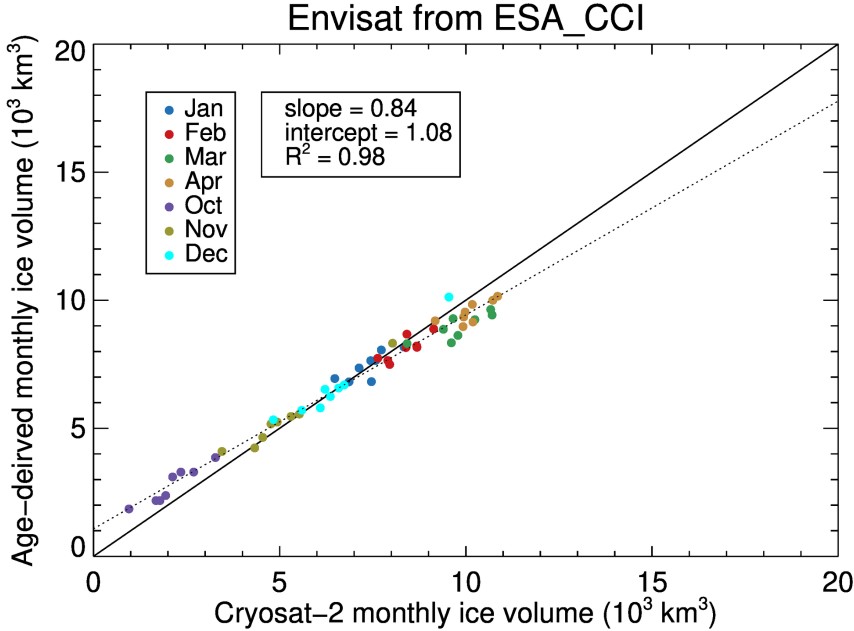

**Figure 11**: Same as Figure 10 except for ice volume.

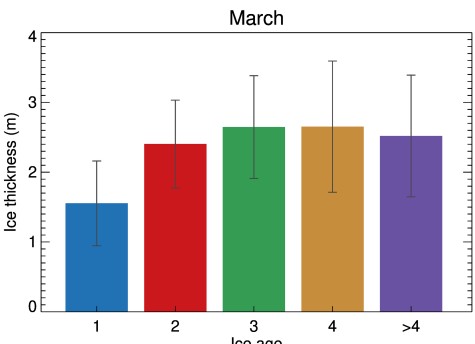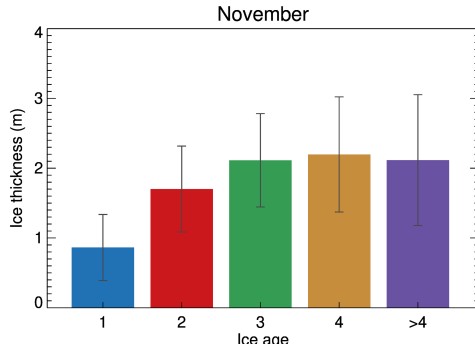


**Figure 12**: Ice age versus ice thickness from collocated ice age and AWI CryoSat-2 ice thickness. The error bar shows one standard deviation above and below the mean ice thickness in each ice age category.

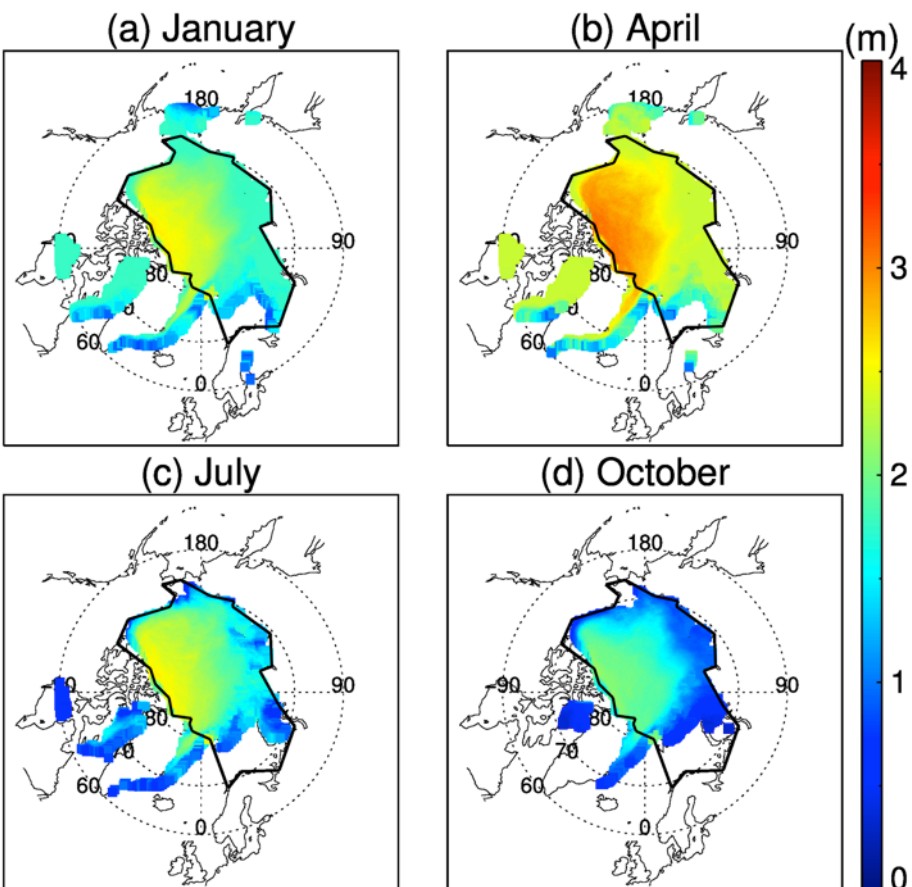

**Figure 13**: Monthly mean sea ice thickness distribution 1984 to 2018 from ice age in the Arctic in January, April, July, and October. The polygon outlines the Arctic Ocean defined in this study, as in Kwok 2018.

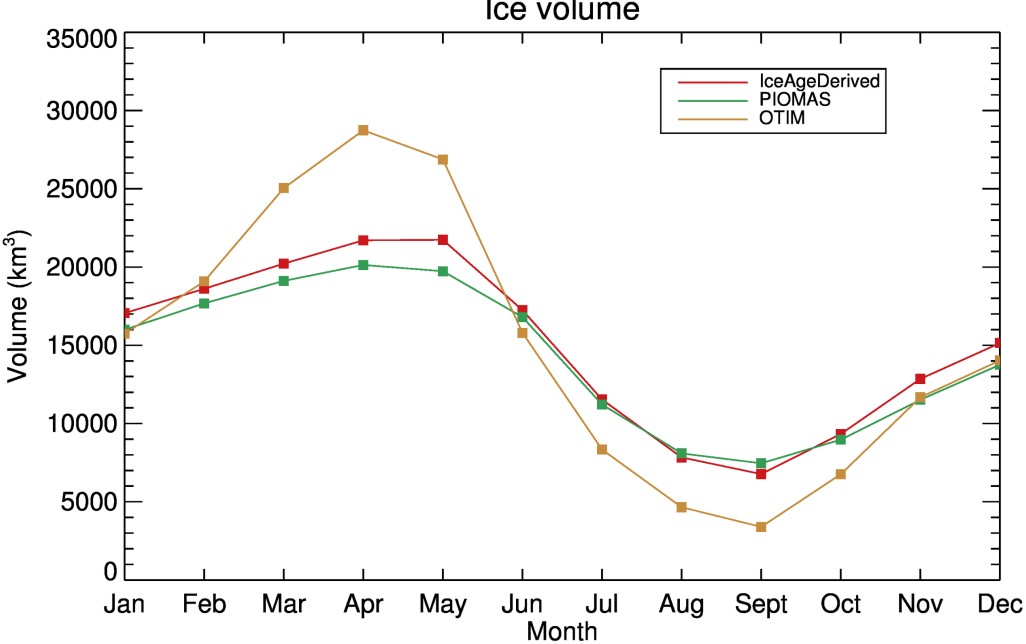


**Figure 14**: Mean annual cycle of ice volume for 1984-2018 over the Arctic Ocean.

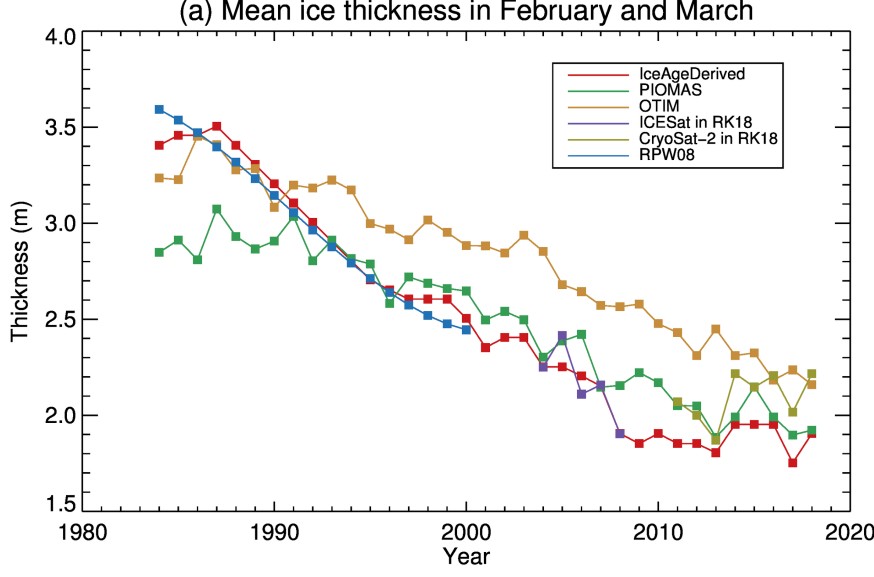

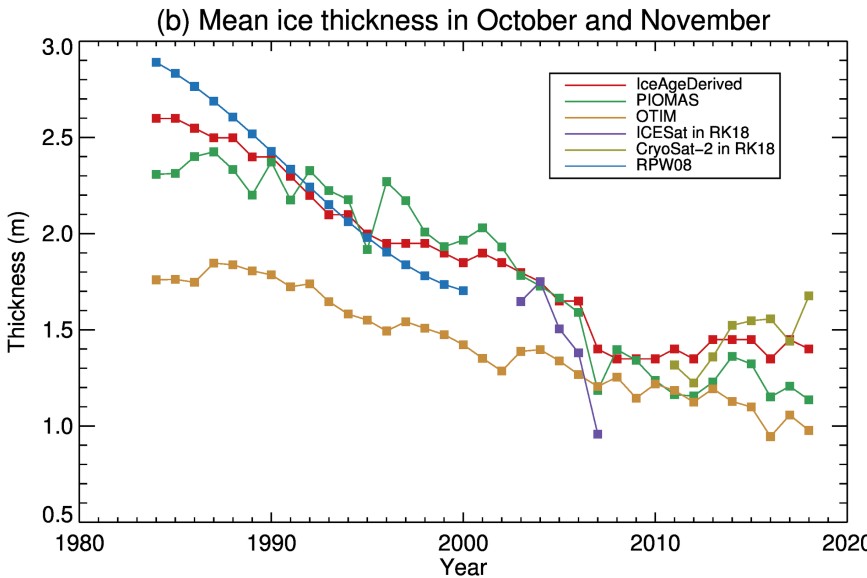


**Figure 15**: Mean ice thickness in February and March (a) and from October and November (b) from 1984 to 2018 derived from ice age, PIOMAS, OTIM, ICESat (2004-2008), CryoSat (2011-2018), and submarine data (1984-2000) over the SCICEX data release area.

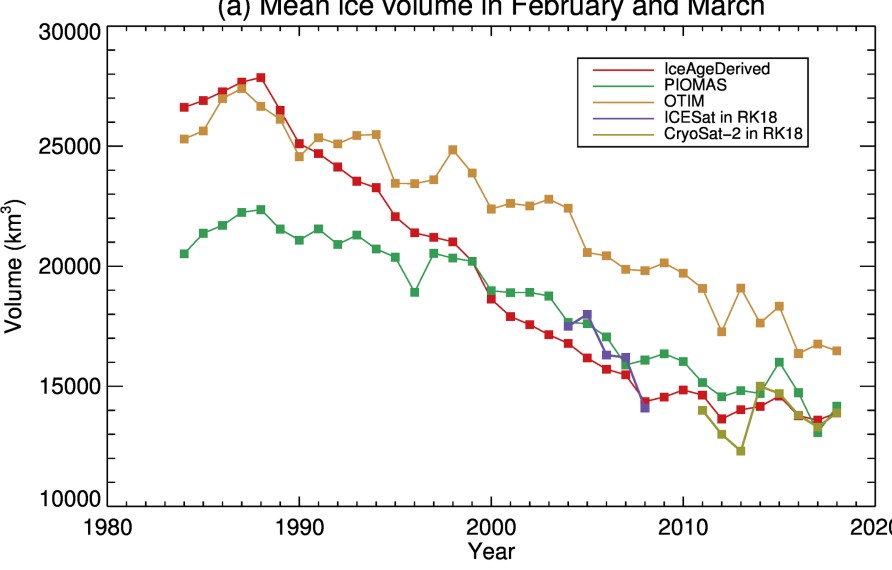


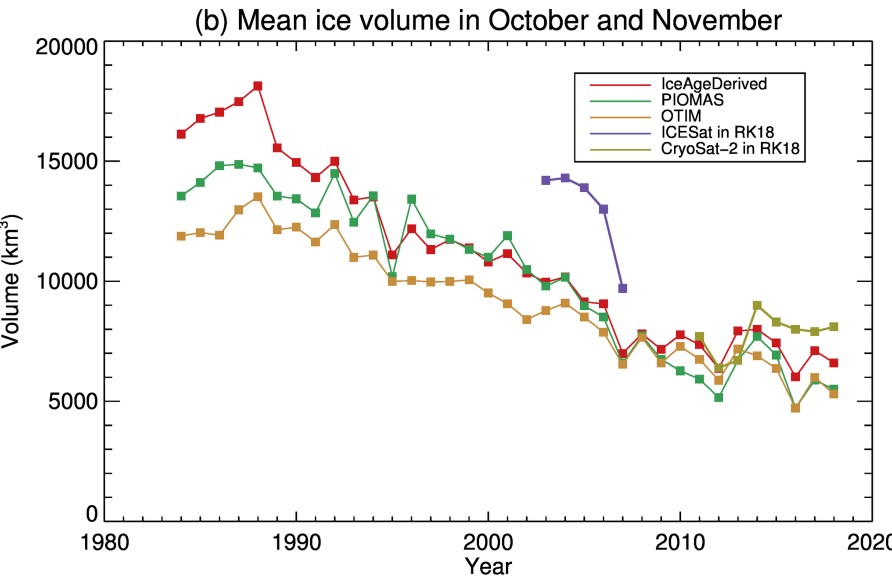

**Figure 16**: Mean Arctic ice volume in February and March (a) and from October and November (b) from 1984 to 2018 derived from ice age, PIOMAS, OTIM, ICESat (2003-2007), and CryoSat (2011-2018) over the Arctic Ocean.

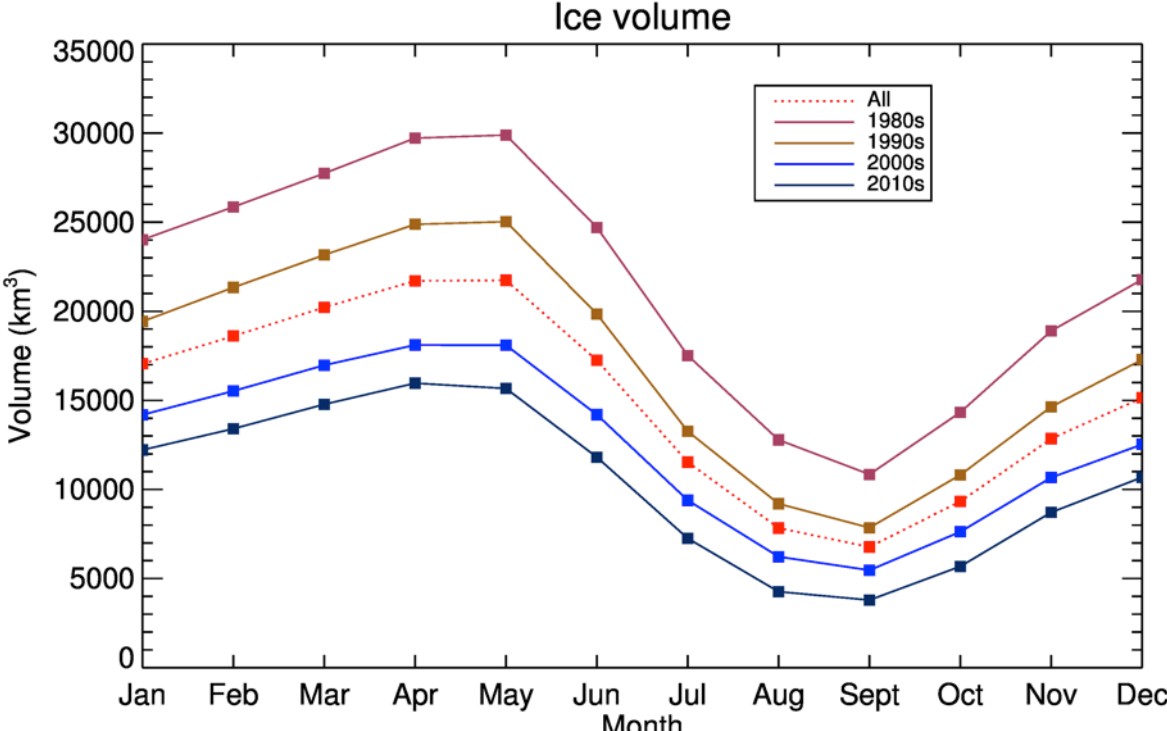

**Figure 17**: IceAgeDerived mean annual cycle of ice volume in 1980s, 1990s, 2000s, 2010s, and in all years over the Arctic Ocean.

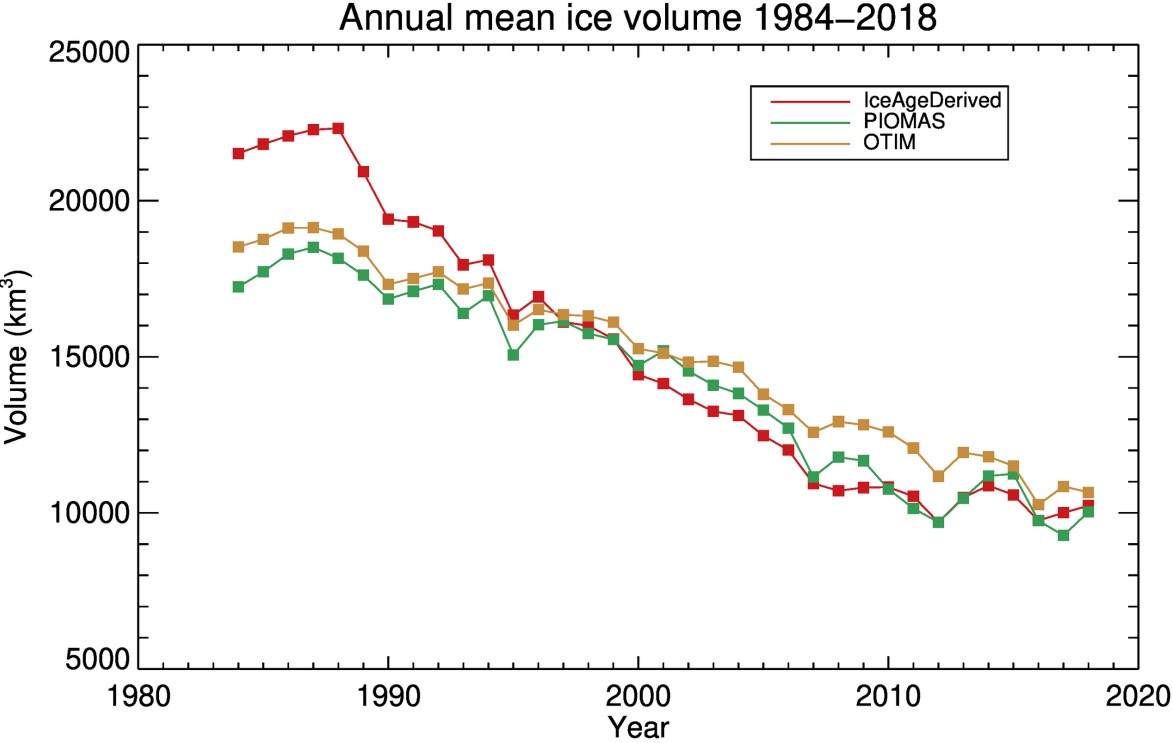

**Figure 18**: Time series of annual mean Arctic ice volume from 1984 to 2018 derived from ice age, PIOMAS, and OTIM over the Arctic Ocean.

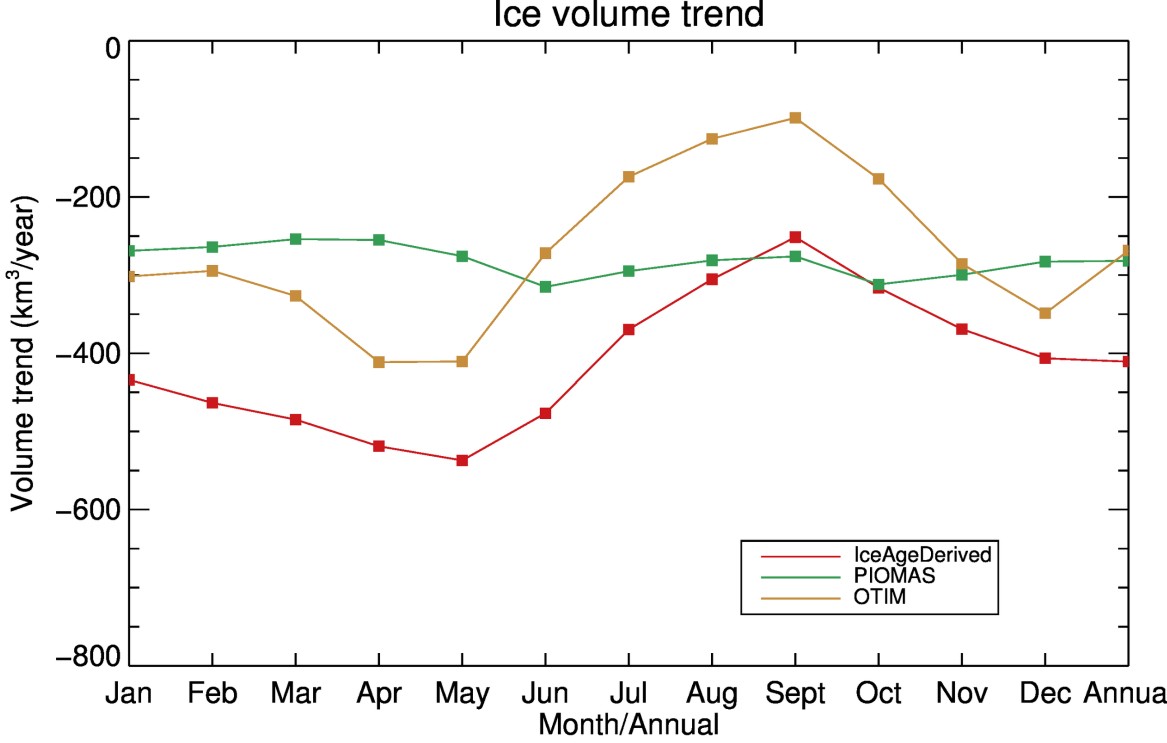

**Figure 19**: Trend of Arctic ice volume in each month and in the annual mean from 1984 to 2018 derived from ice age, PIOMAS, and OTIM
over the Arctic Ocean.

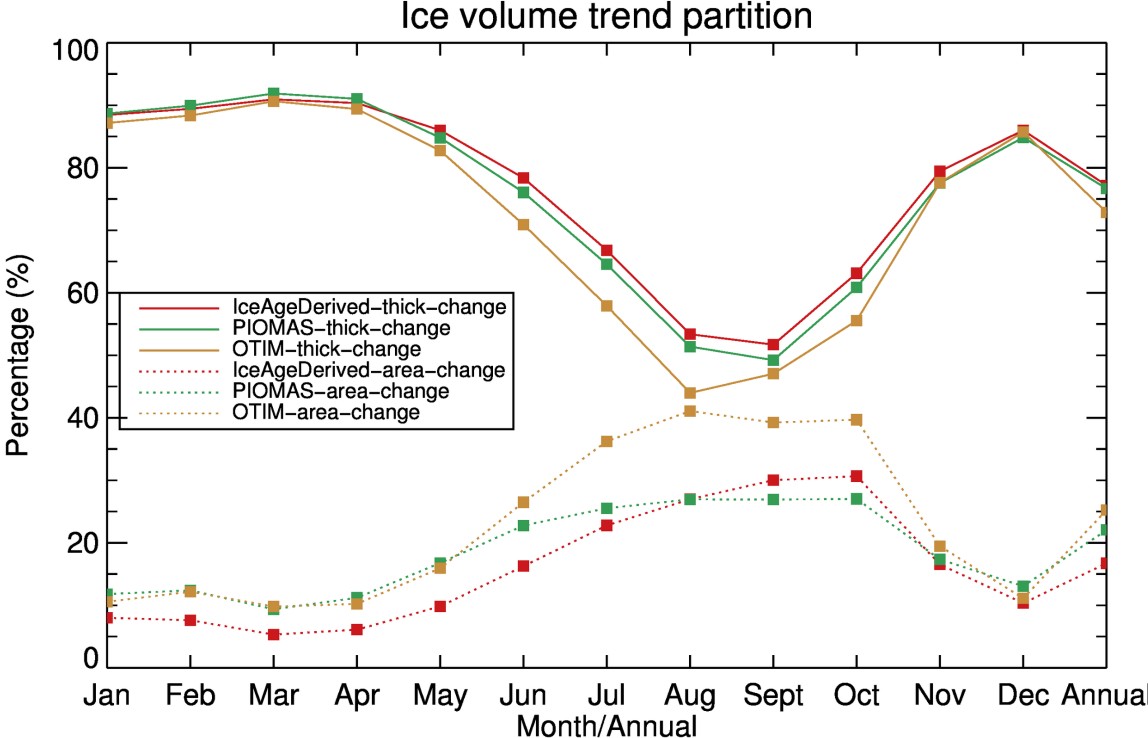

**Figure 20**: Partition of trend of Arctic Ocean ice volume in each month and in the annual mean from 1984 to 2018 to changes in ice area and changes in ice thickness derived from ice age, PIOMAS, and OTIM.


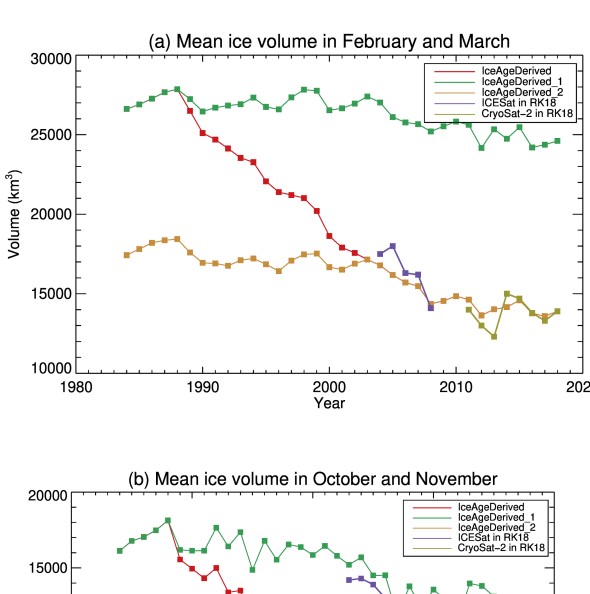

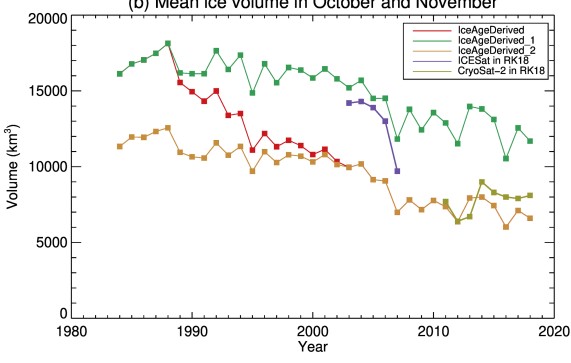

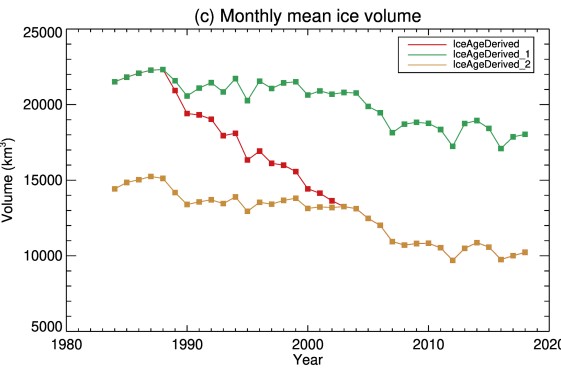

**Figure 21**: Mean ice thickness over the Arctic Ocean from 1984 to 2018 derived from ice age using a varying age-thickness relationship
(IceAgeDerived), using the age-thickness relationship in 1984 (IceAgeDerived_1) and using the age-thickness relationship in 2004-2008
(ICESat) (IceAgeDerived_2) in (a) February and March, (b) October and November, and (c) monthly mean of all months.

**Appendix**


**Table A1**. Monthly mean values of the correction term $\mathfrak{f}(\tau)$ in Eq.1, from Table 4 in Rothrock et al. (2008).

| Month | $\mathfrak{f}(\tau)$,m |
|---|---|
| January | 0.087 |
| February | 0.098 |
| March | 0.110 |
| April | 0.118 |
| May | 0.122 |
| June | 0.113 |
| July | 0.026 |
| August | 0.004 |
| September | 0.025 |
| October | 0.054 |
| November | 0.070 |
| December | 0.081 |

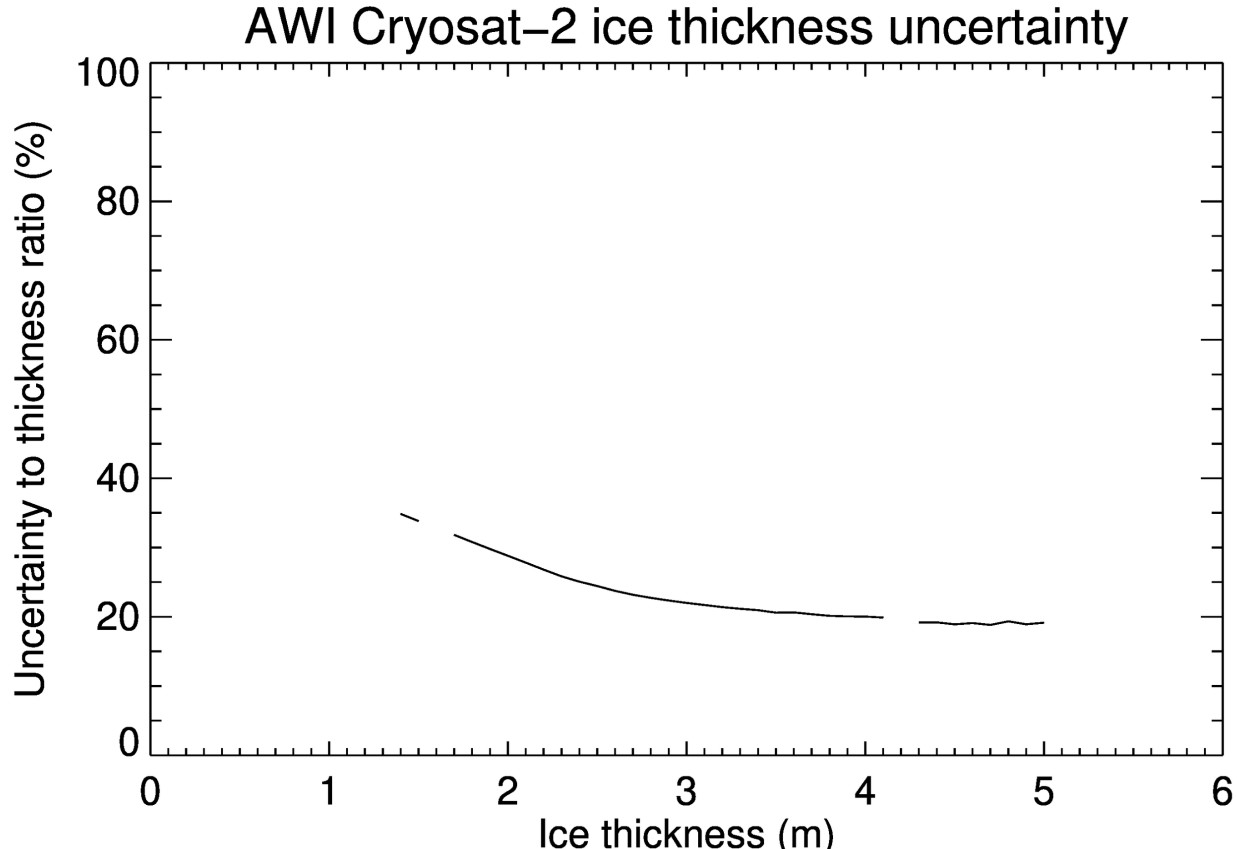

Figure A1: The percentage of uncertainty to sea ice thickness in AWI CryoSat-2 monthly mean ice thickness 2011-2018.

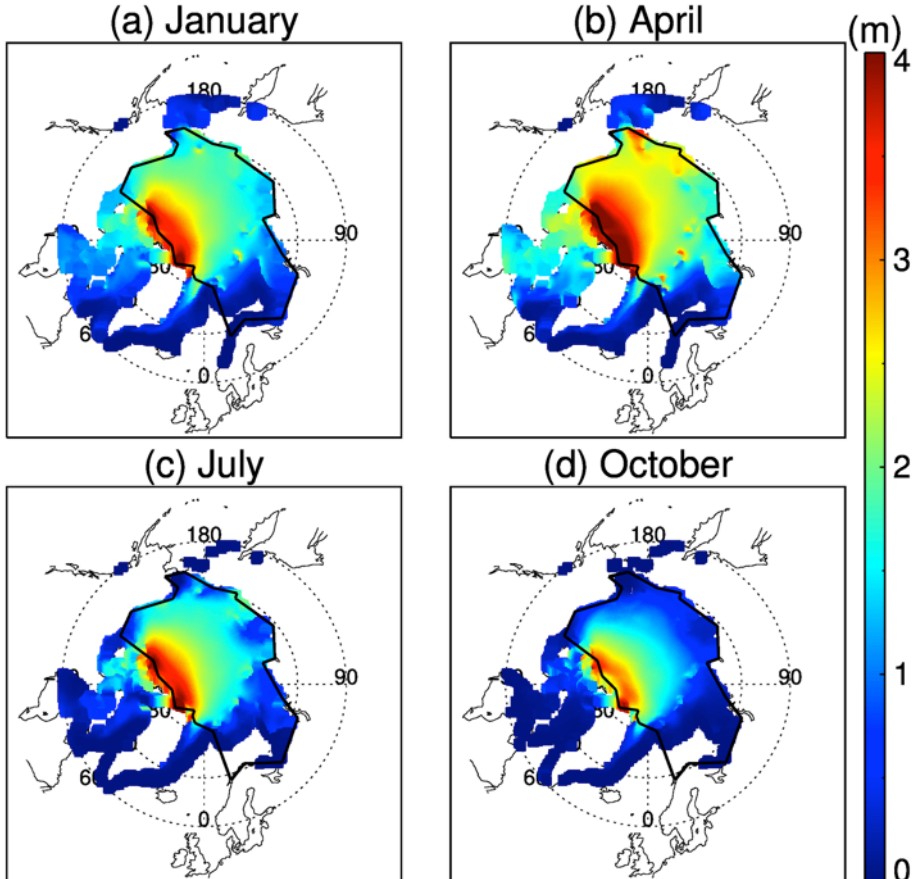

**Figure A2**: Derived climatological mean sea ice thickness distribution in the Arctic from PIOMAS, 1984 to 2018.

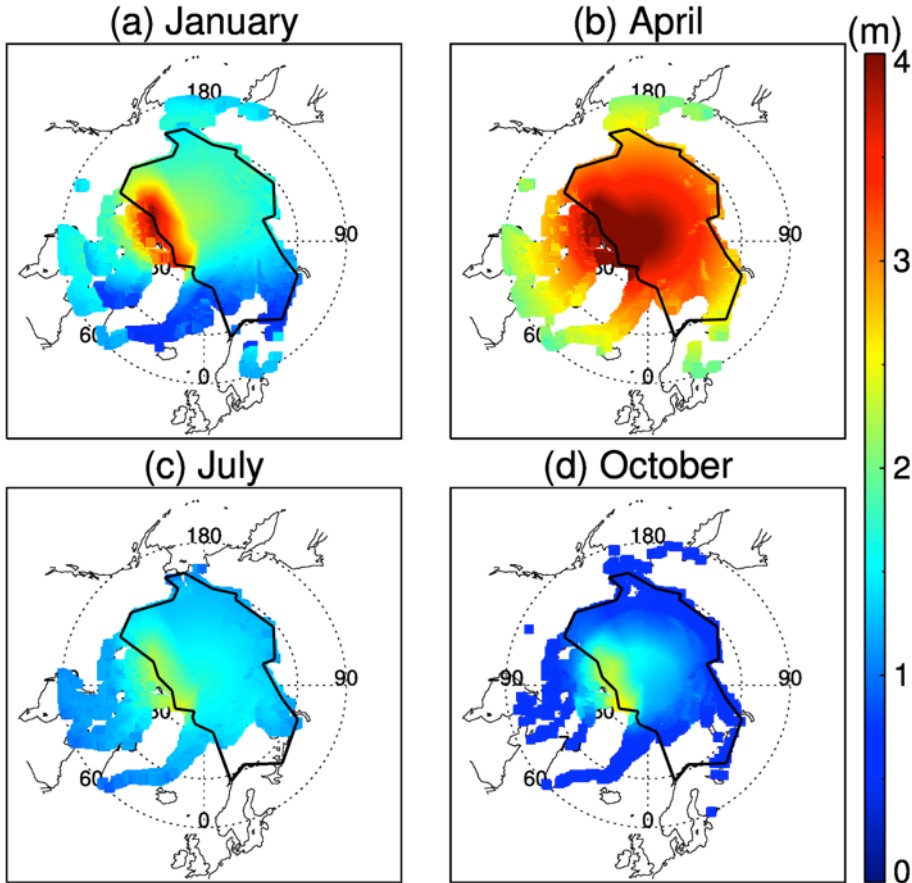

**Figure A3**: Derived climatological mean sea ice thickness distribution in the Arctic from OTIM, 1984 to 2018.
