# Peer review of "Multidecadal Arctic sea ice thickness and volume derived from ice age"

_The Cryosphere, 2019_

## Referee Comment (RC1) · Jack Landy (Referee) · 22 Sep 2019

The comment was uploaded in the form of a supplement:
https://www.the-cryosphere-discuss.net/tc-2019-192/tc-2019-192-RC1-supplement.pdf

———————————————————

---

## Referee Comment (RC2) · Anonymous Referee #2 · 7 Nov 2019

**Review for The Cryosphere Discussions,** https://www.the-cryosphere-discuss.net/tc-2019-192

**Multidecadal Arctic sea ice thickness and volume derived from ice age**
**Liu et al., 2019**

The study of Liu et al. (2019) introduces an Arctic-wide sea ice thickness and volume data product and retrieval method derived from sea ice age. Their product extends all the way back to the early 1980s and presents a data set created with a consistent method, thus providing an interesting novel addition to the existing sea ice thickness products. In addition to complementary information to the more recent satellite altimetry based products, the product could bring additional information about the conditions before the more systematic period of satellite altimeter sea ice measurements.

However, the manuscript currently lacks clarity and detail in explanation of some of the implemented methods. In particular the validation of the product should be improved and justified in order to prove the usefulness of the product. In addition there are some minor cases, included in the review comments below, which should be corrected to accomplish a more finished manuscript.

Considering the novelty and added value in extending the satellite based sea ice thickness records, I recommend this manuscript to be considered for publication in the Cryosphere, after addressing the major review comments.

**General comments:**

1. The data and methods section lacks clarity. There is a great number of data sets used for creating the product and then those used for validation/comparison. And some of the products are used for both purposes. And not really in a chronological order. It would improve the readability if you could structure this section so that it is clear for which purpose the data sets are used, maybe adding separate sections for datasets used in IceAgeDerived creation and for validation data.

2. You seem to use ICESat data as one set of validation data, which is always a bit suspicious if you are using it to construct your data set. The same applies to the draft data. You could either remove the comparisons to these completely from the results or really emphasise and justify more, what comparisonal value these bring.

3. Uncertainties are sometimes painful, but they could be handled more systematically. You mention some, but there is very little analysis. In the data section there are some uncertainty estimates for OTIM, but not really for the other data sets. In results there are brief mentions of ICESat and CryoSat-2 uncertainties. And you mention significance levels for ice thickness and volume trends. Adding more discussion and quantifying the uncertainties in a comparable manner, as well as stating seasonal differences in uncertainties, perhaps adding some discussion on the possible biases

from using submarine vs. laser altimeter in the ice age derived thickness, would add a nice touch to the manuscript.

4. The results, particularly the comparisons with different data sets, should be discussed in detail. Currently the statement about the usefulness of IceAgeDerived is not made that clear. PIOMAS and OTIM seem to be used here as the main comparison sets, and they are good in a sense that both extend to the early 1980s, but to reason the usefulness of IceAgeDerived, you could consider using a satellite altimetry observation based thickness data set with a good temporal extent. I would see some of the main users for the IceAgeDerived being those who are already using altimetry data for sea ice thickness and volume, and thus it would be good to see how these two compare over a longer time period. There are for example datasets combining EnviSat and CryoSat-2, where efforts have been made to bring these to a level of consistency. Such data sets are provided at least by CTOH/LEGOS and ESA CCI. This is only a suggestion for the comparison data, but in case you decide to stay with PIOMAS and OTIM, it would be good to add explanation why you chose these, what are they good for and what do the comparisons really tell about the usefulness of IceAgeDerived.

**Minor comments/edits:**

L23: Have declines -> have declined

L27-29: There could be more sources, perhaps making a stronger statement with results based on satellite observations, if possible. And there could be something newer for the model results, as 2002 was almost two decades ago.

L34: The "relatively high quality of sea ice concentration retrievals from passive microwave data", relative to what?

L39-40: Not mentioning EnviSat? It covers almost a decade of historical data. Of course an exhaustive list might be unnecessary here, but you could consider adding "e.g." if only mentioning CryoSat-2 from the radar altimeters as now it sounds like CryoSat-2 is the only source.

L52: Maybe a newer source than Wang et al. 2010? In Section 4, Discussion and Conclusions, you mention the new snow products and their remaining uncertainties, so perhaps something from there.

L55: Does Laxon belong here? And rather many references for PIOMAS?

L58-68: Nice paragraph!

L59: Individual sea ice *parcels*?

L79: Masnalik et al. 2007 -> Maslanik et al. 2007

L81-83: Each grid cell is tracked as independent parcel, but age of a grid cell of parcels with different ages is assigned to this parcel? The latter sentence in these lines could be more clear.

L117-123: Confusing section, it is a bit unclear what you mean with the "interannual change with the annual cycle superimposed in averaged ice thickness". Also, I and A are not explained too well. These equations should be explained better as you mention they will be used in the results section.

L125-126: You reduce 0.29 m from ice thickness of IceAgeDerived when comparing to submarine derived ice thickness e.g. for the statistics in Tables 1 and 2, or which way?

L127-130: Figures 2 and 3 in RK18 are thickness (Figure 2) and volume (Figure 3), so it would be appropriate to refer to "sea ice thickness and volume" in that order.

L132: Key, et al., -> Key et al.,

L160-161: Is the 10 km necessary to mention here?

L167-168: You use age classes only up 4+ years, but Tschudi et al. 2016 (Fig. 5) have up to 5+. How did you choose this? Using the same classes would increase the consistency and comparability.

L168-170: This method needs more reasoning.

L200-203: Did I understand correctly that you go from weekly to daily to monthly. What is the benefit of doing the daily step?

L244-246: Good that you mention this! How about ICESat? That too was used in the development, right?

L269: How is the partial recovery after summer 2008 visible in these DRA mean ice thicknesses? Particularly in IceAgeDerived?

L295: Arctic sea ice volume for what? Is this still for IceAgeDerived? Maybe add more explanation in the figure caption.

L315-328: Interesting analysis! See comment about Fig. 14.

L348-350: This bullet point does not seem as important as the others, as these findings have been shown in other studies. This could be more of a point to state the consistency between methods, IceAgeDerived succeeds in showing this phenomena that the other sea ice

thickness products have captured, which would encourage the users to take on IceAgeDerived.

L360-363: Extremely interesting! I missed the information for which area this was done.

L366-367: Would love to see more analysis on this. Tschudi et al. (2016) seemed to have thicknesses increasing for each age category up to 5 years. It would be a nice addition to see some speculation about the causes.

L410-411: These references are not used (and maybe never will be)

L475: Malanik -> Maslanik

L484, L487: Please add a and b for Tschudi 2019.

Table 1, and others: SCIEX -> SCICEX

Table 1, Table 2, Fig. 5, Fig. 6, and where relevant: Cryosat-2 -> CryoSat-2

Fig. 1 Consider a different latitudinal cut off, now there is quite a lot of uninformative area in the figures and especially it is hard to see the draft observations. Or if wishing to keep similar cut off to your other figures, consider emphasizing the draft points.

Fig. 2 to1995 -> to 1995 (space). For consistency, consider having the same colorbar as in the other figures, e.g. Fig. 7 [0,4] instead of [0,~4.5]. Consider as well the choice of colormap if 3 m ice, which now stands out with yellow, does not need extra attention. Also, the unit is missing for sea ice thickness.

Fig. 4a v4.0? This maybe refers to the sea ice age product available at NSIDC, but I did not see the version mentioned elsewhere in the manuscript.

Fig. 9 GORE box? And in general, there are a bit too many names for different areas (Arctic Ocean (as in RK18) , SCICEX box, GORE box, DRA). Use only one, unambiguous name for each area.

Fig. 14 I did not see this figure being referred to. If this is correct, please add it somewhere in L315-328.

Fig. A1 t0 -> to

---

## Author Comment (AC1) · 2 Jan 2020

The comments and response are too long to be pasted here. I have them in the supplement.
* * *

---

## Author Comment (AC2) · 2 Jan 2020

Reviewer's comment is in italic, response is in normal font.

*__Review for The Cryosphere Discussions,__ [https://www.the-cryosphere-discuss.net/tc-2019-192](https://www.the-cryosphere-discuss.net/tc-2019-192) __Multidecadal Arctic sea ice thickness and volume derived from ice age__*

*__Liu et al., 2019__*

*The study of Liu et al. (2019) introduces an Arctic-wide sea ice thickness and volume data product and retrieval method derived from sea ice age. Their product extends all the way back to the early 1980s and presents a data set created with a consistent method, thus providing an interesting novel addition to the existing sea ice thickness products. In addition to complementary information to the more recent satellite altimetry based products, the product could bring additional information about the conditions before the more systematic period of satellite altimeter sea ice measurements.*

*However, the manuscript currently lacks clarity and detail in explanation of some of the implemented methods. In particular the validation of the product should be improved and justified in order to prove the usefulness of the product. In addition there are some minor cases, included in the review comments below, which should be corrected to accomplish a more finished manuscript.*

*Considering the novelty and added value in extending the satellite based sea ice thickness records, I recommend this manuscript to be considered for publication in the Cryosphere, after addressing the major review comments.*

We appreciate the reviewer's critical evaluation and constructive suggestions. All of the reviewer's comments have been addressed. All the responses are included in the revised manuscript. We believe revisions responding to reviewer's comments make the manuscript better.

*__General comments:__*

*1. The data and methods section lacks clarity. There is a great number of data sets used for creating the product and then those used for validation/comparison. And some of the products are used for both purposes. And not really in a chronological order. It would improve the readability if you could structure this section so that it is clear for which purpose the data sets are used, maybe adding separate sections for datasets used in IceAgeDerived creation and for validation data.*

The data and method section is updated and re-structured. Detailed information of all the data sets used in this study is added, with information on which data set is used for algorithm development and which data set is used for evaluation/validation, and which is used for both purposes. Subsections are added as the reviewer suggested.

*2. You seem to use ICESat data as one set of validation data, which is always a bit suspicious if you are using it to construct your data set. The same applies to the draft data. You could either remove the comparisons to these completely from the*

*results or really emphasise and justify more, what comparisonal value these bring.*

Another reviewer raised the same questions. We added text to highlight the limitations of such comparisons the reviewer referred to. In the revised manuscript, besides the comparison to ICESat data and draft data, we added independent validation data sets, Cryosat-2 products from NASA GSFC, AWI, and CPOM. Comparisons of IceAgeDerived ice thickness and ice volume with those from Cryosat-2 are included in the revised manuscript.

A comprehensive assessment of the IceAgeDerived ice thickness and ice volume against Cryosat-2 has been carried out. The IceAgeDerived ice thickness and volume are compared to monthly mean Cryosat-2 ice thickness from AWI, NASA GSFC, and CPOM 2011-2018. The following figures, Figure 1 and 2, show the scattering plots of the comparisons, with statistics shown in Table 1. The monthly mean ice thickness shown in the figures is the mean of ice thickness of all pixels in the Arctic.

It shows the IceAgeDerived has slightly smaller monthly ice thickness and volume compared to AWI Cryosat-2 products from January to April, and from October to December, with overall means (standard deviations) of -0.02 m (0.11) and -0.76 $10^3$ km$^3$ (0.86). Comparison to NASA GSFC Cryosat-2 products shows the largest negative bias in those months, with overall means (standard deviations) of -0.27 m (0.15) and -1.79 $10^3$ km$^3$ (0.95) for ice thickness and ice volume respectively. The negative biases to CPOM Cryosat-2 products are in between. Please note, both AWI and CPOM have holes surrounding North Pole not filled, while NASA GSFC fills those holes. We only compared where both products have valid values. Also, you can see the spread between the different Cryosat-2 products.

[Figure]

[Figure]

Figure 1: Scattering plot of IceAgeDerived monthly mean ice thickness and Cryosat-2 monthly mean ice thickness from AWI, NASA GSFC, and CPOM.

[Figure]

[Figure]

Figure 2: Scattering plot of IceAgeDerived monthly ice volume and Cryosat-2 monthly ice volume from AWI, NASA GSFC, and CPOM.

Table 1: Comparison of monthly ice thickness and ice volume between IceAgeDerived and Cryosat-2.

| | | AWI | NASA GSFC | CPOM |
|---|---|---|---|---|
| Comparison of monthly ice thickness of IceAgeDerived and Cryosat-2, 2011-2018 mean (standard deviation) in m | Mean | -0.02 (0.11) | -0.27 (0.15) | -0.18 (0.09) |
| | January | 0.02 (0.09) | -0.24 (0.12) | -0.17 (0.08) |
| | February | -0.03 (0.11) | -0.27 (0.13) | -0.21 (0.10) |
| | March | -0.06 (0.09) | -0.30 (0.11) | -0.24 (0.07) |
| | April | -0.03 (0.08) | -0.14 (0.11) | -0.14 (0.06) |
| | October | 0.00 (0.16) | -0.27 (0.22) | -0.14 (0.12) |
| | November | -0.03 (0.12) | -0.35 (0.14) | -0.19 (0.11) |
| | December | 0.01 (0.10) | -0.29 (0.14) | -0.18 (0.09) |

| Comparison of monthly ice thickness of IceAgeDerived and Cryosat-2, 2011-2018 mean (standard deviation) in $10^3$ km$^3$ | Mean | -0.76 (0.86) | -1.79 (0.95) | -0.98 (0.81) |
| | January | -0.46 (0.64) | -1.89 (0.80) | -0.95 (0.51) |
| | February | -1.03 (0.87) | -2.12 (0.94) | -1.35 (0.68) |
| | March | -1.61 (0.74) | -2.39 (0.76) | -1.79 (0.68) |
| | April | -1.38 (0.59) | -1.37 (0.83) | -1.35 (0.55) |
| | October | -0.11 (0.66) | -0.68 (0.73) | -0.05 (0.66) |
| | November | -0.46 (0.76) | -1.94 (0.87) | -0.80 (0.71) |
| | December | -0.35 (0.75) | -1.79 (0.95) | -0.98 (0.81) |

*3. Uncertainties are sometimes painful, but they could be handled more systematically. You mention some, but there is very little analysis. In the data section there are some uncertainty estimates for OTIM, but not really for the other data sets. In results there are brief mentions of ICESat and CryoSat-2 uncertainties. And you mention significance levels for ice thickness and volume trends. Adding more discussion and quantifying the uncertainties in a comparable manner, as well as stating seasonal differences in uncertainties, perhaps adding some discussion on the possible biases from using submarine vs. laser altimeter in the ice age derived thickness, would add a nice touch to the manuscript.*

The uncertainties of the ICESAt and Cryosat-2 are added in the revised manuscript through literature review. More quantitative analysis of the uncertainties are also included as detailed in the response to comment #2.

"Each of these ICESat and CryoSat-2 ice thickness products has its uncdertainty. The major contributors of these uncdrtainteis are uncertainties in snow depth and snow density, and overall uncertainty in ice thickness is estimated around 0.7 m for ICESat (Kwok and Cunningham 2008). Kwok and Rothrock (2009) estimated the ICESat ice thickness uncertainty around 0.37 m. Comparions with in situ ice thickness observations show unbiased icd thickness estimation in CPOM CryoSat-2 ice thickness, with uncertainties from 34 cm to 66 cm, and error analysis shows the uncertianteis in Arctic-wide sea ice volume are typically about 13.5% (Tilling et al. 2017). Comparsion of NASA GSFC CryoSat-2 ice freeboard to IceBridge data shows a rms difference range from 7.4 to 11.1cm in ice freeboard retrievals(Kurtz et al. 2014). The percentages of ice thickness uncertainty to the ice thickness from AWI CryoSat-2 monthly mean ice thickness from 2011-2018 range from around 35% at mean thickness at 1.4 m to around 20% at mean thickness at 5 m (Figure A1 in appendix). "

[Figure]

Figure A1: The percentage of uncertainty to sea ice thickness in AWI CryoSat-2 monthly mean ice thickness 2011-2018.

4. *The results, particularly the comparisons with different data sets, should be discussed in detail. Currently the statement about the usefulness of IceAgeDerived is not made that clear. PIOMAS and OTIM seem to be used here as the main comparison sets, and they are good in a sense that both extend to the early 1980s, but to reason the usefulness of IceAgeDerived, you could consider using a satellite altimetry observation based thickness data set with a good temporal extent. I would see some of the main users for the IceAgeDerived being those who are already using altimetry data for sea ice thickness and volume, and thus it would be good to see how these two compare over a longer time period. There are for example datasets combining EnviSat and CryoSat-2, where efforts have been made to bring these to a level of consistency. Such data sets are provided at least by CTOH/LEGOS and ESA CCI. This is only a suggestion for the comparison data, but in case you decide to stay with PIOMAS and OTIM, it would be good to add explanation why you chose these, what are they good for and what do the comparisons really tell about the usefulness of IceAgeDerived.*

We thank the reviewer's good point. Besides the comparison to Cryosat-2 as shown above, we also added similar comparison with EnviSat from 2003 to 2010. The results

are shown below and also included in the revised manuscript. Because the spatial coverage of EnviSat and Cryosat-2 are different, different size of hole size without data near the North Pole, we keep these two comparisons separate.

Similar as the comparison to Cryosat-2 product, assessment of the IceAgeDerived ice thickness and ice volume again EnviSat ice product has also been carried out. The EnviSat ice product is from the European Space Agency's (ESA) Climate Change Initiative (CCI) version 2 product, http://cci.esa.int/content/cci-sea-ice-dataset-release-sea-ice-thickness-v20. The IceAgeDerived ice thickness and volume are compared to monthly mean EnviSat ice thickness from 2003-2010. The following figures, Figure 3 and 4, show the scattering plots of the comparisons, with statistics shown in Table 1. The monthly mean ice thickness shown in the figures is the mean of ice thickness of all pixels in a month. It shows the IceAgeDerived has comparable monthly ice thickness and volume to ESA CCI EnviSat products in all months, with overall means (standard deviations) of 0.07 m (0.10) and  -0.08 $10^3$ km$^3$ (0.57).

[Figure]

Figure 3: Scattering plot of IceAgeDerived monthly mean ice thickness and Envisat monthly mean ice thickness from ESA CCI.

[Figure]

Figure 4: Scattering plot of IceAgeDerived monthly ice volume and EnviSat monthly ice volume from ESA CCI.

**Table 2: Comparison of monthly ice thickness and ice volume between IceAgeDerived and Cryosat-2.**

|  |  | AWI |
|---|---|---|
|  | Mean | 0.07 (0.10) |
| Comparison of monthly ice thickness of IceAgeDerived and EnviSat, 2003-2010 mean (standard deviation) in m | January | 0.08 (0.06) |
|  | February | -0.00 (0.06) |
|  | March | -0.00 (0.06) |
|  | April | 0.04 (0.05) |
|  | October | 0.24 (0.11) |
|  | November | 0.06 (0.05) |
|  | December | 0.05 (0.05) |
| Comparison of monthly ice thickness of IceAgeDerived and EnviSat, 2003-2010 mean (standard deviation) in $10^3$ km$^3$ | Mean | -0.08 (0.57) |
|  | January | 0.05 (0.34) |
|  | February | -0.23 (0.28) |
|  | March | -0.84 (0.44) |
|  | April | 0.67 (0.24) |
|  | October | 0.23 (0.23) |
|  | November | 0.13 (0.31) |
|  | December | -0.09 (0.57) |

***Minor comments/edits:***

*L23: Have declines -> have declined*

Corrected.

*L27-29: There could be more sources, perhaps making a stronger statement with results based on satellite observations, if possible. And there could be something newer for the model results, as 2002 was almost two decades ago.*

Added a new reference.

*L34: The "relatively high quality of sea ice concentration retrievals from passive microwave data", relative to what?*

Deleted "relatively".

*L39-40: Not mentioning EnviSat? It covers almost a decade of historical data. Of course an exhaustive list might be unnecessary here, but you could consider adding "e.g." if only mentioning CryoSat-2 from the radar altimeters as now it sounds like CryoSat-2 is the only source.*

Added Envisat, also added a new reference. Analysis using the Envisat products is also added in the revised manuscript.

*L52: Maybe a newer source than Wang et al. 2010? In Section 4, Discussion and Conclusions, you mention the new snow products and their remaining uncertainties, so perhaps something from there.*

A new references are added.

*L55: Does Laxon belong here? And rather many references for PIOMAS?*

Deleted this reference.

*L58-68: Nice paragraph!*

*L59: Individual sea ice parcels ?*

Added.

*L79: Masnalik et al. 2007 -> Maslanik et al. 2007*

Done.

*L81-83: Each grid cell is tracked as independent parcel, but age of a grid cell of parcels with different ages is assigned to this parcel? The latter sentence in these lines could be more clear.*

Changed to "Sea ice thickness can also be derived from sea ice age. An Arctic sea ice age product covering the period from 1984 to the present has been generated based on

Lagrangian tracking of individual sea ice parcels (Tschudi et al. 2019a). Each parcel is tracked independently, and the oldest age of all possible ice parcels within each grid cell is assigned to the cell."

*L117-123: Confusing section, it is a bit unclear what you mean with the "interannual change with the annual cycle superimposed in averaged ice thickness". Also, I and A are not explained too well. These equations should be explained better as you mention they will be used in the results section.*

Rewrote this part.

*L125-126: You reduce 0.29 m from ice thickness of IceAgeDerived when comparing to submarine derived ice thickness e.g. for the statistics in Tables 1 and 2, or which way?*

Reduce 0.29 m in the submarine observations. This is added in the revised manuscript.

*L127-130: Figures 2 and 3 in RK18 are thickness (Figure 2) and volume (Figure 3), so it would be appropriate to refer to "sea ice thickness and volume" in that order.*

Done.

*L132: Key, et al., -> Key et al.,   L160-161: Is the 10 km necessary to mention here?*

To include the 25 km spatial resolution shows that APPx data spatial resolution is comparable to ice age at 12.5 km polar steoreographic projection, and 25 km resolution of Cryosat-2 data. So, we chose to keep this.

*L167-168: You use age classes only up 4+ years, but Tschudi et al. 2016 (Fig. 5) have up to 5+. How did you choose this? Using the same classes would increase the consistency and comparability.*

Tschudi et al. used 5+, which including 5 year old sea ice. We used >4 in this manuscript. They are the same.

*L168-170: This method needs more reasoning.*

We added "However, such information is not available for other months." We agree with the reviewer that such approach leads to uncertainty in the results. We discuss this in the "discussion" section, and propose future fix.

*L200-203: Did I understand correctly that you go from weekly to daily to monthly. What is the benefit of doing the daily step?*

This makes the monthly mean calculation easier, since the uneven distribution of weeks in a month. We also added such text in the discussion "even though the weekly ice age product is converted to weekly ice thickness and interpolated to daily ice thickness for monthly mean calculation. Such daily product lacks detailed temporal information content of ice thickness, and is not intended for direct comparison to

point in situ ice thickness or other daily ice thickness products."

*L244-246: Good that you mention this! How about ICESat? That too was used in the development, right?*

Added "ICESat" in the text. We added the comparison to Cryosat-2 as independent validation/evaluation.

*L269: How is the partial recovery after summer 2008 visible in these DRA mean ice thicknesses? Particularly in IceAgeDerived?*

Older sea ice is generally thicker, and the ice age information from ice age product is utilized in the derivation of ice thickness. So, the partial recovery of multi year sea ice after summer is reflected in the mean ice thickness from IceAgeDerive product. Such discussion is added in the text, "This agreement can be attributed to that the sea ice age information in the ice age product, including intrinsic features of general decreasing and partial recovery of multiyear sea ice after 2008, are utilized to derive the ice thickness."

*L295: Arctic sea ice volume for what? Is this still for IceAgeDerived? Maybe add more explanation in the figure caption.*

It is still for IceAgeDerived. Revision made in the text and in the figure caption.

*L315-328: Interesting analysis! See comment about Fig. 14.*

*L348-350: This bullet point does not seem as important as the others, as these findings have been shown in other studies. This could be more of a point to state the consistency between methods, IceAgeDerived succeeds in showing this phenomena that the other sea ice thickness products have captured, which would encourage the users to take on IceAgeDerived.*

Agree. This point is to show the consistency between method, and the validility of this IceAgeDerived product.

*L360-363: Extremely interesting! I missed the information for which area this was done.*

It is over the Arctic Ocean. Added this information.

*L366-367: Would love to see more analysis on this. Tschudi et al. (2016) seemed to have thicknesses increasing for each age category up to 5 years. It would be a nice addition to see some speculation about the causes.*

We do not a good answer to this. We do have some speculations, and these are not included in the revised manuscript. We will do further investigation on this subject.

Ice ages differently, progressing through growth during freeze-up and decay during the melt seasons. Ice growth varies depending on initial thickness, as well as the air and

ocean temperatures it is exposed to, and ice dynamics. The older ice gets, the more cycles of variable growth it has passed through. Older ice has been observed to be quite thick, up to 2-3m, in accumulation locations such as the Canadian Archipelago, but has also been observed to be rotten and fairly thin.

Submarines measure the ice freeboard from below with sonar, while space-based sensors such as ICESat are used to estimate thickness based on elevation differences between open water and the ice using snow depth estimates, which introduce the greatest level of uncertainty. It's possible that estimates of snow on the ice and/or localized ice deformation is responsible for the difference in thickness measurements between submarines and spaceborne altimetry in particular years.

*L410-411: These references are not used (and maybe never will be)*

Deleted. Those are from the journal template.

*L475: Malanik -> Maslanik*

Done.

*L484, L487: Please add a and b for Tschudi 2019.*

Done, and revised in the manuscript.

*Table 1, and others: SCIEX -> SCICEX*

Done.

*Table 1, Table 2, Fig. 5, Fig. 6, and where relevant: Cryosat-2 -> CryoSat-2*

Done.

*Fig. 1 Consider a different latitudinal cut off, now there is quite a lot of uninformative area in the figures and especially it is hard to see the draft observations. Or if wishing to keep similar cut off to your other figures, consider emphasizing the draft points.*

Changed the latitude cutoff and also emphasized the draft points.

*Fig. 2 to1995 -> to 1995 (space). For consistency, consider having the same colorbar as in the other figures, e.g. Fig. 7 [0,4] instead of [0,~4.5]. Consider as well the choice of colormap if 3 m ice, which now stands out with yellow, does not need extra attention. Also, the unit is missing for sea ice thickness.*

All suggested changes are made. Figure is replaced.

*Fig. 4a v4.0? This maybe refers to the sea ice age product available at NSIDC, but I did not see the version mentioned elsewhere in the manuscript.*

Version of the ice age product is added in the text.

*Fig. 9 GORE box? And in general, there are a bit too many names for different areas (Arctic Ocean (as in RK18) , SCICEX box, GORE box, DRA). Use only one, unambiguous name for each area.*

Corrected.

*Fig. 14 I did not see this figure being referred to. If this is correct, please add it somewhere in L315-328.*

Added.

*Fig. A1 t0 -> to*

Corrected.

---

## Author Comment (AC3) · 2 Jan 2020

Reviewer's comment is in italic, response is in normal font.

***Review for The Cryosphere Discuss.,*** *https://doi.org/10.5194/tc-2019-192* ***Multidecadal Arctic sea ice thickness and volume derived from ice age***

***Liu et al., 2019***

*This study generates a new product for estimated pan-Arctic sea ice thickness, spanning the full satellite era. Observations of sea ice age are used as a proxy for thickness, with the age-thickness relationship derived incrementally over different periods of the satellite data. A consistent sea ice thickness product covering four decades has considerable novelty, because state-of-the-art ice thickness products from satellite altimeters cover only small chunks of this record, with inter- satellite biases not yet properly reconciled.*

*Despite the attraction of such a new long-term sea ice thickness record, I have some concerns with the method used to derive ice thickness from age, particularly regarding the verification approach. The value of this record as a tool for further studies (e.g. for model assimilation, forecasting, understanding decadal Arctic climate/ocean trends) depends entirely on its success reproducing the well-validated altimetry observations; however, there is no evidence presented for this. The new product is also lacking robust estimates for random and systematic ice thickness uncertainties.*

*I have provided a set of general comments on the methodology and recommendations for improving the analysis. I've also made some minor suggestions to improve the readability of the paper and clarify a few confusing statements. I'd recommend this manuscript is reconsidered for publication in The Cryosphere following these major revisions. Please do get in contact if you have questions regarding these comments. Kind regards, Jack Landy*

We appreciate the reviewer's critical evaluation and constructive suggestions. All of the reviewer's comments have been addressed, and reasons are given why some of the suggested work by the reviewer cannot be fully carried out at this time. All the responses are included in the revised manuscript. We believe revisions responding to reviewer's comments make the manuscript better. For that, we thank the reviewer.

***General comments:***

*1. In my view, the derived ice thickness and volume estimates should be described as 'proxies for ice thickness and volume' throughout the paper, as the method uses ice age observations which are a proxy – but not direct replacement for – sea ice thickness observations.*

        We agree with the reviewer that the derived ice thickness and volume are not direct observations or direct replacement for the observations. Thus, we emphasize this in the introduction, and discussion and conclusion section.

        In the "introduction", we added "These ice thickness and ice volume estimates

are a proxy based on ice age, thus are not intended as a direct replacement for sea ice thickness observations", and "These ice thickness and volume estimates are a proxy from ice age products, thus they are not a direct replacement for sea ice thickness observations" in the "discussion and conclusion".

*2. This study desperately requires a detailed evaluation against available sea ice thickness observations from state-of-the-art altimeters, e.g. ICESat or CryoSat-2. The authors use ICESat data in their calibration of the ice age-thickness relationship, which essentially discounts their assessing the final product against ICESat data (but still do), and only compare to annual mean estimates of ice thickness and volume from CryoSat-2. Several gridded ice thickness datasets are available (from CPOM, AWI, NASA GSFC, LEGOS) which the authors could compare their derived product to. As they haven't used CS2 to calibrate their relationship, this would represent a valid independent assessment. As a suggestion, can the authors calculate the spread of CS2 ice thicknesses within each ice age category of the NSIDC product? I would recommend showing this as a plot. This would provide an estimate for the random uncertainty and potential bias in using age as a proxy for thickness. If one sigma of the PDF of CS2 thicknesses in an age category crosses another, it suggests ice age does not provide a valid proxy for thickness. Can the authors also provide maps of average November and march thickness for the coincident period of CS2 and IAD results?*

In the revised manuscript, we added the following analysis.

Monthly mean Cryosat2 ice thickness from CPOM, AWI, and NASA GSFC from January to April, and from October to December of 2011 to 2018 are used to calculate the spread of CS2 ice thickness within each ice age categories, and to evaluate the IceAgeDerived ice thickness and volume as the reviewer suggested,.

We collocated the NSIDC weekly ice age from 2011 to 2018 with correspondent Cryosat2 monthly ice thickness, and the spreads in March and November are as in the following figure.

[Figure]

Figure 1: Ice age versus ice thickness from collocated ice age and AWI Cryosat2 ice

thickness. The error bar shows the one standard deviation of ice thickness in each ice age category.

Ice thickness increases with ice age for ice age from 1 to 4, and then decreases from ice age 4 to 5. This is consistent with what we found based on upward looking sonar data. This trend shows similarity and difference with what were found in Tschudi et al. (2016), where ice thickness increases from ice age from 1 to 5. Similar as those in Tschudi et al. (2016) (Figure 2), one sigma of the PDF of Cryosat2 thickness in an age category crosses another. We respect the Reviewer's view that ". *If one sigma of the PDF of CS2 thicknesses in an age category crosses another, it suggests ice age does not provide a valid proxy for thickness."*, and we think the estimation of ice thickness from ice age can still be valid, however, the crosses lead to uncertainty in the ice thickness estimation based on ice age.

[Figure]

Figure 2: Monthly mean ice thickness from IceAgeDerived (top left), from AWI Cryosat-2 (top middle), and their difference in March 2011-2018; and monthly mean ice thickness from IceAgeDerived (bottom left), from AWI Cryosat-2 (bottom middle), and their difference in November 2011-2018.

Monthly mean ice thickness from IceAgeDrived and Cryosat-2 (AWI, NASA GSFC, and CPOM) shows similar spatial patterns in March and November. The sea ice thickness has the maximum values north of the Canadian Archipelago, and decreases radially toward the coastal regions of Alaska and Russia. The major differences are over the area north of

the Canadian Archipelago, with the IceAgeDrived underestimating the thickness up to 1 m compared to Cryosat-2.

These analysis are added in the revised manuscript.

*3. Validation approach. A majority of the comparisons made between ice age derived thickness and independent data (Line 222-227) are not truly independent, as the datasets were originally used to calibrate ice age-thickness relationships. If they are statistically dependent, i.e. data X is used to calibrate Y then Y is compared against X, it doesn't tell us much. Some evaluation of annual mean ice thickness/volume are made against truly independent CS2 observations, but this gives no evaluation of the spatial/regional accuracy. I would recommend including either a comprehensive assessment against CS2 (as described above) or to reserve a selection of the submarine/ICESat data only for assessing the final product, rather than calibrating with AND assessing it against the same thing. In its present form, I don't believe the validation has 'proven the soundness of the IAD thickness' as suggested on lines 337-339.*

A comprehensive assessment of the IceAgeDerived ice thickness and ice volume against Cryosat-2 has been carried out. The IceAgeDerived ice thickness and volume are compared to monthly mean Cryosat-2 ice thickness from AWI, NASA GSFC, and CPOM 2011-2018. The following figures, Figure 3 and 4, show the scattering plots of the comparisons, with statistics shown in Table 1. The monthly mean ice thickness shown in the figures is the mean of ice thickness of all pixels in the Arctic.

It shows the IceAgeDerived has slightly smaller monthly ice thickness and volume compared to AWI Cryosat-2 products from January to April, and from October to December, with overall means (standard deviations) of -0.02 m (0.11) and -0.76 $\times 10^3$ km$^3$ (0.86). Comparison to NASA GSFC Cryosat-2 products shows the largest negative bias in those months, with overall means (standard deviations) of -0.27 m (0.15) and -1.79 $10^3$ km$^3$ (0.95) for ice thickness and ice volume respectively. The negative biases to CPOM Cryosat-2 products are in between. Please note, both AWI and CPOM have holes surrounding North Pole not filled, while NASA GSFC fills those holes. We only compared where both products have valid values. Also, you can see the spread between the different Cryosat-2 products.

Though the comparison to the Cryosat-2 ice products show overall agreement in both thickness and volume, further investigation and analysis shows that there are rather apparent differences in the ice thickness retrieval spatial distributions as shown in Figure 2. It appears the IceAgeDerived ice thickness underestimates the ice thickness for the older ice while overestimates the ice thickness for the first year ice with comparison to Cryosat-2. It should be also noted that Cryosat-2 also has relatively high uncertainties for very thin and very thick sea ice. In total, these underestimates and overestimates may balance off in the overall mean ice thickness and ice volume comparisons. These noted differences are surely a research topic for future studies.

[Figure]

Figure 3: Scattering plot of IceAgeDerived monthly mean ice thickness and Cryosat-2 monthly mean ice thickness from AWI, NASA GSFC, and CPOM.

[Figure]

Figure 4: Scattering plot of IceAgeDerived monthly ice volume and Cryosat-2 monthly ice volume from AWI, NASA GSFC, and CPOM.

**Table 1: Differences of monthly ice thickness and ice volume between IceAgeDerived and Cryosat-2.**

| | | AWI | NASA GSFC | CPOM |
|---|---|---|---|---|
| Comparison of monthly ice thickness of IceAgeDerived and Cryosat-2, 2011-2018 mean (standard deviation) in m | Mean | -0.02 (0.11) | -0.27 (0.15) | -0.18 (0.09) |
| | January | 0.02 (0.09) | -0.24 (0.12) | -0.17 (0.08) |
| | February | -0.03 (0.11) | -0.27 (0.13) | -0.21 (0.10) |
| | March | -0.06 (0.09) | -0.30 (0.11) | -0.24 (0.07) |
| | April | -0.03 (0.08) | -0.14 (0.11) | -0.14 (0.06) |
| | October | 0.00 (0.16) | -0.27 (0.22) | -0.14 (0.12) |
| | November | -0.03 (0.12) | -0.35 (0.14) | -0.19 (0.11) |
| | December | 0.01 (0.10) | -0.29 (0.14) | -0.18 (0.09) |
| Comparison of monthly ice thickness of IceAgeDerived and Cryosat-2, 2011-2018 mean (standard deviation) in $10^3$ km$^3$ | Mean | -0.76 (0.86) | -1.79 (0.95) | -0.98 (0.81) |
| | January | -0.46 (0.64) | -1.89 (0.80) | -0.95 (0.51) |
| | February | -1.03 (0.87) | -2.12 (0.94) | -1.35 (0.68) |
| | March | -1.61 (0.74) | -2.39 (0.76) | -1.79 (0.68) |
| | April | -1.38 (0.59) | -1.37 (0.83) | -1.35 (0.55) |
| | October | -0.11 (0.66) | -0.68 (0.73) | -0.05 (0.66) |
| | November | -0.46 (0.76) | -1.94 (0.87) | -0.80 (0.71) |
| | December | -0.35 (0.75) | -1.79 (0.95) | -0.98 (0.81) |

We also carried out similar evaluation/validation with Envisat from 2003-2010, and got similar results. Please refer to response to another reviewer's comments.

All these analysis and discussions are added in the revised manuscript.

*4. Uncertainty. I was surprised to see no estimate of uncertainty for the derived sea ice thickness, particularly as this product is a proxy based on the imperfect relationship between ice age and thickness. The underlying sea ice age data have an uncertainty estimate. There are several empirical equations used in the methodology with derived coefficients that will have uncertainties. Several biases are corrected for and these will also have uncertainties, potentially varying over the annual cycle. A proper comparison with independent observations will additionally produce estimates for random and potential systematic uncertainties. I appreciate the added work required to produce robust uncertainty estimates, and for this proxy product they may be high, but for users to trust the new product they need some idea of its accuracy/precision. I expect the authors to make estimates for both the random uncertainty (errors in coefficients, errors in ice age product, noise in comparison to independent data) and systematic uncertainty (uncertainties in bias corrections, errors in extrapolating beyond your data collection period, potential biases compared to independent data) in a revised*

*version of the manuscript. These sources of uncertainty also need to be estimated for each month of the year separately, as one would expect the error to vary considerable across the seasonal cycle.*

According to Tschudi et al. (2019) and discussion with Dr. Tschudi, there is no explicit uncertainty estimation in the sea ice age data. As shown in Figure 1, there is uncertainty, as the one standard deviation, corresponding to each ice age category, and these estimations are comparable to those shown in Figure 2 in Tschudi et al. (2016). To estimate the random uncertainty of the IceAgeDerived ice volume over the Arctic Ocean we applied the ice thickness uncertainty errors in each ice age category when converting the weekly ice age to ice thickness from 1984 to 2018. The uncertainty in weekly or monthly ice volume over the Arctic Ocean is the sum of the ice volume uncertainty of all grid cells, where the ice volume uncertainty in a cell is the product of the sea ice concentration, the grid cell area, and the ice thickness uncertainty. This provides the upper limit on the random uncertainty in ice volume. The overall uncertainties in ice thickness and ice volume in every month from 1984 to 2018 are derived. The average ratios of ice volume uncertainties to the mean range from 21% to 29% over the period 1984 - 2018.

The systematic uncertainties of the IceAgeDerived ice thickness and ice volume are estimated by comparison to independent ice thickness and ice volume data from Cryosat-2, which is shown in the response to reviewer's major comment #3.

All these analysis and discussion are added in the revised manuscript.

*5. The authors argue the decreasing trends in ice thickness and volume from their new product are consistent with observations of MYI replacement since the mid-2000s (Lines 268-270). However, the trend in their product is imposed by systematically changing the relationship between ice age and thickness throughout the time series (i.e. Fig 2). Comparing negative trends to the ice age product is basically fitting to and comparing against the same dataset. What physical explanation is there for the ice age-thickness relationship to change by such a considerable amount over these 5-yr segments of time? Can you provide citations to support this? Surely if the relationship changes by so much over time, it indicates ice age cannot be used alone as a proxy for thickness. Temporal/spatial sampling biases in the calibration data (especially the submarines) are very likely to have introduced systematic biases in these 5-yr relationships. What do the time series in Figs 9-12 look like if you use a fixed ice age- thickness relationship for the duration of the record? Unreasonable low?*

Figure 5 shows the time series of mean ice volume using varying ice age-thickness relationships (as in the manuscript), using relationships in 1984, and in 2004-2008 (ICESat period) respectively. The overall trends are -411, -136, -156 km$^3$/year from 1984 to 2018 respectively. This indicates in our approach that the replacement of multi-year ice may only accounts for a smaller part of the overall trend (~33% or ~38%, -136/-411 or -156/-411), while the changes in ice age and ice thickness relationship contribute more the

overall change. Since the ice age-thickness relationships change is small between the
ICESat period and Cryosat-2 period (see both Figure 1 here and Figure 2 in Tschudi et al.
(2016)), this ice age-thickness relationship changes may mainly happen between middle
1980s and middle 2000s, that ice thickness decreases in each corresponding ice age
category. Sea ice extent in September has been decreasing, with trend from 1997 to 2014
four times as large as that from 1979 to 1996 (Serreze and Stroeve, 2015). More solar
heating that the ocean absorbs through the open water area is expected to thin the
remaining ice for all ice categories, leading to even less sea ice in the next summer and
more solar heating. This may explain the decreasing ice thickness for corresponding ice
age. However, it appears the accelerated decrease of ice thickness to corresponding ice
age happens before the accelerated decreasing ice extent in September, which needs
further investigation.

[Figure]

[Figure]

[Figure]

Figure 5: Monthly ice volume over the Arctic Ocean from 1984 to 2018 derived from ice age using varying age-thickness relationship (IceAgeDerived), using age-thickness relationship in 1984 (IceAgeDerived_1) and using age-thickness relationship in 2004-2008 (ICESat) (IceAgeDerived_2) in (a) February and March, (b) October and November, and (c) monthly mean of all months.

***Minor comments/edits:***

*Line 18. Affecting what about the volume?*

Added "the sea ice volume trend"

*L23. 'declines'. Check spelling errors throughout.*

*Changed to "declined". Done.*

*L26. What anomalous ice export? Volume or area export? Needs citations to back up.*

Revised and added a reference (Smedstrud et al. 2011)

*L28-29. Unclear argument – why does this mean it is more sensitive?*

The signal is more apparent with a higher change in percentage, I think.

*L37-38. Sentence seems a bit out of place. Is this here just for the citation..?*

No. It is a good place to introduce related data set and potential applications.

*L48-49. Include the point that the sensor signal must first be sensitive to ice thickness, before modelling or statistical parameters can be used to estimate the thickness.*

Added.

*L55. Laxon citation is not relevant to this point.*

Removed.

*L59. Sea ice floes? Grid cells?*

Added "parcels" after "sea ice".

*L60. Can you comment on the uncertainty of this relationship? Was this reported in the Maslanik paper?*

Added "The uncertainty of this relationship appears to increase from new ice to older ice, with values ranging from approximately 0.2 to 1.0 m (Figure 2 in Maslanik et al. 2007, and Figure 2 in Tschudi et al. 2016)."

*L64. Not necessarily more robust, but more comprehensive definitely.*

Changed to "comprehensive".

*L91. Is Cavalieri 1996 the most up to date reference?*

Definitely not. But that is the one NSIDC asks for to refer if that specific data set is used.

*L93. POP model?*

Not sure. That is beyond my knowledge.

*L107. Are the ice draft data from submarines analysed entirely by yourselves or do you use statistics produced by others (NSIDC)? How do you do the processing? How do you account for unknown snow depth/density at the ice surface? What is the uncertainty on these estimates?*

I used the processed data at NSIDC. Added "Assessment shows the ice thickness has a positive bias of 0.29 m, and the standard deviation if 0.25 m (Rothrock and Wensnahan 2007). The ice thickness from submarine data from 1984 to 2000 from NSIDC are used here"

*Eq 1. Please provide explanation for this coefficient and estimate the uncertainty.*

This is a equation that Rothrock et al. derived and showed in their 2008 paper. I noted this in the revised manuscript.

*L114-115. Should f(tau) not depend on the ice type itself, i.e. accounting for different snow accumulation rates between seasonal and old ice?*

Again, such information is available in Rothrock et al. (2008)

*L118-124. This section is very confusing and requires a re-write. What exactly is I? What does 'interannual change with the annual cycle' mean? What are these equations used*

*for? This part seems like method, rather than data.*

I rewrote this section, and emphasized all the equations were derived and details are available in Rothrock et al. (2008).

*L124. Determined how?*

Added "Details about the bias determination is available in Rothrock and Wensnahan (2007)."

*L125-126. You reduce the submarine ice thicknesses by this bias? Or do you reduce both submarine and your final ice-type derived thicknesses by this? Is this bias applicable for the entire seasonal cycle?*

Changed to "In this study, we therefore reduce individual ice thickness observations by 0.29 m in all the original submarine observations."

*L129. Be more specific about the processing chain used to derive the CS2 data. Is this the JPL product from Kwok and Cunningham, 2014?*

To be honest, I tried to figure out what exactly product these values are based by reading Kwok's 2018 paper and his other papers. I could not figure it out. The best I can do is to cite his 2018 paper, stating that the values are from that paper. I can not confirm if that product is from Kwok and Cunningham 2014.

*L143-145. The recent comparison paper by Sallila et al 2019 has shown very different results between OTIM and CS2 products. Is it worth comparing both to your independent ice-type product, when there show so much systematic uncertainty? Which are you use as your 'true' reference?*

The differences are due to the different retrieval approaches for CS2 and OTIM. As mentioned in the paper by *Sallila et al 2019,* CS2 can only estimate ice thicker than ~0.5m, and OTIM can do for ice thickness between 0 ~ 6m. So both need to be calibrated and validated with in-situ direct measurements from such as submarines and stations for further improvements.  So I would say not to take either of them as 'true' reference, just a 'product' reference

*L149. Can you comment on the positive bias that may be introduced to the derived relationship from your calibrations against submarine data being focused in the central Arctic Ocean?*

In the discussion and conclusion section, we have such discussion: "Third, in deriving the relation of ice age to ice thickness in the years before 2000, only ice draft measurements from submarine ULS over the DRA, e.g. over or near the central Arctic Ocean, are available. The derived relationship may be skewed to higher ice thicknesses. Thus, Arctic ice volume derived in this study before 2004 might be overestimated. Correcting this relationship requires more spatially representative ice thickness measurements, or a well-designed parameterization scheme."

*L158-160. Confusing, please reword.*

It is changed to "All matched ice thickness and age samples in a month within a 10-year moving window are used to derive the relationship of ice age and ice thickness in that month at the fifth of the ten years."

*L169-170. This is a very speculative approach – picking bias corrections from a plot. Would you not expect this relationship to be different between fall and soring, as thinner ice grows more rapidly over winter?*

*We changed to* "However, information of such relationship is not available for other months. According to Figure 2 in RK18, the mean ice thickness in October and November is approximately 0.7 m less than the mean in February and March. Therefore, in October we assign the relationship of ice age and ice thickness the same as that in March except that ice thickness in each age category is 0.70 m less." Also added in the discussion and conclusion section that "The ice age-thickness relationship is not available for months other than in March, and we assumed such relationship is the same in October with ice thickness of 0.7 m less. With CryoSat-2 ice thickness available from October to April, we can derive such relationship in other months, and assess the linear ice thickness growth/decline assumption we made."

*L174-175. What is your physical explanation for this?*

I do not have a clear physical explanation for this. This can be a research topic for future studies. However, we have some speculations, and they are not included in the revised manuscript. We will do further investigation on this subject.

Ice ages differently, progressing through growth during freeze-up and decay during the melt seasons. Ice growth varies depending on initial thickness, as well as the air and ocean temperatures it is exposed to, and ice dynamics. The older ice gets, the more cycles of variable growth it has passed through. Older ice has been observed to be quite thick, up to 2-3m, in accumulation locations such as the Canadian Archipelago, but has also been observed to be rotten and fairly thin.

Submarines measure the ice freeboard from below with sonar, while space-based sensors such as ICESat are used to estimate thickness based on elevation differences between open water and the ice using snow depth estimates, which introduce the greatest level of uncertainty. It's possible that estimates of snow on the ice and/or localized ice deformation is responsible for the difference in thickness measurements between submarines and spaceborne altimetry in particular years.

*L177-178. 'keeping the relationship for ice older than four years', what do you mean by this? Extrapolating the thickness for very old ice?*

Changed to "As in Tschudi et al. (2016), we use linear regression to derive the relationship between ice age and thickness for ice ages from one to four years, while

the relationship for ice older than four years remain unchanged."

*L185. Flux of what?*

*Added "energy flux"*

*L198-99. Is this realistic? There are so many simplifications and assumptions here that the final result will barely reflect the underlying data.*

Since such relationship are not available from the years between, that is all can do. Once observations over those years become available, we will be more than happy to derive those relationship using those observations. Meanwhile, we have to use some assumptions and simple approaches.

*L202-3. Weekly to daily to monthly thickness. Why?*

This makes the monthly mean calculation easier, since the uneven distribution of weeks in a month. We also added such text in the discussion "even though the weekly ice age product is converted to weekly ice thickness and interpolated to daily ice thickness for monthly mean calculation. Such daily product lacks detailed temporal information content of ice thickness, and is not intended for direct comparison to point in situ ice thickness or other daily ice thickness products."

*L215. You need to explain this above with Eq 2. L232-234. Links to comment 2 above.*

Please see the response to comments regarding to Eq1 and 2 before this .

*L254-255. PIOMAS is almost being treated as the true reference here. I would urge the authors to consider comparing climatological thickness from CS2 (2010-2019) to the same years of their IAD record.*

In response to your major comments, we have carried out evaluation/validation with CS2 from 2011 to 2018. Also, we carried out evaluation/validation with Envisat from 2003 to 2010.

*L261. Also the imposed seasonal cycle, with highest ice thickness in May. PIOMAS is highest in April.*

This is based on the surface energy annual cycle. This also shows we do not tune our product based on PIOMAS. We generate our product independently, and compare our product with PIOMAS.

*L296-298. It looks like the largest decadal volume drop occurred between the 80s and 90s. Does this make sense with respect to the literature? Can you provide citations to support this? Would we not expect largest volume losses in the most recent decades, when concentration has declined strongest? Could this finding perhaps come from the trend in ice age-thickness relationship that you impose yourselves?*

All the ice-thickness relationship is based on data. We speculate that the ice thickness decrease may start to accelerate before the ice extent decrease starts to accelerate. This may be a reserch topic that needs further investigation.

*L328. Confusing. Please explain in more detail.*

Rewrote to "It should be noted that the sum of these two contributions is not 100% because the production of area means of thickness and ice area is only approximately equal to the total ice volume as shown in Eq.7."

*L360-363. Although this is simplified, it is a reasonable analysis and I would be interested to see these contributions per Arctic region as well as in total.*

I would think there are regional differences because differences in thickness and concentration spatial differences.This would be interesting to seen in future studies.

*L373-374. You need to consider and suggest an explanation for this.*

I hope I have a simple answer, but I do not. Without detailed and further analysis, I would speculate that this might be related to the linear ice growth/melting model we applied. But how exactly they are related, I am not sure. Added "The annual cycle of trends in ice volume over the Arctic Ocean appears to be opposite to the annual cycle of ice growth, which suggests this trend feature may be related to linear sea ice growth/melting model applied. How they are related and whether a more sophisticated model would remove this feature require further investigation."

*L375. Good point.*

*L378. Have you considered there may be a fundamental limit in the accuracy of ice thickness estimation for which ice age acts as a proxy? Checking the PDFs of CS2 ice thickness within each ice age category for the same month would be a perfect way to evaluate this limit, i.e. the intrinsic uncertainty of the ice age-thickness relationship.*

I will consider doing this in the future.

*L385-6. Could you have tried evaluating against the entire icebridge thickness archive or for example airborne EMI thickness datasets?*

The icebridge thickness observations can be used to derive the ice age-thickness relationship as shown in Tschudi et al. (2016). For the evaluation, IceAgeDerived product assign one single thickness value for sea ice of the same age category, thus lacks spatial changes. For that reason, I do not think the IceAgeDerived ice thickness is suitable for point comparisons. Added " even though the weekly ice age product is converted to weekly ice thickness and interpolated to daily ice thickness for monthly mean calculation. Such daily product lacks detailed temporal and spatial information content of ice thickness on the daily scale, and is not intended for direct comparison to point in situ ice thickness or other daily ice thickness products, such as CryoSat-2."

*Fig 3. Could you not base the shape of this approximation on e.g. the mean seasonal cycle of ice thickness from CryoSat-2 data?*

I think you meant "Could you base…?"
Yes. That will be the next step. In the discussion, added "With CryoSat-2 ice thickness available from October to April, we can derive ice age-thickness relationship in all these months, and assess the linear ice thickness growth/decline assumption we made."

*Fig 5. Panels a-d are comparing the derived product against in situ observations used to calibrate them. There is evidently much higher scatter versus the CS2 data, that were not used in the calibration. Add r^2, rmse and bias to these plots.*

*We added r^2, rmse and bias in table 1 and 2.*

*Fig 6 caption. Volume.*

Corrected.

*Fig 12 caption. Annual mean ice volume?*

Yes, it is monthly ice volume. Corrected.

*Fig 13. A great deal of this pattern reflects the annual cycle that was imposed from Fig 3. Can you comment on this?*

As the response to the reviewer's previous comment, I agree on this assessment and we noted this in the manuscript. We added the following text: "The annual cycle of trends in ice volume over the Arctic Ocean appears to be opposite to the annual cycle of ice growth, which suggests this trend feature may be related to linear sea ice growth/melting model applied. How they are related and whether a more sophisticated model would remove this feature require further investigation."

---

## Author Comment (AC4) · 2 Jan 2020

[revised manuscript text omitted]

Deleted: M…nthly mean Cryosat…ryoSat-2 ice thickness from CPOM, AWI, and NASA GSFC from January to April, and from October to December of 2011 to 2018 are used to calculate the spread of CryoSat-2 ice thickness within each ice age categoryies…as those in Tschudi et al. (2016). The collocated NSIDC weekly ice age with Cryosat…ryoSat-2 monthly ice thickness from all available months from…ver the period 2011 to 2018 can be used to derive such spreads in all months, as shown forthose…in …arch and November in Figure 12. Ice thickness increases with ice age for ice …ges from 1 to 4 years,…and then decreases from ice …ges 4 to 5. This is consistent with what we found…as found based on upward looking sonar data. In This trend shows similarity and difference with what were found in …schudi et al. (2016) , where …ce thickness increases from ice age from 1 to 5. Similar as…o those in Tschudi et al. (2016) (Figure 2 in their paper), one sigma…tandard deviation of the probability distribution function of Cryosat…ryoSat-2 thickness in an age category crosses another…verlaps with adjacent age categories. Please note that… t…e crosses…verlap may be a result of come from …ismatches of…n the collocation of weekly ice age with monthly ice thickness. The estimation of ice thickness from ice age can still be valid, however, the crosses lead to uncertainty in the ice thickness estimation based on ice age. …o estimate the random uncertainty of the IceAgeDerived ice volume over the Arctic Ocean,…we applied the ice thickness uncertainty,…errors bar in Figure 12, …n each ice age category (Figure 12) while…hen converting the weekly ice age to ice thickness from 1984 to 2018. The uncertainty in weekly or monthly ice volume over the Arctic Ocean is the sum of absolute of the ice volume uncertainty of all grid cells, where the ice volume uncertainty in a cell is the production…of the sea ice concentration, the grid cell area, and the ice thickness uncertainty … This at that grid, which …rovides the upper limit of…n the random uncd…rtainty in the …ce volume. The overall uncertainties in the 
[revised manuscript text omitted]

| | | **0.87** | **0.95** | 0.42 |

Formatted ... [67]
Formatted ... [69]
Formatted ... [71]
Formatted ... [66]
Formatted ... [68]
Formatted ... [70]
Formatted ... [72]
Formatted ... [74]
Formatted ... [76]
Formatted ... [78]
Formatted ... [73]
Formatted ... [65]
Formatted ... [75]
Formatted ... [77]
Formatted ... [79]
Formatted ... [81]
Formatted ... [83]
Formatted ... [85]
Formatted ... [80]
Formatted ... [82]
Formatted ... [84]
Formatted ... [86]

985

**Table 2**: Statistics of comparison of Arctic ice volume from IceAgeDerived, PIOMAS, and OTIM to ICESat 2004-2008, and CryoSat-2 2011-2018 in February/March (top row), and October/November (bottom row) over the Arctic Ocean. Correlation squared with higher than 95% confidence level is in bold.

| Ice volume | | ICESat 2004-2008 Feb/Mar (Oct/Nov) | CryoSat-2 2011-2018 Feb/Mar (Oct/Nov) |
|---|---|---|---|
| IceAgeDerived | Bias ($10^3$ km$^3$) | -0.72 | 0.29 |
| | | -3.95 | -0.66 |
| | RMSE ($10^3$ km$^3$) | 0.74 | 0.75 |
| | | 0.76 | 0.98 |
| | $R^2$ | **0.87** | 0.28 |
| | | **0.95** | 0.051 |
| PIOMAS | Bias ($10^3$ km$^3$) | 0.44 | 0.90 |
| | | -4.21 | -1.70 |
| | RMSE ($10^3$ km$^3$) | 0.98 | 0.96 |
| | | 0.68 | 0.98 |
| | $R^2$ | 0.64 | 0.14 |
| | | **0.93** | 0.19 |

| OTIM | Bias ($10^3$ km$^3$) | 4.20 | 3.87 |
|------|---------------------|------|------|
|      |                     | -4.86 | -1.63 |
|      | RMSE ($10^3$ km$^3$) | 1.20 | 1.48 |
|      |                      | 0.96 | 1.23 |
|      | $R^2$ | 0.38 | 0.011 |
|      |       | **0.96** | 0.012 |

**Formatted** ... [132]
**Formatted** ... [133]
**Formatted** ... [134]
**Formatted** ... [137]
**Formatted** ... [138]
**Formatted** ... [141]
**Formatted** ... [142]
**Formatted** ... [135]
**Formatted** ... [130]
**Formatted** ... [131]
**Formatted** ... [136]
**Formatted** ... [139]
**Formatted** ... [140]
**Formatted** ... [143]
**Formatted** ... [144]
**Formatted** ... [147]
**Formatted** ... [148]
**Formatted** ... [151]
**Formatted** ... [152]
**Formatted** ... [145]
**Formatted** ... [146]
**Formatted** ... [149]
**Formatted** ... [150]
**Formatted** ... [153]
**Formatted** ... [154]

[revised manuscript text omitted]

Superscript

| Page 10: [1] Formatted | YINGHUI LIU | 12/12/19 5:30:00 PM |

Superscript

| Page 10: [1] Formatted | YINGHUI LIU | 12/12/19 5:30:00 PM |

Superscript

| Page 10: [2] Formatted | YINGHUI LIU | 12/12/19 5:32:00 PM |

Superscript

| Page 10: [2] Formatted | YINGHUI LIU | 12/12/19 5:32:00 PM |

Superscript

| Page 10: [3] Deleted | Microsoft Office User | 12/17/19 3:56:00 PM |

Cryosat

| Page 10: [3] Deleted | Microsoft Office User | 12/17/19 3:56:00 PM |

Cryosat

| Page 10: [3] Deleted | Microsoft Office User | 12/17/19 3:56:00 PM |

Cryosat

| Page 10: [4] Deleted | Microsoft Office User | 12/17/19 3:56:00 PM |

/

| Page 10: [4] Deleted | Microsoft Office User | 12/17/19 3:56:00 PM |

/

| Page 10: [4] Deleted | Microsoft Office User | 12/17/19 3:56:00 PM |

/

| Page 10: [4] Deleted | Microsoft Office User | 12/17/19 3:56:00 PM |

/

| Page 10: [4] Deleted | Microsoft Office User | 12/17/19 3:56:00 PM |

/

| Page 10: [4] Deleted | Microsoft Office User | 12/17/19 3:56:00 PM |

/

| Page 10: [4] Deleted | Microsoft Office User | 12/17/19 3:56:00 PM |

/

| Page 10: [4] Deleted | Microsoft Office User | 12/17/19 3:56:00 PM |

/

| Page 10: [4] Deleted | Microsoft Office User | 12/17/19 3:56:00 PM |

/

| Page 10: [4] Deleted | Microsoft Office User | 12/17/19 3:56:00 PM |

/

| Page 10: [5] Formatted | YINGHUI LIU | 12/12/19 5:54:00 PM |

Superscript

| Page 10: [5] Formatted | YINGHUI LIU | 12/12/19 5:54:00 PM |

Superscript

| Page 10: [6] Deleted | Microsoft Office User | 12/17/19 3:58:00 PM |

M

| Page 10: [6] Deleted | Microsoft Office User | 12/17/19 3:58:00 PM |

M

| Page 10: [6] Deleted | Microsoft Office User | 12/17/19 3:58:00 PM |

M

| Page 10: [6] Deleted | Microsoft Office User | 12/17/19 3:58:00 PM |

M

| Page 10: [6] Deleted | Microsoft Office User | 12/17/19 3:58:00 PM |

M

| Page 10: [6] Deleted | Microsoft Office User | 12/17/19 3:58:00 PM |

M

| Page 10: [6] Deleted | Microsoft Office User | 12/17/19 3:58:00 PM |

M

| Page 10: [6] Deleted | Microsoft Office User | 12/17/19 3:58:00 PM |

M

| Page 10: [6] Deleted | Microsoft Office User | 12/17/19 3:58:00 PM |

M

| Page 10: [6] Deleted | Microsoft Office User | 12/17/19 3:58:00 PM |

M

| Page 10: [6] Deleted | Microsoft Office User | 12/17/19 3:58:00 PM |

M

| Page 10: [6] Deleted | Microsoft Office User | 12/17/19 3:58:00 PM |

M

| Page 10: [6] Deleted | Microsoft Office User | 12/17/19 3:58:00 PM |

M

| Page 10: [6] Deleted | Microsoft Office User | 12/17/19 3:58:00 PM |

M

| Page 10: [6] Deleted | Microsoft Office User | 12/17/19 3:58:00 PM |

M

| Page 10: [6] Deleted | Microsoft Office User | 12/17/19 3:58:00 PM |

M

| Page 10: [6] Deleted | Microsoft Office User | 12/17/19 3:58:00 PM |

M

| Page 10: [6] Deleted | Microsoft Office User | 12/17/19 3:58:00 PM |

M

| Page 10: [6] Deleted | Microsoft Office User | 12/17/19 3:58:00 PM |

M

| Page 10: [6] Deleted | Microsoft Office User | 12/17/19 3:58:00 PM |

M

| Page 10: [6] Deleted | Microsoft Office User | 12/17/19 3:58:00 PM |

M

| Page 10: [6] Deleted | Microsoft Office User | 12/17/19 3:58:00 PM |

M

| Page 10: [6] Deleted | Microsoft Office User | 12/17/19 3:58:00 PM |

M

| Page 10: [6] Deleted | Microsoft Office User | 12/17/19 3:58:00 PM |

M

| Page 10: [6] Deleted | Microsoft Office User | 12/17/19 3:58:00 PM |
|---|---|---|

M

| Page 10: [6] Deleted | Microsoft Office User | 12/17/19 3:58:00 PM |
|---|---|---|

M

| Page 10: [6] Deleted | Microsoft Office User | 12/17/19 3:58:00 PM |
|---|---|---|

M

| Page 10: [6] Deleted | Microsoft Office User | 12/17/19 3:58:00 PM |
|---|---|---|

M

| Page 10: [6] Deleted | Microsoft Office User | 12/17/19 3:58:00 PM |
|---|---|---|

M

| Page 10: [6] Deleted | Microsoft Office User | 12/17/19 3:58:00 PM |
|---|---|---|

M

| Page 10: [6] Deleted | Microsoft Office User | 12/17/19 3:58:00 PM |
|---|---|---|

M

| Page 10: [6] Deleted | Microsoft Office User | 12/17/19 3:58:00 PM |
|---|---|---|

M

| Page 10: [6] Deleted | Microsoft Office User | 12/17/19 3:58:00 PM |
|---|---|---|

M

| Page 10: [6] Deleted | Microsoft Office User | 12/17/19 3:58:00 PM |
|---|---|---|

M

| Page 20: [7] Formatted | Jeff Key | 12/17/19 6:51:00 AM |
|---|---|---|

Font: Bold, Complex Script Font: Not Bold

| Page 20: [8] Deleted | YINGHUI LIU | 1/1/20 8:33:00 PM |
|---|---|---|

| Page 20: [8] Deleted | YINGHUI LIU | 1/1/20 8:33:00 PM |
|---|---|---|

| Page 20: [8] Deleted | YINGHUI LIU | 1/1/20 8:33:00 PM |
|---|---|---|

| Page 20: [9] Formatted | Microsoft Office User | 12/17/19 9:54:00 AM |
|---|---|---|

Font: (Default) +Body (Times New Roman), Complex Script Font: +Body (Times New Roman), 10 pt

| Page 20: [10] Formatted Table | Microsoft Office User | 12/17/19 9:56:00 AM |
|---|---|---|

Formatted Table

**Page 20: [11] Formatted**            **Microsoft Office User**            **12/17/19 9:54:00 AM**

Font: (Default) +Body (Times New Roman), 10 pt, Font color: Auto, Complex Script Font: +Body (Times New Roman), 10 pt

**Page 20: [12] Formatted**            **Microsoft Office User**            **12/17/19 9:54:00 AM**

Font: (Default) +Body (Times New Roman), 10 pt, Not Bold, Complex Script Font: +Body (Times New Roman), 10 pt

**Page 20: [13] Formatted**            **Microsoft Office User**            **12/17/19 9:54:00 AM**

Font: (Default) +Body (Times New Roman), Complex Script Font: +Body (Times New Roman), 10 pt

**Page 20: [13] Formatted**            **Microsoft Office User**            **12/17/19 9:54:00 AM**

Font: (Default) +Body (Times New Roman), Complex Script Font: +Body (Times New Roman), 10 pt

**Page 20: [14] Formatted**            **Microsoft Office User**            **12/17/19 9:54:00 AM**

Font: (Default) +Body (Times New Roman), Complex Script Font: +Body (Times New Roman), 10 pt

**Page 20: [14] Formatted**            **Microsoft Office User**            **12/17/19 9:54:00 AM**

Font: (Default) +Body (Times New Roman), Complex Script Font: +Body (Times New Roman), 10 pt

**Page 20: [15] Formatted**            **Microsoft Office User**            **12/17/19 9:54:00 AM**

Font: (Default) +Body (Times New Roman), Complex Script Font: +Body (Times New Roman), 10 pt

**Page 20: [15] Formatted**            **Microsoft Office User**            **12/17/19 9:54:00 AM**

Font: (Default) +Body (Times New Roman), Complex Script Font: +Body (Times New Roman), 10 pt

**Page 20: [16] Formatted**            **Microsoft Office User**            **12/17/19 9:54:00 AM**

Font: (Default) +Body (Times New Roman), Complex Script Font: +Body (Times New Roman), 10 pt

**Page 20: [16] Formatted**            **Microsoft Office User**            **12/17/19 9:54:00 AM**

Font: (Default) +Body (Times New Roman), Complex Script Font: +Body (Times New Roman), 10 pt

**Page 20: [17] Formatted**            **Microsoft Office User**            **12/17/19 9:54:00 AM**

Font: (Default) +Body (Times New Roman), Complex Script Font: +Body (Times New Roman), 10 pt

**Page 20: [17] Formatted**            **Microsoft Office User**            **12/17/19 9:54:00 AM**

Font: (Default) +Body (Times New Roman), Complex Script Font: +Body (Times New Roman), 10 pt

**Page 20: [18] Formatted**            **Microsoft Office User**            **12/17/19 9:54:00 AM**

Font: (Default) +Body (Times New Roman), Complex Script Font: +Body (Times New Roman), 10 pt

**Page 20: [18] Formatted**            **Microsoft Office User**            **12/17/19 9:54:00 AM**

Font: (Default) +Body (Times New Roman), Complex Script Font: +Body (Times New Roman), 10 pt

**Page 20: [19] Formatted**            **Microsoft Office User**            **12/17/19 9:54:00 AM**

Font: (Default) +Body (Times New Roman), Complex Script Font: +Body (Times New Roman), 10 pt

**Page 20: [19] Formatted**            **Microsoft Office User**            **12/17/19 9:54:00 AM**

Font: (Default) +Body (Times New Roman), Complex Script Font: +Body (Times New Roman), 10 pt

| Page 20: [20] Formatted | Microsoft Office User | 12/17/19 9:54:00 AM |
|---|---|---|

Font: (Default) +Body (Times New Roman), Complex Script Font: +Body (Times New Roman), 10 pt

| Page 20: [20] Formatted | Microsoft Office User | 12/17/19 9:54:00 AM |
|---|---|---|

Font: (Default) +Body (Times New Roman), Complex Script Font: +Body (Times New Roman), 10 pt

| Page 20: [21] Formatted | Microsoft Office User | 12/17/19 9:54:00 AM |
|---|---|---|

Font: (Default) +Body (Times New Roman), Complex Script Font: +Body (Times New Roman), 10 pt

| Page 20: [21] Formatted | Microsoft Office User | 12/17/19 9:54:00 AM |
|---|---|---|

Font: (Default) +Body (Times New Roman), Complex Script Font: +Body (Times New Roman), 10 pt

| Page 20: [22] Formatted | Microsoft Office User | 12/17/19 9:54:00 AM |
|---|---|---|

Font: (Default) +Body (Times New Roman), Complex Script Font: +Body (Times New Roman), 10 pt

| Page 20: [22] Formatted | Microsoft Office User | 12/17/19 9:54:00 AM |
|---|---|---|

Font: (Default) +Body (Times New Roman), Complex Script Font: +Body (Times New Roman), 10 pt

| Page 20: [23] Formatted | Microsoft Office User | 12/17/19 9:54:00 AM |
|---|---|---|

Font: (Default) +Body (Times New Roman), Complex Script Font: +Body (Times New Roman), 10 pt

| Page 20: [23] Formatted | Microsoft Office User | 12/17/19 9:54:00 AM |
|---|---|---|

Font: (Default) +Body (Times New Roman), Complex Script Font: +Body (Times New Roman), 10 pt

| Page 20: [24] Formatted | Microsoft Office User | 12/17/19 9:54:00 AM |
|---|---|---|

Font: (Default) +Body (Times New Roman), Complex Script Font: +Body (Times New Roman), 10 pt

| Page 20: [24] Formatted | Microsoft Office User | 12/17/19 9:54:00 AM |
|---|---|---|

Font: (Default) +Body (Times New Roman), Complex Script Font: +Body (Times New Roman), 10 pt

| Page 20: [25] Formatted | Microsoft Office User | 12/17/19 9:54:00 AM |
|---|---|---|

Font: (Default) +Body (Times New Roman), Complex Script Font: +Body (Times New Roman), 10 pt

| Page 20: [25] Formatted | Microsoft Office User | 12/17/19 9:54:00 AM |
|---|---|---|

Font: (Default) +Body (Times New Roman), Complex Script Font: +Body (Times New Roman), 10 pt

| Page 20: [26] Formatted | Microsoft Office User | 12/17/19 9:54:00 AM |
|---|---|---|

Font: (Default) +Body (Times New Roman), Complex Script Font: +Body (Times New Roman), 10 pt

| Page 20: [26] Formatted | Microsoft Office User | 12/17/19 9:54:00 AM |
|---|---|---|

Font: (Default) +Body (Times New Roman), Complex Script Font: +Body (Times New Roman), 10 pt

| Page 20: [27] Formatted | Microsoft Office User | 12/17/19 9:54:00 AM |
|---|---|---|

Font: (Default) +Body (Times New Roman), Complex Script Font: +Body (Times New Roman), 10 pt

| Page 20: [27] Formatted | Microsoft Office User | 12/17/19 9:54:00 AM |
|---|---|---|

Font: (Default) +Body (Times New Roman), Complex Script Font: +Body (Times New Roman), 10 pt

| Page 20: [28] Formatted | Microsoft Office User | 12/17/19 9:54:00 AM |
|---|---|---|

Font: (Default) +Body (Times New Roman), Complex Script Font: +Body (Times New Roman), 10 pt

| Page 20: [29] Formatted | Microsoft Office User | 12/17/19 9:54:00 AM |
|---|---|---|

Font: (Default) +Body (Times New Roman), Complex Script Font: +Body (Times New Roman), 10 pt

| Page 20: [30] Formatted | Microsoft Office User | 12/17/19 9:54:00 AM |
|---|---|---|

Font: (Default) +Body (Times New Roman), Complex Script Font: +Body (Times New Roman), 10 pt

| Page 20: [31] Formatted | Microsoft Office User | 12/17/19 9:54:00 AM |
|---|---|---|

Font: (Default) +Body (Times New Roman), Complex Script Font: +Body (Times New Roman), 10 pt

| Page 20: [32] Formatted | Microsoft Office User | 12/17/19 9:54:00 AM |
|---|---|---|

Font: (Default) +Body (Times New Roman), Complex Script Font: +Body (Times New Roman), 10 pt

| Page 20: [33] Formatted | Microsoft Office User | 12/17/19 9:54:00 AM |
|---|---|---|

Font: (Default) +Body (Times New Roman), Complex Script Font: +Body (Times New Roman), 10 pt

| Page 20: [34] Formatted | Microsoft Office User | 12/17/19 9:54:00 AM |
|---|---|---|

Font: (Default) +Body (Times New Roman), Complex Script Font: +Body (Times New Roman), 10 pt

| Page 20: [35] Formatted | Microsoft Office User | 12/17/19 9:54:00 AM |
|---|---|---|

Font: (Default) +Body (Times New Roman), Complex Script Font: +Body (Times New Roman), 10 pt

**Page 20: [36] Formatted**      **Microsoft Office User**      **12/17/19 9:54:00 AM**

Font: (Default) +Body (Times New Roman), Complex Script Font: +Body (Times New Roman), 10 pt

**Page 20: [36] Formatted**      **Microsoft Office User**      **12/17/19 9:54:00 AM**

Font: (Default) +Body (Times New Roman), Complex Script Font: +Body (Times New Roman), 10 pt

**Page 20: [36] Formatted**      **Microsoft Office User**      **12/17/19 9:54:00 AM**

Font: (Default) +Body (Times New Roman), Complex Script Font: +Body (Times New Roman), 10 pt

**Page 20: [37] Formatted**      **Microsoft Office User**      **12/17/19 9:54:00 AM**

Font: (Default) +Body (Times New Roman), Bold, Complex Script Font: +Body (Times New Roman), 10 pt, Bold

**Page 20: [37] Formatted**      **Microsoft Office User**      **12/17/19 9:54:00 AM**

Font: (Default) +Body (Times New Roman), Bold, Complex Script Font: +Body (Times New Roman), 10 pt, Bold

**Page 20: [38] Formatted**      **Microsoft Office User**      **12/17/19 9:54:00 AM**

Font: (Default) +Body (Times New Roman), Bold, Complex Script Font: +Body (Times New Roman), 10 pt, Bold

**Page 20: [38] Formatted**      **Microsoft Office User**      **12/17/19 9:54:00 AM**

Font: (Default) +Body (Times New Roman), Bold, Complex Script Font: +Body (Times New Roman), 10 pt, Bold

**Page 20: [39] Formatted**      **Microsoft Office User**      **12/17/19 9:54:00 AM**

Font: (Default) +Body (Times New Roman), Complex Script Font: +Body (Times New Roman), 10 pt

**Page 20: [39] Formatted**      **Microsoft Office User**      **12/17/19 9:54:00 AM**

Font: (Default) +Body (Times New Roman), Complex Script Font: +Body (Times New Roman), 10 pt

**Page 20: [40] Formatted**      **Microsoft Office User**      **12/17/19 9:54:00 AM**

Font: (Default) +Body (Times New Roman), Bold, Complex Script Font: +Body (Times New Roman), 10 pt, Bold

**Page 20: [40] Formatted**      **Microsoft Office User**      **12/17/19 9:54:00 AM**

Font: (Default) +Body (Times New Roman), Bold, Complex Script Font: +Body (Times New Roman), 10 pt, Bold

**Page 20: [41] Formatted**      **Microsoft Office User**      **12/17/19 9:54:00 AM**

Font: (Default) +Body (Times New Roman), Bold, Complex Script Font: +Body (Times New Roman), 10 pt, Bold

**Page 20: [41] Formatted**      **Microsoft Office User**      **12/17/19 9:54:00 AM**

Font: (Default) +Body (Times New Roman), Bold, Complex Script Font: +Body (Times New Roman), 10 pt, Bold

**Page 20: [42] Formatted**      **Microsoft Office User**      **12/17/19 9:54:00 AM**

Font: (Default) +Body (Times New Roman), Complex Script Font: +Body (Times New Roman), 10 pt

**Page 20: [42] Formatted**      **Microsoft Office User**      **12/17/19 9:54:00 AM**

Font: (Default) +Body (Times New Roman), Complex Script Font: +Body (Times New Roman), 10 pt

**Page 20: [43] Formatted**      **Microsoft Office User**      **12/17/19 9:54:00 AM**

Font: (Default) +Body (Times New Roman), Complex Script Font: +Body (Times New Roman), 10 pt

**Page 20: [44] Formatted** | **Microsoft Office User** | **12/17/19 9:54:00 AM**

Font: (Default) +Body (Times New Roman), Complex Script Font: +Body (Times New Roman), 10 pt

**Page 20: [44] Formatted** | **Microsoft Office User** | **12/17/19 9:54:00 AM**

Font: (Default) +Body (Times New Roman), Complex Script Font: +Body (Times New Roman), 10 pt

**Page 20: [45] Formatted** | **Microsoft Office User** | **12/17/19 9:54:00 AM**

Font: (Default) +Body (Times New Roman), Complex Script Font: +Body (Times New Roman), 10 pt

**Page 20: [45] Formatted** | **Microsoft Office User** | **12/17/19 9:54:00 AM**

Font: (Default) +Body (Times New Roman), Complex Script Font: +Body (Times New Roman), 10 pt

**Page 20: [46] Formatted** | **Microsoft Office User** | **12/17/19 9:54:00 AM**

Font: (Default) +Body (Times New Roman), Complex Script Font: +Body (Times New Roman), 10 pt

**Page 20: [46] Formatted** | **Microsoft Office User** | **12/17/19 9:54:00 AM**

Font: (Default) +Body (Times New Roman), Complex Script Font: +Body (Times New Roman), 10 pt

**Page 20: [47] Formatted** | **Microsoft Office User** | **12/17/19 9:54:00 AM**

Font: (Default) +Body (Times New Roman), Complex Script Font: +Body (Times New Roman), 10 pt

**Page 20: [47] Formatted** | **Microsoft Office User** | **12/17/19 9:54:00 AM**

Font: (Default) +Body (Times New Roman), Complex Script Font: +Body (Times New Roman), 10 pt

**Page 20: [48] Formatted** | **Microsoft Office User** | **12/17/19 9:54:00 AM**

Font: (Default) +Body (Times New Roman), Complex Script Font: +Body (Times New Roman), 10 pt

**Page 20: [48] Formatted** | **Microsoft Office User** | **12/17/19 9:54:00 AM**

Font: (Default) +Body (Times New Roman), Complex Script Font: +Body (Times New Roman), 10 pt

**Page 20: [49] Formatted** | **Microsoft Office User** | **12/17/19 9:54:00 AM**

Font: (Default) +Body (Times New Roman), Complex Script Font: +Body (Times New Roman), 10 pt

**Page 20: [49] Formatted** | **Microsoft Office User** | **12/17/19 9:54:00 AM**

Font: (Default) +Body (Times New Roman), Complex Script Font: +Body (Times New Roman), 10 pt

**Page 20: [50] Formatted** | **Microsoft Office User** | **12/17/19 9:54:00 AM**

Font: (Default) +Body (Times New Roman), Complex Script Font: +Body (Times New Roman), 10 pt

**Page 20: [50] Formatted** | **Microsoft Office User** | **12/17/19 9:54:00 AM**

Font: (Default) +Body (Times New Roman), Complex Script Font: +Body (Times New Roman), 10 pt

**Page 20: [51] Formatted** | **Microsoft Office User** | **12/17/19 9:54:00 AM**

Font: (Default) +Body (Times New Roman), Complex Script Font: +Body (Times New Roman), 10 pt

| Page 20: [51] Formatted | Microsoft Office User | 12/17/19 9:54:00 AM |
|---|---|---|

Font: (Default) +Body (Times New Roman), Complex Script Font: +Body (Times New Roman), 10 pt

| Page 20: [52] Formatted | Microsoft Office User | 12/17/19 9:54:00 AM |
|---|---|---|

Font: (Default) +Body (Times New Roman), Complex Script Font: +Body (Times New Roman), 10 pt

| Page 20: [52] Formatted | Microsoft Office User | 12/17/19 9:54:00 AM |
|---|---|---|

Font: (Default) +Body (Times New Roman), Complex Script Font: +Body (Times New Roman), 10 pt

| Page 20: [53] Formatted | Microsoft Office User | 12/17/19 9:54:00 AM |
|---|---|---|

Font: (Default) +Body (Times New Roman), Complex Script Font: +Body (Times New Roman), 10 pt

| Page 20: [53] Formatted | Microsoft Office User | 12/17/19 9:54:00 AM |
|---|---|---|

Font: (Default) +Body (Times New Roman), Complex Script Font: +Body (Times New Roman), 10 pt

| Page 20: [54] Formatted | Microsoft Office User | 12/17/19 9:54:00 AM |
|---|---|---|

Font: (Default) +Body (Times New Roman), Complex Script Font: +Body (Times New Roman), 10 pt

| Page 20: [54] Formatted | Microsoft Office User | 12/17/19 9:54:00 AM |
|---|---|---|

Font: (Default) +Body (Times New Roman), Complex Script Font: +Body (Times New Roman), 10 pt

| Page 20: [55] Formatted | Microsoft Office User | 12/17/19 9:54:00 AM |
|---|---|---|

Font: (Default) +Body (Times New Roman), Complex Script Font: +Body (Times New Roman), 10 pt

| Page 20: [55] Formatted | Microsoft Office User | 12/17/19 9:54:00 AM |
|---|---|---|

Font: (Default) +Body (Times New Roman), Complex Script Font: +Body (Times New Roman), 10 pt

| Page 20: [56] Formatted | Microsoft Office User | 12/17/19 9:54:00 AM |
|---|---|---|

Font: (Default) +Body (Times New Roman), Complex Script Font: +Body (Times New Roman), 10 pt

| Page 20: [56] Formatted | Microsoft Office User | 12/17/19 9:54:00 AM |
|---|---|---|

Font: (Default) +Body (Times New Roman), Complex Script Font: +Body (Times New Roman), 10 pt

| Page 20: [57] Formatted | Microsoft Office User | 12/17/19 9:54:00 AM |
|---|---|---|

Font: (Default) +Body (Times New Roman), Complex Script Font: +Body (Times New Roman), 10 pt

| Page 20: [57] Formatted | Microsoft Office User | 12/17/19 9:54:00 AM |
|---|---|---|

Font: (Default) +Body (Times New Roman), Complex Script Font: +Body (Times New Roman), 10 pt

| Page 20: [58] Formatted | Microsoft Office User | 12/17/19 9:54:00 AM |
|---|---|---|

Font: (Default) +Body (Times New Roman), Complex Script Font: +Body (Times New Roman), 10 pt

| Page 20: [58] Formatted | Microsoft Office User | 12/17/19 9:54:00 AM |
|---|---|---|

Font: (Default) +Body (Times New Roman), Complex Script Font: +Body (Times New Roman), 10 pt

| Page 20: [59] Formatted | Microsoft Office User | 12/17/19 9:54:00 AM |
|---|---|---|

Font: (Default) +Body (Times New Roman), Bold, Complex Script Font: +Body (Times New Roman), 10 pt, Bold

| Page 20: [59] Formatted | Microsoft Office User | 12/17/19 9:54:00 AM |
|---|---|---|

Font: (Default) +Body (Times New Roman), Bold, Complex Script Font: +Body (Times New Roman), 10 pt, Bold

| Page 20: [60] Formatted | Microsoft Office User | 12/17/19 9:54:00 AM |
|---|---|---|

Font: (Default) +Body (Times New Roman), Bold, Complex Script Font: +Body (Times New Roman), 10 pt, Bold

| Page 20: [60] Formatted | Microsoft Office User | 12/17/19 9:54:00 AM |
|---|---|---|

Font: (Default) +Body (Times New Roman), Bold, Complex Script Font: +Body (Times New Roman), 10 pt, Bold

| Page 20: [61] Formatted | Microsoft Office User | 12/17/19 9:54:00 AM |
|---|---|---|

Font: (Default) +Body (Times New Roman), Complex Script Font: +Body (Times New Roman), 10 pt

| Page 20: [61] Formatted | Microsoft Office User | 12/17/19 9:54:00 AM |
|---|---|---|

Font: (Default) +Body (Times New Roman), Complex Script Font: +Body (Times New Roman), 10 pt

| Page 20: [62] Formatted | Microsoft Office User | 12/17/19 9:54:00 AM |
|---|---|---|

Font: (Default) +Body (Times New Roman), Bold, Complex Script Font: +Body (Times New Roman), 10 pt, Bold

| Page 20: [62] Formatted | Microsoft Office User | 12/17/19 9:54:00 AM |
|---|---|---|

Font: (Default) +Body (Times New Roman), Bold, Complex Script Font: +Body (Times New Roman), 10 pt, Bold

| Page 20: [63] Formatted | Microsoft Office User | 12/17/19 9:54:00 AM |
|---|---|---|

Font: (Default) +Body (Times New Roman), Complex Script Font: +Body (Times New Roman), 10 pt

| Page 20: [63] Formatted | Microsoft Office User | 12/17/19 9:54:00 AM |
|---|---|---|

Font: (Default) +Body (Times New Roman), Complex Script Font: +Body (Times New Roman), 10 pt

| Page 20: [64] Formatted | Microsoft Office User | 12/17/19 9:54:00 AM |
|---|---|---|

Font: (Default) +Body (Times New Roman), Complex Script Font: +Body (Times New Roman), 10 pt

| Page 20: [64] Formatted | Microsoft Office User | 12/17/19 9:54:00 AM |
|---|---|---|

Font: (Default) +Body (Times New Roman), Complex Script Font: +Body (Times New Roman), 10 pt

| Page 21: [65] Formatted | Microsoft Office User | 12/17/19 9:54:00 AM |
|---|---|---|

Font: (Default) +Body (Times New Roman), Complex Script Font: +Body (Times New Roman), 10 pt

| Page 21: [65] Formatted | Microsoft Office User | 12/17/19 9:54:00 AM |
|---|---|---|

Font: (Default) +Body (Times New Roman), Complex Script Font: +Body (Times New Roman), 10 pt

| Page 21: [66] Formatted | Microsoft Office User | 12/17/19 9:54:00 AM |
|---|---|---|

Font: (Default) +Body (Times New Roman), Complex Script Font: +Body (Times New Roman), 10 pt

| Page 21: [66] Formatted | Microsoft Office User | 12/17/19 9:54:00 AM |
|---|---|---|

Font: (Default) +Body (Times New Roman), Complex Script Font: +Body (Times New Roman), 10 pt

| Page 21: [67] Formatted | Microsoft Office User | 12/17/19 9:54:00 AM |
|---|---|---|

Font: (Default) +Body (Times New Roman), Complex Script Font: +Body (Times New Roman), 10 pt

| Page 21: [67] Formatted | Microsoft Office User | 12/17/19 9:54:00 AM |
|---|---|---|

Font: (Default) +Body (Times New Roman), Complex Script Font: +Body (Times New Roman), 10 pt

| Page 21: [68] Formatted | Microsoft Office User | 12/17/19 9:54:00 AM |
|---|---|---|

Font: (Default) +Body (Times New Roman), Complex Script Font: +Body (Times New Roman), 10 pt

| Page 21: [68] Formatted | Microsoft Office User | 12/17/19 9:54:00 AM |
|---|---|---|

Font: (Default) +Body (Times New Roman), Complex Script Font: +Body (Times New Roman), 10 pt

| Page 21: [69] Formatted | Microsoft Office User | 12/17/19 9:54:00 AM |
|---|---|---|

Font: (Default) +Body (Times New Roman), Complex Script Font: +Body (Times New Roman), 10 pt

| Page 21: [69] Formatted | Microsoft Office User | 12/17/19 9:54:00 AM |
|---|---|---|

Font: (Default) +Body (Times New Roman), Complex Script Font: +Body (Times New Roman), 10 pt

| Page 21: [70] Formatted | Microsoft Office User | 12/17/19 9:54:00 AM |
|---|---|---|

Font: (Default) +Body (Times New Roman), Complex Script Font: +Body (Times New Roman), 10 pt

| Page 21: [70] Formatted | Microsoft Office User | 12/17/19 9:54:00 AM |
|---|---|---|

Font: (Default) +Body (Times New Roman), Complex Script Font: +Body (Times New Roman), 10 pt

| Page 21: [71] Formatted | Microsoft Office User | 12/17/19 9:54:00 AM |
|---|---|---|

Font: (Default) +Body (Times New Roman), Complex Script Font: +Body (Times New Roman), 10 pt

| Page 21: [71] Formatted | Microsoft Office User | 12/17/19 9:54:00 AM |
|---|---|---|

Font: (Default) +Body (Times New Roman), Complex Script Font: +Body (Times New Roman), 10 pt

| Page 21: [72] Formatted | Microsoft Office User | 12/17/19 9:54:00 AM |
|---|---|---|

Font: (Default) +Body (Times New Roman), Complex Script Font: +Body (Times New Roman), 10 pt

| Page 21: [72] Formatted | Microsoft Office User | 12/17/19 9:54:00 AM |
|---|---|---|

Font: (Default) +Body (Times New Roman), Complex Script Font: +Body (Times New Roman), 10 pt

| Page 21: [73] Formatted | Microsoft Office User | 12/17/19 9:54:00 AM |
|---|---|---|

Font: (Default) +Body (Times New Roman), Complex Script Font: +Body (Times New Roman), 10 pt

| Page 21: [73] Formatted | Microsoft Office User | 12/17/19 9:54:00 AM |
|---|---|---|

Font: (Default) +Body (Times New Roman), Complex Script Font: +Body (Times New Roman), 10 pt

| Page 21: [74] Formatted | Microsoft Office User | 12/17/19 9:54:00 AM |
|---|---|---|

Font: (Default) +Body (Times New Roman), Complex Script Font: +Body (Times New Roman), 10 pt

| Page 21: [74] Formatted | Microsoft Office User | 12/17/19 9:54:00 AM |
|---|---|---|

Font: (Default) +Body (Times New Roman), Complex Script Font: +Body (Times New Roman), 10 pt

| Page 21: [75] Formatted | Microsoft Office User | 12/17/19 9:54:00 AM |
|---|---|---|

Font: (Default) +Body (Times New Roman), Complex Script Font: +Body (Times New Roman), 10 pt

| Page 21: [75] Formatted | Microsoft Office User | 12/17/19 9:54:00 AM |
|---|---|---|

Font: (Default) +Body (Times New Roman), Complex Script Font: +Body (Times New Roman), 10 pt

| Page 21: [76] Formatted | Microsoft Office User | 12/17/19 9:54:00 AM |
|---|---|---|

Font: (Default) +Body (Times New Roman), Complex Script Font: +Body (Times New Roman), 10 pt

| Page 21: [76] Formatted | Microsoft Office User | 12/17/19 9:54:00 AM |
|---|---|---|

Font: (Default) +Body (Times New Roman), Complex Script Font: +Body (Times New Roman), 10 pt

| Page 21: [77] Formatted | Microsoft Office User | 12/17/19 9:54:00 AM |
|---|---|---|

Font: (Default) +Body (Times New Roman), Complex Script Font: +Body (Times New Roman), 10 pt

| Page 21: [77] Formatted | Microsoft Office User | 12/17/19 9:54:00 AM |
|---|---|---|

Font: (Default) +Body (Times New Roman), Complex Script Font: +Body (Times New Roman), 10 pt

| Page 21: [78] Formatted | Microsoft Office User | 12/17/19 9:54:00 AM |
|---|---|---|

Font: (Default) +Body (Times New Roman), Complex Script Font: +Body (Times New Roman), 10 pt

| Page 21: [78] Formatted | Microsoft Office User | 12/17/19 9:54:00 AM |
|---|---|---|

Font: (Default) +Body (Times New Roman), Complex Script Font: +Body (Times New Roman), 10 pt

| Page 21: [79] Formatted | Microsoft Office User | 12/17/19 9:54:00 AM |
|---|---|---|

Font: (Default) +Body (Times New Roman), Complex Script Font: +Body (Times New Roman), 10 pt

| Page 21: [79] Formatted | Microsoft Office User | 12/17/19 9:54:00 AM |
|---|---|---|

Font: (Default) +Body (Times New Roman), Complex Script Font: +Body (Times New Roman), 10 pt

| Page 21: [80] Formatted | Microsoft Office User | 12/17/19 9:54:00 AM |
|---|---|---|

Font: (Default) +Body (Times New Roman), Complex Script Font: +Body (Times New Roman), 10 pt

| Page 21: [80] Formatted | Microsoft Office User | 12/17/19 9:54:00 AM |
|---|---|---|

Font: (Default) +Body (Times New Roman), Complex Script Font: +Body (Times New Roman), 10 pt

| Page 21: [81] Formatted | Microsoft Office User | 12/17/19 9:54:00 AM |
|---|---|---|

Font: (Default) +Body (Times New Roman), Bold, Complex Script Font: +Body (Times New Roman), 10 pt, Bold

| Page 21: [81] Formatted | Microsoft Office User | 12/17/19 9:54:00 AM |
|---|---|---|

Font: (Default) +Body (Times New Roman), Bold, Complex Script Font: +Body (Times New Roman), 10 pt, Bold

| Page 21: [82] Formatted | Microsoft Office User | 12/17/19 9:54:00 AM |
|---|---|---|

Font: (Default) +Body (Times New Roman), Bold, Complex Script Font: +Body (Times New Roman), 10 pt, Bold

| Page 21: [82] Formatted | Microsoft Office User | 12/17/19 9:54:00 AM |
|---|---|---|

Font: (Default) +Body (Times New Roman), Bold, Complex Script Font: +Body (Times New Roman), 10 pt, Bold

| Page 21: [83] Formatted | Microsoft Office User | 12/17/19 9:54:00 AM |
|---|---|---|

Font: (Default) +Body (Times New Roman), Complex Script Font: +Body (Times New Roman), 10 pt

| Page 21: [83] Formatted | Microsoft Office User | 12/17/19 9:54:00 AM |

Font: (Default) +Body (Times New Roman), Complex Script Font: +Body (Times New Roman), 10 pt

| Page 21: [84] Formatted | Microsoft Office User | 12/17/19 9:54:00 AM |

Font: (Default) +Body (Times New Roman), Bold, Complex Script Font: +Body (Times New Roman), 10 pt, Bold

| Page 21: [84] Formatted | Microsoft Office User | 12/17/19 9:54:00 AM |

Font: (Default) +Body (Times New Roman), Bold, Complex Script Font: +Body (Times New Roman), 10 pt, Bold

| Page 21: [85] Formatted | Microsoft Office User | 12/17/19 9:54:00 AM |

Font: (Default) +Body (Times New Roman), Complex Script Font: +Body (Times New Roman), 10 pt

| Page 21: [85] Formatted | Microsoft Office User | 12/17/19 9:54:00 AM |

Font: (Default) +Body (Times New Roman), Complex Script Font: +Body (Times New Roman), 10 pt

| Page 21: [86] Formatted | Microsoft Office User | 12/17/19 9:54:00 AM |

Font: (Default) +Body (Times New Roman), Complex Script Font: +Body (Times New Roman), 10 pt

| Page 21: [86] Formatted | Microsoft Office User | 12/17/19 9:54:00 AM |

Font: (Default) +Body (Times New Roman), Complex Script Font: +Body (Times New Roman), 10 pt

| Page 22: [87] Formatted | Jeff Key | 12/17/19 6:52:00 AM |

Font: Bold, Complex Script Font: Not Bold

| Page 22: [88] Formatted | Microsoft Office User | 12/17/19 9:54:00 AM |

Font: (Default) +Body (Times New Roman), Complex Script Font: +Body (Times New Roman), 10 pt

| Page 22: [89] Formatted Table | Jeff Key | 12/17/19 6:52:00 AM |

Formatted Table

| Page 22: [90] Formatted | Microsoft Office User | 12/17/19 9:54:00 AM |

Font: (Default) +Body (Times New Roman), 10 pt, Font color: Auto, Complex Script Font: +Body (Times New Roman), 10 pt

| Page 22: [91] Formatted | Microsoft Office User | 12/17/19 9:54:00 AM |

Font: (Default) +Body (Times New Roman), 10 pt, Not Bold, Complex Script Font: +Body (Times New Roman), 10 pt

| Page 22: [92] Formatted | Microsoft Office User | 12/17/19 9:54:00 AM |

Font: (Default) +Body (Times New Roman), Complex Script Font: +Body (Times New Roman), 10 pt

| Page 22: [92] Formatted | Microsoft Office User | 12/17/19 9:54:00 AM |

Font: (Default) +Body (Times New Roman), Complex Script Font: +Body (Times New Roman), 10 pt

| Page 22: [93] Formatted | Microsoft Office User | 12/17/19 9:54:00 AM |

Font: (Default) +Body (Times New Roman), Complex Script Font: +Body (Times New Roman), 10 pt

| Page 22: [93] Formatted | Microsoft Office User | 12/17/19 9:54:00 AM |

Font: (Default) +Body (Times New Roman), Complex Script Font: +Body (Times New Roman), 10 pt

| Page 22: [94] Formatted | Microsoft Office User | 12/17/19 9:54:00 AM |
|---|---|---|

Font: (Default) +Body (Times New Roman), Complex Script Font: +Body (Times New Roman), 10 pt

| Page 22: [94] Formatted | Microsoft Office User | 12/17/19 9:54:00 AM |
|---|---|---|

Font: (Default) +Body (Times New Roman), Complex Script Font: +Body (Times New Roman), 10 pt

| Page 22: [95] Formatted | Microsoft Office User | 12/17/19 9:54:00 AM |
|---|---|---|

Font: (Default) +Body (Times New Roman), Complex Script Font: +Body (Times New Roman), 10 pt

| Page 22: [95] Formatted | Microsoft Office User | 12/17/19 9:54:00 AM |
|---|---|---|

Font: (Default) +Body (Times New Roman), Complex Script Font: +Body (Times New Roman), 10 pt

| Page 22: [96] Formatted | Microsoft Office User | 12/17/19 9:54:00 AM |
|---|---|---|

Font: (Default) +Body (Times New Roman), Complex Script Font: +Body (Times New Roman), 10 pt

| Page 22: [96] Formatted | Microsoft Office User | 12/17/19 9:54:00 AM |
|---|---|---|

Font: (Default) +Body (Times New Roman), Complex Script Font: +Body (Times New Roman), 10 pt

| Page 22: [97] Formatted | Microsoft Office User | 12/17/19 9:54:00 AM |
|---|---|---|

Font: (Default) +Body (Times New Roman), Complex Script Font: +Body (Times New Roman), 10 pt

| Page 22: [97] Formatted | Microsoft Office User | 12/17/19 9:54:00 AM |
|---|---|---|

Font: (Default) +Body (Times New Roman), Complex Script Font: +Body (Times New Roman), 10 pt

| Page 22: [98] Formatted | Microsoft Office User | 12/17/19 9:54:00 AM |
|---|---|---|

Font: (Default) +Body (Times New Roman), Complex Script Font: +Body (Times New Roman), 10 pt

| Page 22: [98] Formatted | Microsoft Office User | 12/17/19 9:54:00 AM |
|---|---|---|

Font: (Default) +Body (Times New Roman), Complex Script Font: +Body (Times New Roman), 10 pt

| Page 22: [99] Formatted | Microsoft Office User | 12/17/19 9:54:00 AM |
|---|---|---|

Font: (Default) +Body (Times New Roman), Complex Script Font: +Body (Times New Roman), 10 pt

| Page 22: [99] Formatted | Microsoft Office User | 12/17/19 9:54:00 AM |
|---|---|---|

Font: (Default) +Body (Times New Roman), Complex Script Font: +Body (Times New Roman), 10 pt

| Page 22: [100] Formatted | Microsoft Office User | 12/17/19 9:54:00 AM |
|---|---|---|

Font: (Default) +Body (Times New Roman), Complex Script Font: +Body (Times New Roman), 10 pt

| Page 22: [100] Formatted | Microsoft Office User | 12/17/19 9:54:00 AM |
|---|---|---|

Font: (Default) +Body (Times New Roman), Complex Script Font: +Body (Times New Roman), 10 pt

| Page 22: [101] Formatted | Microsoft Office User | 12/17/19 9:54:00 AM |
|---|---|---|

Font: (Default) +Body (Times New Roman), Complex Script Font: +Body (Times New Roman), 10 pt

| Page 22: [101] Formatted | Microsoft Office User | 12/17/19 9:54:00 AM |
|---|---|---|

Font: (Default) +Body (Times New Roman), Complex Script Font: +Body (Times New Roman), 10 pt

**Page 22: [102] Formatted**      **Microsoft Office User**      **12/17/19 9:54:00 AM**

Font: (Default) +Body (Times New Roman), Complex Script Font: +Body (Times New Roman), 10 pt

**Page 22: [103] Formatted**      **Microsoft Office User**      **12/17/19 9:54:00 AM**

Font: (Default) +Body (Times New Roman), Complex Script Font: +Body (Times New Roman), 10 pt

**Page 22: [104] Formatted**      **Microsoft Office User**      **12/17/19 9:54:00 AM**

Font: (Default) +Body (Times New Roman), Complex Script Font: +Body (Times New Roman), 10 pt

**Page 22: [105] Formatted**      **Microsoft Office User**      **12/17/19 9:54:00 AM**

Font: (Default) +Body (Times New Roman), Complex Script Font: +Body (Times New Roman), 10 pt

**Page 22: [106] Formatted**      **Microsoft Office User**      **12/17/19 9:54:00 AM**

Font: (Default) +Body (Times New Roman), Complex Script Font: +Body (Times New Roman), 10 pt

**Page 22: [107] Formatted**      **Microsoft Office User**      **12/17/19 9:54:00 AM**

Font: (Default) +Body (Times New Roman), Complex Script Font: +Body (Times New Roman), 10 pt

**Page 22: [108] Formatted**      **Microsoft Office User**      **12/17/19 9:54:00 AM**

Font: (Default) +Body (Times New Roman), Complex Script Font: +Body (Times New Roman), 10 pt

**Page 22: [109] Formatted**      **Microsoft Office User**      **12/17/19 9:54:00 AM**

Font: (Default) +Body (Times New Roman), Complex Script Font: +Body (Times New Roman), 10 pt

| Page 22: [110] Formatted | Microsoft Office User | 12/17/19 9:54:00 AM |
|---|---|---|

Font: (Default) +Body (Times New Roman), Bold, Complex Script Font: +Body (Times New Roman), 10 pt, Bold

| Page 22: [110] Formatted | Microsoft Office User | 12/17/19 9:54:00 AM |
|---|---|---|

Font: (Default) +Body (Times New Roman), Bold, Complex Script Font: +Body (Times New Roman), 10 pt, Bold

| Page 22: [111] Formatted | Microsoft Office User | 12/17/19 9:54:00 AM |
|---|---|---|

Font: (Default) +Body (Times New Roman), Bold, Complex Script Font: +Body (Times New Roman), 10 pt, Bold

| Page 22: [111] Formatted | Microsoft Office User | 12/17/19 9:54:00 AM |
|---|---|---|

Font: (Default) +Body (Times New Roman), Bold, Complex Script Font: +Body (Times New Roman), 10 pt, Bold

| Page 22: [112] Formatted | Microsoft Office User | 12/17/19 9:54:00 AM |
|---|---|---|

Font: (Default) +Body (Times New Roman), Complex Script Font: +Body (Times New Roman), 10 pt

| Page 22: [112] Formatted | Microsoft Office User | 12/17/19 9:54:00 AM |
|---|---|---|

Font: (Default) +Body (Times New Roman), Complex Script Font: +Body (Times New Roman), 10 pt

| Page 22: [113] Formatted | Microsoft Office User | 12/17/19 9:54:00 AM |
|---|---|---|

Font: (Default) +Body (Times New Roman), Complex Script Font: +Body (Times New Roman), 10 pt

| Page 22: [113] Formatted | Microsoft Office User | 12/17/19 9:54:00 AM |
|---|---|---|

Font: (Default) +Body (Times New Roman), Complex Script Font: +Body (Times New Roman), 10 pt

| Page 22: [114] Formatted | Microsoft Office User | 12/17/19 9:54:00 AM |
|---|---|---|

Font: (Default) +Body (Times New Roman), Complex Script Font: +Body (Times New Roman), 10 pt

| Page 22: [114] Formatted | Microsoft Office User | 12/17/19 9:54:00 AM |
|---|---|---|

Font: (Default) +Body (Times New Roman), Complex Script Font: +Body (Times New Roman), 10 pt

| Page 22: [115] Formatted | Microsoft Office User | 12/17/19 9:54:00 AM |
|---|---|---|

Font: (Default) +Body (Times New Roman), Complex Script Font: +Body (Times New Roman), 10 pt

| Page 22: [115] Formatted | Microsoft Office User | 12/17/19 9:54:00 AM |
|---|---|---|

Font: (Default) +Body (Times New Roman), Complex Script Font: +Body (Times New Roman), 10 pt

| Page 22: [116] Formatted | Microsoft Office User | 12/17/19 9:54:00 AM |
|---|---|---|

Font: (Default) +Body (Times New Roman), Complex Script Font: +Body (Times New Roman), 10 pt

| Page 22: [116] Formatted | Microsoft Office User | 12/17/19 9:54:00 AM |
|---|---|---|

Font: (Default) +Body (Times New Roman), Complex Script Font: +Body (Times New Roman), 10 pt

| Page 22: [117] Formatted | Microsoft Office User | 12/17/19 9:54:00 AM |
|---|---|---|

Font: (Default) +Body (Times New Roman), Complex Script Font: +Body (Times New Roman), 10 pt

| Page 22: [117] Formatted | Microsoft Office User | 12/17/19 9:54:00 AM |
|---|---|---|

Font: (Default) +Body (Times New Roman), Complex Script Font: +Body (Times New Roman), 10 pt

| Page 22: [118] Formatted | Microsoft Office User | 12/17/19 9:54:00 AM |
|---|---|---|

Font: (Default) +Body (Times New Roman), Complex Script Font: +Body (Times New Roman), 10 pt

| Page 22: [119] Formatted | Microsoft Office User | 12/17/19 9:54:00 AM |
|---|---|---|

Font: (Default) +Body (Times New Roman), Complex Script Font: +Body (Times New Roman), 10 pt

| Page 22: [120] Formatted | Microsoft Office User | 12/17/19 9:54:00 AM |
|---|---|---|

Font: (Default) +Body (Times New Roman), Complex Script Font: +Body (Times New Roman), 10 pt

| Page 22: [121] Formatted | Microsoft Office User | 12/17/19 9:54:00 AM |
|---|---|---|

Font: (Default) +Body (Times New Roman), Complex Script Font: +Body (Times New Roman), 10 pt

| Page 22: [122] Formatted | Microsoft Office User | 12/17/19 9:54:00 AM |
|---|---|---|

Font: (Default) +Body (Times New Roman), Complex Script Font: +Body (Times New Roman), 10 pt

| Page 22: [123] Formatted | Microsoft Office User | 12/17/19 9:54:00 AM |
|---|---|---|

Font: (Default) +Body (Times New Roman), Complex Script Font: +Body (Times New Roman), 10 pt

| Page 22: [124] Formatted | Microsoft Office User | 12/17/19 9:54:00 AM |
|---|---|---|

Font: (Default) +Body (Times New Roman), Complex Script Font: +Body (Times New Roman), 10 pt

| Page 22: [125] Formatted | Microsoft Office User | 12/17/19 9:54:00 AM |
|---|---|---|

Font: (Default) +Body (Times New Roman), Complex Script Font: +Body (Times New Roman), 10 pt

**Page 22: [126] Formatted**      **Microsoft Office User**      **12/17/19 9:54:00 AM**

Font: (Default) +Body (Times New Roman), Complex Script Font: +Body (Times New Roman), 10 pt

**Page 22: [126] Formatted**      **Microsoft Office User**      **12/17/19 9:54:00 AM**

Font: (Default) +Body (Times New Roman), Complex Script Font: +Body (Times New Roman), 10 pt

**Page 22: [127] Formatted**      **Microsoft Office User**      **12/17/19 9:54:00 AM**

Font: (Default) +Body (Times New Roman), Bold, Complex Script Font: +Body (Times New Roman), 10 pt, Bold

**Page 22: [127] Formatted**      **Microsoft Office User**      **12/17/19 9:54:00 AM**

Font: (Default) +Body (Times New Roman), Bold, Complex Script Font: +Body (Times New Roman), 10 pt, Bold

**Page 22: [128] Formatted**      **Microsoft Office User**      **12/17/19 9:54:00 AM**

Font: (Default) +Body (Times New Roman), Complex Script Font: +Body (Times New Roman), 10 pt

**Page 22: [128] Formatted**      **Microsoft Office User**      **12/17/19 9:54:00 AM**

Font: (Default) +Body (Times New Roman), Complex Script Font: +Body (Times New Roman), 10 pt

**Page 22: [129] Formatted**      **Microsoft Office User**      **12/17/19 9:54:00 AM**

Font: (Default) +Body (Times New Roman), Complex Script Font: +Body (Times New Roman), 10 pt

**Page 22: [129] Formatted**      **Microsoft Office User**      **12/17/19 9:54:00 AM**

Font: (Default) +Body (Times New Roman), Complex Script Font: +Body (Times New Roman), 10 pt

**Page 23: [130] Formatted**      **Microsoft Office User**      **12/17/19 9:54:00 AM**

Font: (Default) +Body (Times New Roman), Complex Script Font: +Body (Times New Roman), 10 pt

**Page 23: [131] Formatted**      **Microsoft Office User**      **12/17/19 9:54:00 AM**

Font: (Default) +Body (Times New Roman), 10 pt, Font color: Auto, Complex Script Font: +Body (Times New Roman), 10 pt

**Page 23: [132] Formatted**      **Microsoft Office User**      **12/17/19 9:54:00 AM**

Font: (Default) +Body (Times New Roman), 10 pt, Font color: Auto, Complex Script Font: +Body (Times New Roman), 10 pt

**Page 23: [133] Formatted**      **Microsoft Office User**      **12/17/19 9:54:00 AM**

Font: (Default) +Body (Times New Roman), Complex Script Font: +Body (Times New Roman), 10 pt

**Page 23: [134] Formatted**      **Microsoft Office User**      **12/17/19 9:54:00 AM**

Font: (Default) +Body (Times New Roman), 10 pt, Font color: Auto, Complex Script Font: +Body (Times New Roman), 10 pt

**Page 23: [135] Formatted**      **Microsoft Office User**      **12/17/19 9:54:00 AM**

Font: (Default) +Body (Times New Roman), Complex Script Font: +Body (Times New Roman), 10 pt

**Page 23: [136] Formatted**      **Microsoft Office User**      **12/17/19 9:54:00 AM**

Font: (Default) +Body (Times New Roman), 10 pt, Font color: Auto, Complex Script Font: +Body (Times New Roman), 10 pt

| Page 23: [137] Formatted | Microsoft Office User | 12/17/19 9:54:00 AM |

Font: (Default) +Body (Times New Roman), Complex Script Font: +Body (Times New Roman), 10 pt

| Page 23: [138] Formatted | Microsoft Office User | 12/17/19 9:54:00 AM |

Font: (Default) +Body (Times New Roman), 10 pt, Font color: Auto, Complex Script Font: +Body (Times New Roman), 10 pt

| Page 23: [139] Formatted | Microsoft Office User | 12/17/19 9:54:00 AM |

Font: (Default) +Body (Times New Roman), Complex Script Font: +Body (Times New Roman), 10 pt

| Page 23: [140] Formatted | Microsoft Office User | 12/17/19 9:54:00 AM |

Font: (Default) +Body (Times New Roman), 10 pt, Font color: Auto, Complex Script Font: +Body (Times New Roman), 10 pt

| Page 23: [141] Formatted | Microsoft Office User | 12/17/19 9:54:00 AM |

Font: (Default) +Body (Times New Roman), Complex Script Font: +Body (Times New Roman), 10 pt

| Page 23: [142] Formatted | Microsoft Office User | 12/17/19 9:54:00 AM |

Font: (Default) +Body (Times New Roman), 10 pt, Font color: Auto, Complex Script Font: +Body (Times New Roman), 10 pt

| Page 23: [143] Formatted | Microsoft Office User | 12/17/19 9:54:00 AM |

Font: (Default) +Body (Times New Roman), Complex Script Font: +Body (Times New Roman), 10 pt

| Page 23: [144] Formatted | Microsoft Office User | 12/17/19 9:54:00 AM |

Font: (Default) +Body (Times New Roman), 10 pt, Font color: Auto, Complex Script Font: +Body (Times New Roman), 10 pt

| Page 23: [145] Formatted | Microsoft Office User | 12/17/19 9:54:00 AM |

Font: (Default) +Body (Times New Roman), Complex Script Font: +Body (Times New Roman), 10 pt

| Page 23: [146] Formatted | Microsoft Office User | 12/17/19 9:54:00 AM |

Font: (Default) +Body (Times New Roman), 10 pt, Complex Script Font: +Body (Times New Roman), 10 pt

| Page 23: [147] Formatted | Microsoft Office User | 12/17/19 9:54:00 AM |

Font: (Default) +Body (Times New Roman), Complex Script Font: +Body (Times New Roman), 10 pt

| Page 23: [148] Formatted | Microsoft Office User | 12/17/19 9:54:00 AM |

Font: (Default) +Body (Times New Roman), 10 pt, Complex Script Font: +Body (Times New Roman), 10 pt

| Page 23: [149] Formatted | Microsoft Office User | 12/17/19 9:54:00 AM |

Font: (Default) +Body (Times New Roman), Bold, Complex Script Font: +Body (Times New Roman), 10 pt, Bold

| Page 23: [150] Formatted | Microsoft Office User | 12/17/19 9:54:00 AM |

Font: (Default) +Body (Times New Roman), 10 pt, Bold, Complex Script Font: +Body (Times New Roman), 10 pt, Bold

| Page 23: [151] Formatted | Microsoft Office User | 12/17/19 9:54:00 AM |
|---|---|---|

Font: (Default) +Body (Times New Roman), Complex Script Font: +Body (Times New Roman), 10 pt

| Page 23: [152] Formatted | Microsoft Office User | 12/17/19 9:54:00 AM |
|---|---|---|

Font: (Default) +Body (Times New Roman), 10 pt, Complex Script Font: +Body (Times New Roman), 10 pt

| Page 23: [153] Formatted | Microsoft Office User | 12/17/19 9:54:00 AM |
|---|---|---|

Font: (Default) +Body (Times New Roman), Complex Script Font: +Body (Times New Roman), 10 pt

| Page 23: [154] Formatted | Microsoft Office User | 12/17/19 9:54:00 AM |
|---|---|---|

Font: (Default) +Body (Times New Roman), 10 pt, Complex Script Font: +Body (Times New Roman), 10 pt

| Page 24: [155] Formatted | Jeff Key | 12/17/19 6:52:00 AM |
|---|---|---|

Font: 9 pt, Bold, Complex Script Font: 9 pt, Not Bold

| Page 24: [155] Formatted | Jeff Key | 12/17/19 6:52:00 AM |
|---|---|---|

Font: 9 pt, Bold, Complex Script Font: 9 pt, Not Bold

| Page 24: [156] Formatted | YINGHUI LIU | 12/12/19 5:43:00 PM |
|---|---|---|

Font: 9 pt, Complex Script Font: 9 pt

| Page 24: [157] Formatted | YINGHUI LIU | 12/12/19 5:43:00 PM |
|---|---|---|

Font: 9 pt, Complex Script Font: 9 pt

| Page 24: [158] Formatted | Microsoft Office User | 12/17/19 9:56:00 AM |
|---|---|---|

Font: 10 pt, Not Bold, Complex Script Font: Times New Roman, 10 pt

| Page 24: [159] Formatted Table | Microsoft Office User | 12/17/19 2:34:00 PM |
|---|---|---|

Formatted Table

| Page 24: [160] Formatted | Microsoft Office User | 12/17/19 9:56:00 AM |
|---|---|---|

Font: 10 pt, Complex Script Font: Times New Roman, 10 pt

| Page 24: [161] Formatted | Microsoft Office User | 12/17/19 9:56:00 AM |
|---|---|---|

Font: 10 pt, Complex Script Font: Times New Roman, 10 pt

| Page 24: [162] Formatted | Microsoft Office User | 12/17/19 9:56:00 AM |
|---|---|---|

Font: 10 pt, Complex Script Font: Times New Roman, 10 pt

| Page 24: [163] Formatted | Microsoft Office User | 12/17/19 9:56:00 AM |
|---|---|---|

Font: 10 pt, Complex Script Font: Times New Roman, 10 pt

| Page 24: [164] Formatted | Microsoft Office User | 12/17/19 9:56:00 AM |
|---|---|---|

Font: 10 pt, Complex Script Font: Times New Roman, 10 pt

| **Page 24: [165] Formatted** | **Microsoft Office User** | **12/17/19 9:56:00 AM** |
|---|---|---|

Font: 10 pt, Complex Script Font: Times New Roman, 10 pt

| **Page 24: [166] Formatted** | **Microsoft Office User** | **12/17/19 9:56:00 AM** |
|---|---|---|

Font: 10 pt, Complex Script Font: Times New Roman, 10 pt

| **Page 24: [167] Formatted** | **Microsoft Office User** | **12/17/19 9:56:00 AM** |
|---|---|---|

Font: 10 pt, Complex Script Font: Times New Roman, 10 pt

| **Page 24: [168] Formatted** | **Microsoft Office User** | **12/17/19 9:56:00 AM** |
|---|---|---|

Font: 10 pt, Complex Script Font: Times New Roman, 10 pt

| **Page 24: [169] Formatted** | **Microsoft Office User** | **12/17/19 9:56:00 AM** |
|---|---|---|

Font: 10 pt, Complex Script Font: Times New Roman, 10 pt

| **Page 24: [170] Formatted** | **Microsoft Office User** | **12/17/19 9:56:00 AM** |
|---|---|---|

Font: 10 pt, Complex Script Font: Times New Roman, 10 pt

| **Page 24: [171] Formatted** | **Microsoft Office User** | **12/17/19 9:56:00 AM** |
|---|---|---|

Font: 10 pt, Complex Script Font: Times New Roman, 10 pt

| **Page 24: [172] Formatted** | **Microsoft Office User** | **12/17/19 9:56:00 AM** |
|---|---|---|

Font: 10 pt, Complex Script Font: Times New Roman, 10 pt

| **Page 24: [173] Formatted** | **Microsoft Office User** | **12/17/19 9:56:00 AM** |
|---|---|---|

Font: 10 pt, Complex Script Font: Times New Roman, 10 pt

| **Page 24: [174] Formatted** | **Microsoft Office User** | **12/17/19 9:56:00 AM** |
|---|---|---|

Font: 10 pt, Complex Script Font: Times New Roman, 10 pt

| **Page 24: [175] Formatted** | **Microsoft Office User** | **12/17/19 9:56:00 AM** |
|---|---|---|

Font: 10 pt, Complex Script Font: Times New Roman, 10 pt

| **Page 24: [176] Formatted** | **Microsoft Office User** | **12/17/19 9:56:00 AM** |
|---|---|---|

Font: 10 pt, Complex Script Font: Times New Roman, 10 pt

| **Page 24: [177] Formatted** | **Microsoft Office User** | **12/17/19 9:56:00 AM** |
|---|---|---|

Font: 10 pt, Complex Script Font: Times New Roman, 10 pt

| **Page 24: [178] Formatted** | **Microsoft Office User** | **12/17/19 9:56:00 AM** |
|---|---|---|

Font: 10 pt, Complex Script Font: Times New Roman, 10 pt

| **Page 24: [179] Formatted** | **Microsoft Office User** | **12/17/19 9:56:00 AM** |
|---|---|---|

Font: 10 pt, Complex Script Font: Times New Roman, 10 pt

| **Page 24: [180] Formatted** | **Microsoft Office User** | **12/17/19 9:56:00 AM** |
|---|---|---|

Font: 10 pt, Complex Script Font: Times New Roman, 10 pt

**Page 24: [181] Formatted** | **Microsoft Office User** | **12/17/19 9:56:00 AM**

Font: 10 pt, Complex Script Font: Times New Roman, 10 pt

**Page 24: [182] Formatted** | **Microsoft Office User** | **12/17/19 9:56:00 AM**

Font: 10 pt, Complex Script Font: Times New Roman, 10 pt

**Page 24: [183] Formatted** | **Microsoft Office User** | **12/17/19 9:56:00 AM**

Font: 10 pt, Complex Script Font: Times New Roman, 10 pt

**Page 24: [184] Formatted** | **Microsoft Office User** | **12/17/19 9:56:00 AM**

Font: 10 pt, Complex Script Font: Times New Roman, 10 pt

**Page 24: [185] Formatted** | **Microsoft Office User** | **12/17/19 9:56:00 AM**

Font: 10 pt, Complex Script Font: Times New Roman, 10 pt

**Page 24: [186] Formatted** | **Microsoft Office User** | **12/17/19 9:56:00 AM**

Font: 10 pt, Complex Script Font: Times New Roman, 10 pt

**Page 24: [187] Formatted** | **Microsoft Office User** | **12/17/19 9:56:00 AM**

Font: 10 pt, Complex Script Font: Times New Roman, 10 pt

**Page 24: [188] Formatted** | **Microsoft Office User** | **12/17/19 9:56:00 AM**

Font: 10 pt, Complex Script Font: Times New Roman, 10 pt

**Page 24: [189] Formatted** | **Microsoft Office User** | **12/17/19 9:56:00 AM**

Font: 10 pt, Complex Script Font: Times New Roman, 10 pt

**Page 24: [190] Formatted** | **Microsoft Office User** | **12/17/19 9:56:00 AM**

Font: 10 pt, Complex Script Font: Times New Roman, 10 pt

**Page 24: [191] Formatted** | **Microsoft Office User** | **12/17/19 9:56:00 AM**

Font: 10 pt, Complex Script Font: Times New Roman, 10 pt

**Page 24: [192] Formatted** | **Microsoft Office User** | **12/17/19 9:56:00 AM**

Font: 10 pt, Complex Script Font: Times New Roman, 10 pt

**Page 24: [193] Formatted** | **Microsoft Office User** | **12/17/19 9:56:00 AM**

Font: 10 pt, Complex Script Font: Times New Roman, 10 pt

**Page 24: [194] Formatted** | **Microsoft Office User** | **12/17/19 9:56:00 AM**

Font: 10 pt, Complex Script Font: Times New Roman, 10 pt

**Page 24: [195] Formatted** | **Microsoft Office User** | **12/17/19 9:56:00 AM**

Font: 10 pt, Complex Script Font: Times New Roman, 10 pt

**Page 24: [196] Deleted** | **Microsoft Office User** | **12/17/19 2:34:00 PM**

thickness

**Page 24: [196] Deleted          Microsoft Office User          12/17/19 2:34:00 PM**

thickness

**Page 24: [197] Formatted          Microsoft Office User          12/17/19 9:56:00 AM**

Font: 10 pt, Complex Script Font: Times New Roman, 10 pt

**Page 24: [198] Formatted          Microsoft Office User          12/17/19 9:56:00 AM**

Font: 10 pt, Complex Script Font: Times New Roman, 10 pt

**Page 24: [199] Formatted          Microsoft Office User          12/17/19 9:56:00 AM**

Font: 10 pt, Complex Script Font: Times New Roman, 10 pt

**Page 24: [200] Formatted          Microsoft Office User          12/17/19 9:56:00 AM**

Font: 10 pt, Complex Script Font: Times New Roman, 10 pt

**Page 24: [201] Formatted          Microsoft Office User          12/17/19 9:56:00 AM**

Font: 10 pt, Complex Script Font: Times New Roman, 10 pt

**Page 24: [202] Formatted          Microsoft Office User          12/17/19 9:56:00 AM**

Font: 10 pt, Complex Script Font: Times New Roman, 10 pt

**Page 24: [203] Formatted          Microsoft Office User          12/17/19 9:56:00 AM**

Font: 10 pt, Complex Script Font: Times New Roman, 10 pt

**Page 24: [204] Formatted          Microsoft Office User          12/17/19 9:56:00 AM**

Font: 10 pt, Complex Script Font: Times New Roman, 10 pt

**Page 24: [205] Formatted          Microsoft Office User          12/17/19 9:56:00 AM**

Font: 10 pt, Complex Script Font: Times New Roman, 10 pt

**Page 24: [206] Formatted          Microsoft Office User          12/17/19 9:56:00 AM**

Font: 10 pt, Complex Script Font: Times New Roman, 10 pt

**Page 24: [207] Formatted          Microsoft Office User          12/17/19 9:56:00 AM**

Font: 10 pt, Complex Script Font: Times New Roman, 10 pt

**Page 24: [208] Formatted          Microsoft Office User          12/17/19 9:56:00 AM**

Font: 10 pt, Complex Script Font: Times New Roman, 10 pt

**Page 24: [209] Formatted          Microsoft Office User          12/17/19 9:56:00 AM**

Font: 10 pt, Complex Script Font: Times New Roman, 10 pt

**Page 24: [210] Formatted          Microsoft Office User          12/17/19 9:56:00 AM**

Font: 10 pt, Complex Script Font: Times New Roman, 10 pt

**Page 24: [211] Formatted          Microsoft Office User          12/17/19 9:56:00 AM**

Font: 10 pt, Complex Script Font: Times New Roman, 10 pt

| **Page 24: [212] Formatted** | **Microsoft Office User** | **12/17/19 9:56:00 AM** |
|---|---|---|

Font: 10 pt, Complex Script Font: Times New Roman, 10 pt

| **Page 26: [213] Formatted** | **Jeff Key** | **12/17/19 6:59:00 AM** |
|---|---|---|

Font: 9 pt, Bold, Complex Script Font: 9 pt, Not Bold

| **Page 26: [214] Formatted** | **YINGHUI LIU** | **12/12/19 5:49:00 PM** |
|---|---|---|

Font: 9 pt, Complex Script Font: 9 pt

| **Page 26: [215] Formatted** | **YINGHUI LIU** | **12/12/19 5:49:00 PM** |
|---|---|---|

Font: 9 pt, Complex Script Font: 9 pt

| **Page 26: [216] Formatted** | **Microsoft Office User** | **12/17/19 9:56:00 AM** |
|---|---|---|

Font: (Default) +Body (Times New Roman), 10 pt, Not Bold, Complex Script Font: +Body (Times New Roman), 10 pt

| **Page 26: [217] Formatted Table** | **Jeff Key** | **12/17/19 6:58:00 AM** |
|---|---|---|

Formatted Table

| **Page 26: [218] Formatted** | **Microsoft Office User** | **12/17/19 9:56:00 AM** |
|---|---|---|

Font: (Default) +Body (Times New Roman), Complex Script Font: +Body (Times New Roman), 10 pt

| **Page 26: [218] Formatted** | **Microsoft Office User** | **12/17/19 9:56:00 AM** |
|---|---|---|

Font: (Default) +Body (Times New Roman), Complex Script Font: +Body (Times New Roman), 10 pt

| **Page 26: [219] Formatted** | **Microsoft Office User** | **12/17/19 9:56:00 AM** |
|---|---|---|

Font: (Default) +Body (Times New Roman), Complex Script Font: +Body (Times New Roman), 10 pt

| **Page 26: [220] Formatted** | **Microsoft Office User** | **12/17/19 9:56:00 AM** |
|---|---|---|

Font: (Default) +Body (Times New Roman), Complex Script Font: +Body (Times New Roman), 10 pt

| **Page 26: [220] Formatted** | **Microsoft Office User** | **12/17/19 9:56:00 AM** |
|---|---|---|

Font: (Default) +Body (Times New Roman), Complex Script Font: +Body (Times New Roman), 10 pt

| **Page 26: [221] Formatted** | **Microsoft Office User** | **12/17/19 9:56:00 AM** |
|---|---|---|

Font: (Default) +Body (Times New Roman), Complex Script Font: +Body (Times New Roman), 10 pt

| **Page 26: [221] Formatted** | **Microsoft Office User** | **12/17/19 9:56:00 AM** |
|---|---|---|

Font: (Default) +Body (Times New Roman), Complex Script Font: +Body (Times New Roman), 10 pt

| **Page 26: [222] Formatted** | **Microsoft Office User** | **12/17/19 9:56:00 AM** |
|---|---|---|

Font: (Default) +Body (Times New Roman), Complex Script Font: +Body (Times New Roman), 10 pt

| **Page 26: [222] Formatted** | **Microsoft Office User** | **12/17/19 9:56:00 AM** |
|---|---|---|

Font: (Default) +Body (Times New Roman), Complex Script Font: +Body (Times New Roman), 10 pt

**Page 26: [223] Formatted**            **Microsoft Office User**            **12/17/19 9:56:00 AM**

Font: (Default) +Body (Times New Roman), Complex Script Font: +Body (Times New Roman), 10 pt

**Page 26: [224] Formatted**            **Microsoft Office User**            **12/17/19 9:56:00 AM**

Font: (Default) +Body (Times New Roman), Complex Script Font: +Body (Times New Roman), 10 pt

**Page 26: [225] Formatted**            **Microsoft Office User**            **12/17/19 9:56:00 AM**

Font: (Default) +Body (Times New Roman), Complex Script Font: +Body (Times New Roman), 10 pt

**Page 26: [226] Formatted**            **Microsoft Office User**            **12/17/19 9:56:00 AM**

Font: (Default) +Body (Times New Roman), Complex Script Font: +Body (Times New Roman), 10 pt

**Page 26: [227] Formatted**            **Microsoft Office User**            **12/17/19 9:56:00 AM**

Font: (Default) +Body (Times New Roman), Complex Script Font: +Body (Times New Roman), 10 pt

**Page 26: [228] Formatted**            **Microsoft Office User**            **12/17/19 9:56:00 AM**

Font: (Default) +Body (Times New Roman), Complex Script Font: +Body (Times New Roman), 10 pt

**Page 26: [229] Formatted**            **Microsoft Office User**            **12/17/19 9:56:00 AM**

Font: (Default) +Body (Times New Roman), Complex Script Font: +Body (Times New Roman), 10 pt

**Page 26: [230] Formatted**            **Microsoft Office User**            **12/17/19 9:56:00 AM**

Font: (Default) +Body (Times New Roman), Complex Script Font: +Body (Times New Roman), 10 pt

**Page 26: [231] Formatted**         **Microsoft Office User**         **12/17/19 9:56:00 AM**

Font: (Default) +Body (Times New Roman), Complex Script Font: +Body (Times New Roman), 10 pt

**Page 26: [232] Formatted**         **Microsoft Office User**         **12/17/19 9:56:00 AM**

Font: (Default) +Body (Times New Roman), Complex Script Font: +Body (Times New Roman), 10 pt

**Page 26: [233] Formatted**         **Microsoft Office User**         **12/17/19 9:56:00 AM**

Font: (Default) +Body (Times New Roman), Complex Script Font: +Body (Times New Roman), 10 pt

**Page 26: [234] Formatted**         **Microsoft Office User**         **12/17/19 9:56:00 AM**

Font: (Default) +Body (Times New Roman), Complex Script Font: +Body (Times New Roman), 10 pt

**Page 26: [235] Formatted**         **Microsoft Office User**         **12/17/19 9:56:00 AM**

Font: (Default) +Body (Times New Roman), Complex Script Font: +Body (Times New Roman), 10 pt

**Page 26: [236] Formatted**         **Microsoft Office User**         **12/17/19 9:56:00 AM**

Font: (Default) +Body (Times New Roman), Complex Script Font: +Body (Times New Roman), 10 pt

**Page 26: [237] Formatted**         **Microsoft Office User**         **12/17/19 9:56:00 AM**

Font: (Default) +Body (Times New Roman), Complex Script Font: +Body (Times New Roman), 10 pt

**Page 26: [238] Formatted**         **Microsoft Office User**         **12/17/19 9:56:00 AM**

Font: (Default) +Body (Times New Roman), Complex Script Font: +Body (Times New Roman), 10 pt

**Page 26: [239] Formatted**         **Microsoft Office User**         **12/17/19 9:56:00 AM**

Font: (Default) +Body (Times New Roman), Complex Script Font: +Body (Times New Roman), 10 pt

**Page 26: [240] Formatted**     **Microsoft Office User**     **12/17/19 9:56:00 AM**

Font: (Default) +Body (Times New Roman), Complex Script Font: +Body (Times New Roman), 10 pt

**Page 26: [240] Formatted**     **Microsoft Office User**     **12/17/19 9:56:00 AM**

Font: (Default) +Body (Times New Roman), Complex Script Font: +Body (Times New Roman), 10 pt

**Page 26: [241] Formatted**     **Microsoft Office User**     **12/17/19 9:56:00 AM**

Font: (Default) +Body (Times New Roman), Complex Script Font: +Body (Times New Roman), 10 pt

**Page 26: [241] Formatted**     **Microsoft Office User**     **12/17/19 9:56:00 AM**

Font: (Default) +Body (Times New Roman), Complex Script Font: +Body (Times New Roman), 10 pt

**Page 26: [242] Formatted**     **Microsoft Office User**     **12/17/19 9:56:00 AM**

Font: (Default) +Body (Times New Roman), Complex Script Font: +Body (Times New Roman), 10 pt

**Page 26: [242] Formatted**     **Microsoft Office User**     **12/17/19 9:56:00 AM**

Font: (Default) +Body (Times New Roman), Complex Script Font: +Body (Times New Roman), 10 pt

**Page 26: [243] Formatted**     **Microsoft Office User**     **12/17/19 9:56:00 AM**

Font: (Default) +Body (Times New Roman), Complex Script Font: +Body (Times New Roman), 10 pt

**Page 26: [243] Formatted**     **Microsoft Office User**     **12/17/19 9:56:00 AM**

Font: (Default) +Body (Times New Roman), Complex Script Font: +Body (Times New Roman), 10 pt

**Page 26: [244] Formatted**     **Microsoft Office User**     **12/17/19 9:56:00 AM**

Font: (Default) +Body (Times New Roman), Complex Script Font: +Body (Times New Roman), 10 pt

**Page 26: [244] Formatted**     **Microsoft Office User**     **12/17/19 9:56:00 AM**

Font: (Default) +Body (Times New Roman), Complex Script Font: +Body (Times New Roman), 10 pt

**Page 26: [245] Formatted**     **Microsoft Office User**     **12/17/19 9:56:00 AM**

Font: (Default) +Body (Times New Roman), Complex Script Font: +Body (Times New Roman), 10 pt

**Page 26: [245] Formatted**     **Microsoft Office User**     **12/17/19 9:56:00 AM**

Font: (Default) +Body (Times New Roman), Complex Script Font: +Body (Times New Roman), 10 pt

**Page 26: [246] Formatted**     **Microsoft Office User**     **12/17/19 9:56:00 AM**

Font: (Default) +Body (Times New Roman), Complex Script Font: +Body (Times New Roman), 10 pt

**Page 26: [246] Formatted**     **Microsoft Office User**     **12/17/19 9:56:00 AM**

Font: (Default) +Body (Times New Roman), Complex Script Font: +Body (Times New Roman), 10 pt

**Page 26: [247] Formatted**     **Microsoft Office User**     **12/17/19 9:56:00 AM**

Font: (Default) +Body (Times New Roman), Complex Script Font: +Body (Times New Roman), 10 pt

**Page 26: [247] Formatted**     **Microsoft Office User**     **12/17/19 9:56:00 AM**

Font: (Default) +Body (Times New Roman), Complex Script Font: +Body (Times New Roman), 10 pt

**Page 26: [248] Formatted**       **Microsoft Office User**       **12/17/19 9:56:00 AM**

Font: (Default) +Body (Times New Roman), Complex Script Font: +Body (Times New Roman), 10 pt

**Page 26: [248] Formatted**       **Microsoft Office User**       **12/17/19 9:56:00 AM**

Font: (Default) +Body (Times New Roman), Complex Script Font: +Body (Times New Roman), 10 pt

**Page 26: [249] Formatted**       **Microsoft Office User**       **12/17/19 9:56:00 AM**

Font: (Default) +Body (Times New Roman), Complex Script Font: +Body (Times New Roman), 10 pt

**Page 26: [249] Formatted**       **Microsoft Office User**       **12/17/19 9:56:00 AM**

Font: (Default) +Body (Times New Roman), Complex Script Font: +Body (Times New Roman), 10 pt

**Page 26: [250] Formatted**       **Microsoft Office User**       **12/17/19 9:56:00 AM**

Font: (Default) +Body (Times New Roman), Complex Script Font: +Body (Times New Roman), 10 pt

**Page 26: [250] Formatted**       **Microsoft Office User**       **12/17/19 9:56:00 AM**

Font: (Default) +Body (Times New Roman), Complex Script Font: +Body (Times New Roman), 10 pt

**Page 26: [251] Formatted**       **Microsoft Office User**       **12/17/19 9:56:00 AM**

Font: (Default) +Body (Times New Roman), Complex Script Font: +Body (Times New Roman), 10 pt

**Page 26: [251] Formatted**       **Microsoft Office User**       **12/17/19 9:56:00 AM**

Font: (Default) +Body (Times New Roman), Complex Script Font: +Body (Times New Roman), 10 pt

**Page 26: [252] Formatted**       **Microsoft Office User**       **12/17/19 9:56:00 AM**

Font: (Default) +Body (Times New Roman), Complex Script Font: +Body (Times New Roman), 10 pt

**Page 26: [252] Formatted**       **Microsoft Office User**       **12/17/19 9:56:00 AM**

Font: (Default) +Body (Times New Roman), Complex Script Font: +Body (Times New Roman), 10 pt

**Page 26: [253] Formatted**       **Microsoft Office User**       **12/17/19 9:56:00 AM**

Font: (Default) +Body (Times New Roman), Complex Script Font: +Body (Times New Roman), 10 pt

**Page 26: [253] Formatted**       **Microsoft Office User**       **12/17/19 9:56:00 AM**

Font: (Default) +Body (Times New Roman), Complex Script Font: +Body (Times New Roman), 10 pt

**Page 26: [254] Formatted**       **Microsoft Office User**       **12/17/19 9:56:00 AM**

Font: (Default) +Body (Times New Roman), Complex Script Font: +Body (Times New Roman), 10 pt

**Page 26: [254] Formatted**       **Microsoft Office User**       **12/17/19 9:56:00 AM**

Font: (Default) +Body (Times New Roman), Complex Script Font: +Body (Times New Roman), 10 pt

**Page 26: [255] Formatted**       **Microsoft Office User**       **12/17/19 9:56:00 AM**

Font: (Default) +Body (Times New Roman), Complex Script Font: +Body (Times New Roman), 10 pt

**Page 26: [255] Formatted**       **Microsoft Office User**       **12/17/19 9:56:00 AM**

Font: (Default) +Body (Times New Roman), Complex Script Font: +Body (Times New Roman), 10 pt

**Page 49: [256] Formatted**      **Microsoft Office User**      **12/17/19 2:39:00 PM**

Font: (Default) +Body (Times New Roman), Complex Script Font: +Body (Times New Roman), 10 pt

**Page 49: [257] Formatted**      **Microsoft Office User**      **12/17/19 2:39:00 PM**

Font: (Default) +Body (Times New Roman), Complex Script Font: +Body (Times New Roman), 10 pt

**Page 49: [258] Formatted**      **Microsoft Office User**      **12/17/19 2:39:00 PM**

Font: (Default) +Body (Times New Roman), Complex Script Font: +Body (Times New Roman), 10 pt

**Page 49: [259] Formatted**      **Microsoft Office User**      **12/17/19 2:39:00 PM**

Font: (Default) +Body (Times New Roman), Complex Script Font: +Body (Times New Roman), 10 pt

**Page 49: [260] Formatted**      **Microsoft Office User**      **12/17/19 2:39:00 PM**

Font: (Default) +Body (Times New Roman), Complex Script Font: +Body (Times New Roman), 10 pt

**Page 49: [261] Formatted**      **Microsoft Office User**      **12/17/19 2:39:00 PM**

Font: (Default) +Body (Times New Roman), Complex Script Font: +Body (Times New Roman), 10 pt

**Page 49: [262] Formatted**      **Microsoft Office User**      **12/17/19 2:39:00 PM**

Font: (Default) +Body (Times New Roman), Complex Script Font: +Body (Times New Roman), 10 pt

**Page 49: [263] Formatted**      **Microsoft Office User**      **12/17/19 2:39:00 PM**

Font: (Default) +Body (Times New Roman), Complex Script Font: +Body (Times New Roman), 10 pt

| **Page 49: [264] Formatted** | **Microsoft Office User** | **12/17/19 2:39:00 PM** |
| --- | --- | --- |

Font: (Default) +Body (Times New Roman), Complex Script Font: +Body (Times New Roman), 10 pt

| **Page 49: [264] Formatted** | **Microsoft Office User** | **12/17/19 2:39:00 PM** |
| --- | --- | --- |

Font: (Default) +Body (Times New Roman), Complex Script Font: +Body (Times New Roman), 10 pt

| **Page 49: [265] Formatted** | **Microsoft Office User** | **12/17/19 2:39:00 PM** |
| --- | --- | --- |

Font: (Default) +Body (Times New Roman), Complex Script Font: +Body (Times New Roman), 10 pt

| **Page 49: [265] Formatted** | **Microsoft Office User** | **12/17/19 2:39:00 PM** |
| --- | --- | --- |

Font: (Default) +Body (Times New Roman), Complex Script Font: +Body (Times New Roman), 10 pt

| **Page 49: [266] Formatted** | **Microsoft Office User** | **12/17/19 2:39:00 PM** |
| --- | --- | --- |

Font: (Default) +Body (Times New Roman), Complex Script Font: +Body (Times New Roman), 10 pt

| **Page 49: [266] Formatted** | **Microsoft Office User** | **12/17/19 2:39:00 PM** |
| --- | --- | --- |

Font: (Default) +Body (Times New Roman), Complex Script Font: +Body (Times New Roman), 10 pt

| **Page 49: [267] Formatted** | **Microsoft Office User** | **12/17/19 2:39:00 PM** |
| --- | --- | --- |

Font: (Default) +Body (Times New Roman), Complex Script Font: +Body (Times New Roman), 10 pt

| **Page 49: [267] Formatted** | **Microsoft Office User** | **12/17/19 2:39:00 PM** |
| --- | --- | --- |

Font: (Default) +Body (Times New Roman), Complex Script Font: +Body (Times New Roman), 10 pt

| **Page 49: [268] Formatted** | **Microsoft Office User** | **12/17/19 2:39:00 PM** |
| --- | --- | --- |

Font: (Default) +Body (Times New Roman), Complex Script Font: +Body (Times New Roman), 10 pt

| **Page 49: [268] Formatted** | **Microsoft Office User** | **12/17/19 2:39:00 PM** |
| --- | --- | --- |

Font: (Default) +Body (Times New Roman), Complex Script Font: +Body (Times New Roman), 10 pt

| **Page 49: [269] Formatted** | **Microsoft Office User** | **12/17/19 2:39:00 PM** |
| --- | --- | --- |

Font: (Default) +Body (Times New Roman), Complex Script Font: +Body (Times New Roman), 10 pt

| **Page 49: [269] Formatted** | **Microsoft Office User** | **12/17/19 2:39:00 PM** |
| --- | --- | --- |

Font: (Default) +Body (Times New Roman), Complex Script Font: +Body (Times New Roman), 10 pt

| **Page 49: [270] Formatted** | **Microsoft Office User** | **12/17/19 2:39:00 PM** |
| --- | --- | --- |

Font: (Default) +Body (Times New Roman), Complex Script Font: +Body (Times New Roman), 10 pt

| **Page 49: [270] Formatted** | **Microsoft Office User** | **12/17/19 2:39:00 PM** |
| --- | --- | --- |

Font: (Default) +Body (Times New Roman), Complex Script Font: +Body (Times New Roman), 10 pt

| **Page 49: [271] Formatted** | **Microsoft Office User** | **12/17/19 2:39:00 PM** |
| --- | --- | --- |

Font: (Default) +Body (Times New Roman), Complex Script Font: +Body (Times New Roman), 10 pt

| **Page 49: [271] Formatted** | **Microsoft Office User** | **12/17/19 2:39:00 PM** |
| --- | --- | --- |

Font: (Default) +Body (Times New Roman), Complex Script Font: +Body (Times New Roman), 10 pt

| Page 49: [272] Formatted | Microsoft Office User | 12/17/19 2:39:00 PM |
|---|---|---|

Font: (Default) +Body (Times New Roman), Complex Script Font: +Body (Times New Roman), 10 pt

| Page 49: [272] Formatted | Microsoft Office User | 12/17/19 2:39:00 PM |
|---|---|---|

Font: (Default) +Body (Times New Roman), Complex Script Font: +Body (Times New Roman), 10 pt

| Page 49: [273] Formatted | Microsoft Office User | 12/17/19 2:39:00 PM |
|---|---|---|

Font: (Default) +Body (Times New Roman), Complex Script Font: +Body (Times New Roman), 10 pt

| Page 49: [273] Formatted | Microsoft Office User | 12/17/19 2:39:00 PM |
|---|---|---|

Font: (Default) +Body (Times New Roman), Complex Script Font: +Body (Times New Roman), 10 pt

| Page 49: [274] Formatted | Microsoft Office User | 12/17/19 2:39:00 PM |
|---|---|---|

Font: (Default) +Body (Times New Roman), Complex Script Font: +Body (Times New Roman), 10 pt

| Page 49: [274] Formatted | Microsoft Office User | 12/17/19 2:39:00 PM |
|---|---|---|

Font: (Default) +Body (Times New Roman), Complex Script Font: +Body (Times New Roman), 10 pt

| Page 49: [275] Formatted | Microsoft Office User | 12/17/19 2:39:00 PM |
|---|---|---|

Font: (Default) +Body (Times New Roman), Complex Script Font: +Body (Times New Roman), 10 pt

| Page 49: [275] Formatted | Microsoft Office User | 12/17/19 2:39:00 PM |
|---|---|---|

Font: (Default) +Body (Times New Roman), Complex Script Font: +Body (Times New Roman), 10 pt

| Page 49: [276] Formatted | Microsoft Office User | 12/17/19 2:39:00 PM |
|---|---|---|

Font: (Default) +Body (Times New Roman), Complex Script Font: +Body (Times New Roman), 10 pt

| Page 49: [276] Formatted | Microsoft Office User | 12/17/19 2:39:00 PM |
|---|---|---|

Font: (Default) +Body (Times New Roman), Complex Script Font: +Body (Times New Roman), 10 pt

| Page 49: [277] Formatted | Microsoft Office User | 12/17/19 2:39:00 PM |
|---|---|---|

Font: (Default) +Body (Times New Roman), Complex Script Font: +Body (Times New Roman), 10 pt

| Page 49: [277] Formatted | Microsoft Office User | 12/17/19 2:39:00 PM |
|---|---|---|

Font: (Default) +Body (Times New Roman), Complex Script Font: +Body (Times New Roman), 10 pt

| Page 49: [278] Formatted | Microsoft Office User | 12/17/19 2:39:00 PM |
|---|---|---|

Font: (Default) +Body (Times New Roman), Complex Script Font: +Body (Times New Roman), 10 pt

| Page 49: [278] Formatted | Microsoft Office User | 12/17/19 2:39:00 PM |
|---|---|---|

Font: (Default) +Body (Times New Roman), Complex Script Font: +Body (Times New Roman), 10 pt

| Page 49: [279] Formatted | Microsoft Office User | 12/17/19 2:39:00 PM |
|---|---|---|

Font: (Default) +Body (Times New Roman), Complex Script Font: +Body (Times New Roman), 10 pt

| Page 49: [279] Formatted | Microsoft Office User | 12/17/19 2:39:00 PM |
|---|---|---|

Font: (Default) +Body (Times New Roman), Complex Script Font: +Body (Times New Roman), 10 pt

**Page 49: [280] Formatted** | **Microsoft Office User** | **12/17/19 2:39:00 PM**

Font: (Default) +Body (Times New Roman), Complex Script Font: +Body (Times New Roman), 10 pt

**Page 49: [280] Formatted** | **Microsoft Office User** | **12/17/19 2:39:00 PM**

Font: (Default) +Body (Times New Roman), Complex Script Font: +Body (Times New Roman), 10 pt

**Page 49: [281] Formatted** | **Microsoft Office User** | **12/17/19 2:39:00 PM**

Font: (Default) +Body (Times New Roman), Complex Script Font: +Body (Times New Roman), 10 pt

**Page 49: [281] Formatted** | **Microsoft Office User** | **12/17/19 2:39:00 PM**

Font: (Default) +Body (Times New Roman), Complex Script Font: +Body (Times New Roman), 10 pt

---

## Author Response (AR2)

Reviewer 1, revision 2

The reviewer's comments are in italics. Our responses are in normal font.

*I commend the authors for making significant edits to their manuscript, including valuable new analysis of their results and constraining uncertainties. The new figures are particularly useful.*

*One minor final suggestion: It seems to me that differences in the spatial distributions of ice thickness in Fig 9 are likely caused by the lower sensitivity of the ice age-thickness relationship towards the upper end of the thickness scale (i.e. Fig 12). Rather than noting the differences should be the subject of future research at Line 301, it would be useful here to speculate directly on the cause of this ice thickness underestimation.*

*Otherwise, all OK from me. Congratulations on a fine paper! Jack*

We thank the reviewer for his comments and suggestions, which definitely improved the manuscript.

For the reviewer's minor comment, **we modified the text** to:

"Though the comparison to the CryoSat-2 ice products show overall agreement in both thickness and volume, further investigation and analysis shows that there are rather apparent differences in the ice thickness retrieval spatial distributions as shown in Figure 9. It appears the IceAgeDerived ice thickness underestimates the ice thickness for the older ice while overestimates the ice thickness for the new ice with comparison to CryoSat-2. The underestimation of ice thickness north of the Canadian Archipelago and the Greenland from the IceAgeDerived may be attributed to the lower sensitivity of sea ice age   thickness towards older sea ice, as will be discussed later and shown in Figure 12. This reduction of sensitivity may come from higher uncertainty with older sea ice age because of higher uncertainty with longer lagrangian track of sea ice parcels in theory. Such uncertainty estimation is not available in the current sea ice age product.  This reduction of sensitivity may also be related to the fact that oldest age of all possible ice parcels within each grid cell is assigned to the cell, and thus the ice age may overestimate the sea ice age of some cells. It should be also noted that CryoSat-2 also has relatively high uncertainties for very thin and very thick sea ice. In total, these underestimates and overestimates may balance off in the overall mean ice thickness and ice volume comparisons. Diagnosis and resolving this difference is worth further investigation."

Reviewer 2, revision 2

The reviewer's comments are in italics. Our responses are in normal font.

*The authors have reacted on the review comments in detailed manner and made modifications to the manuscript accordingly. I was pleased to see they included CryoSat-2 and EnviSat comparisons as requested. However, there are still some minor issues with the revised version that I would like to raise.*

*1) Formatting of sections. Reviewer 1 raised a question about the set of equations in Section 2.1.1 that it would better belong under Methods (than Data), which was left unanswered or uncorrected.*

In response to Reviewer 1's comments on Eqs. 1 to 4, **we rewrote the paragraphs** explaining those equations, and referred the readers to Rothrock et al. (2008) for more details on the equations 1-4 in the revised manuscript.

Nevertheless, **we added the following text** regarding this reviewer's comment and also Reviewer 1's comments that we did not address directly before,
"$A(\tau)$ as the ice thickness annual cycle, $I(t-1988)$ as the ice thickness interannual change centered around 1988 and $\bar{I}$ with a value of −0.12 m is the mean of I", and
"Eq.2 provides the interannual change ($I(t–1988)$) with the annual cycle ($A(\tau)$) superimposed in averaged ice thickness over the SCICEX box. Eqs 1, 2, 3, and 4 were taken from RPW08, in which details on the derivation of these equations are available.. Eq. 2 will be used to calculate the monthly mean ice thickness in this study."

Regarding to the suggestion to move this part from "data" to "method" from both reviewers, we think there might be some confusion that we did not clearly address in the manuscript. We use the submarine ice thickness data available to us, and we did not generate these ice thickness data. These equations and associated text are used to help the readers to better understand how the submarine ice thickness are derived, and all the equations are from paper by Rothrock et al. (2008). **This is noted in the revised manuscript.** For this reason, we think it is appropriate to keep this section in the "data". We thank the reviewer's suggestion, though.

*Also, e.g. Section 2.1.2 could still profit from a small rearrangement as now there is first mentioned ICESat, then CryoSat-2 (Kwok) then OTIM/AVHRR and then CryoSat-2 (others).*

Good point. We divided the ICESat and CryoSat-2 in RK18, and moved the ICESat part to section 2.1.1. In section 2.1.2, we moved the CryoSat-2 in RK18 close to other CryoSat-2 part.

*2) Roles of products. PIOMAS and OTIM. PIOMAS was mentioned in Section 2.1.1 Data for Algorithm Development, OTIM in 2.1.2 Data for Evaluation/Validation, both mentioned in 2.2 Method. Is PIOMAS used in algorithm development? This might be a question related to previous point (1). Also different use of CryoSat-2 products [3 (AWI, CPOM, NASA GSFC) + 1 (Kwok)] is confusing.*

Good point. PIOMAS is not used in the algorithm development, so **we moved the PIOMAS to section 2.1.2**. We moved the RK18 CryoSat-2 close to all other CryoSat-2 products and Envisat product, and clarify the difference between CryoSat-2 from Kwok and other three products in the revised manuscript.

*3) Tables. There is a AWI column in Table 4, but this was the ESA CCI EnviSat?*

Yes, it is. Thanks for pointing out this error. We **corrected it** in the revised manuscript.

*4) Figures. Would you call Figures 7 and 8 differences as the figure titles state? Wouldn't scatterplot be more about correlation. Figure 9, differences would make more sense with a different colormap (than the one used for actual thicknesses) but this is only a very minor suggestion.*

These are all good suggestions. We **changed the titles** of Figure 7 and 8, **added slope, intercept, and R2** in Figure 7, 8, 10, and 11. For Figure 9, we **changed the colormap** for the difference figures.

We also changed the title of Figure 18, and updated Figure 18 in the revised manuscript, and updated the captions of some figures.

*Discussion and conclusions is good and the manuscript in overall raises an interesting set of additional research questions. The manuscript also lists potential points for improvement to their method, and provides a good setup for future work.*

Thanks.

We also read through the manuscript, and made many editorial changes.

[revised manuscript text omitted]

from

| Page 6: [2] Deleted | YINGHUI LIU | 2/26/20 4:04:00 PM |
|---|---|---|

 are

| Page 6: [3] Formatted | Jeff Key | 3/1/20 4:49:00 PM |
|---|---|---|

English (US)

| Page 6: [4] Deleted | Jeff Key | 3/1/20 6:48:00 PM |
|---|---|---|

| Page 6: [5] Formatted | Jeff Key | 3/1/20 4:49:00 PM |
|---|---|---|

English (US)

| Page 6: [6] Formatted | Jeff Key | 3/1/20 4:49:00 PM |
|---|---|---|

English (US)

| Page 6: [7] Deleted | YINGHUI LIU | 2/28/20 4:20:00 PM |
|---|---|---|

d

| Page 6: [7] Deleted | YINGHUI LIU | 2/28/20 4:20:00 PM |
|---|---|---|

d

| Page 6: [8] Deleted | YINGHUI LIU | 2/28/20 4:23:00 PM |
|---|---|---|

at

| Page 6: [9] Deleted | YINGHUI LIU | 2/29/20 9:01:00 AM |
|---|---|---|

for the times between

| Page 6: [10] Deleted | YINGHUI LIU | 2/28/20 7:22:00 PM |
|---|---|---|

. Each ice draft profile segment is

| Page 6: [10] Deleted | YINGHUI LIU | 2/28/20 7:22:00 PM |
| --- | --- | --- |

. Each ice draft profile segment is

| Page 6: [10] Deleted | YINGHUI LIU | 2/28/20 7:22:00 PM |
| --- | --- | --- |

. Each ice draft profile segment is

| Page 6: [10] Deleted | YINGHUI LIU | 2/28/20 7:22:00 PM |
| --- | --- | --- |

. Each ice draft profile segment is

| Page 33: [11] Deleted | YINGHUI LIU | 2/27/20 9:26:00 PM |
| --- | --- | --- |